# ROBUST MULTI-AGENT REINFORCEMENT LEARNING WITH STATE UNCERTAINTIES

## ABSTRACT

In real-world multi-agent reinforcement learning (MARL) applications, agents may not have perfect state information (e.g., due to inaccurate measurement or malicious attacks), which challenges the robustness of agents' policies. Though robustness is getting important in MARL deployment, little prior work has studied state uncertainties in MARL, neither in problem formulation nor algorithm design. Motivated by this robustness issue, we study the problem of MARL with state uncertainty in this work. We provide the first attempt to the theoretical and empirical analysis of this challenging problem. We first model the problem as a Markov Game with state perturbation adversaries (MG-SPA), and introduce Robust Equilibrium as the solution concept. We conduct a fundamental analysis regarding MG-SPA and give conditions under which such an equilibrium exists. Then we propose a robust multi-agent Q-learning (RMAQ) algorithm to find such an equilibrium, with convergence guarantees. To handle high-dimensional state-action space, we design a robust multi-agent actor-critic (RMAAC) algorithm based on an analytical expression of the policy gradient derived in the paper. Our experiments show that the proposed RMAQ algorithm converges to the optimal value function; our RMAAC algorithm outperforms several MARL methods that do not consider state uncertainty in several multi-agent environments.

## 1 INTRODUCTION

Reinforcement Learning (RL) recently has achieved remarkable success in many decision-making problems, such as robotics, autonomous driving, traffic control, and game playing (Espeholt et al., 2018; Silver et al., 2017; Mnih et al., 2015). However, in real-world applications, the agent may face *state uncertainty* in that accurate information about the state is unavailable. This uncertainty may be caused by unavoidable sensor measurement errors, noise, missing information, communication issues, and/or malicious attacks. A policy not robust to state uncertainty can result in unsafe behaviors and even catastrophic out-

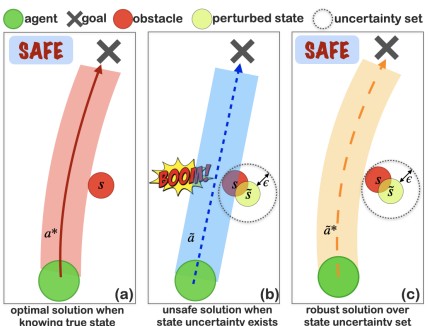

Figure 1: Motivation of considering state uncertainty in RL.

comes. For instance, consider the path planning problem shown in Figure 1, where the agent (green ball) observes the position of an obstacle (red ball) through sensors and plans a safe (no collision) and shortest path to the goal (black cross). In Figure 1-(a), the agent can observe the true state $s$ (red ball) and choose an optimal and collision-free curve $a^*$ (in red) tangent to the obstacle. In comparison, when the agent can only observe the perturbed state $\tilde{s}$ (yellow ball) caused by inaccurate sensing or state perturbation adversaries (Figure 1-(b)), it will choose a straight line $\tilde{a}$ (in blue) as the shortest and collision-free path tangent to $\tilde{s}$. However, by following $\tilde{a}$, the agent actually crashes into the obstacle. To avoid collision in the worst case, one can construct a state uncertainty set that contains the true state based on the observed state. Then the robustly optimal path under state uncertainty becomes the yellow curve $\tilde{a}^*$ tangent to the uncertainty set, as shown in Figure 1-(c).

In single-agent RL, imperfect information about the state has been studied in the literature of partially observable Markov decision process (POMDP) (Kaelbling et al., 1998). However, as pointed out in recent literature (Huang et al., 2017; Kos & Song, 2017; Yu et al., 2021b; Zhang et al., 2020a), the conditional observation probabilities in POMDP cannot capture the *worst-case*

(or adversarial) scenario, and the learned policy without considering state uncertainties may fail to achieve the agent's goal. Similarly, the existing literature of decentralized partially observable Markov decision process (Dec-POMDP) (Oliehoek et al., 2016) does not provide theoretical analysis or algorithmic tools for MARL under worst-case state uncertainties either. Dealing with state uncertainty becomes even more challenging for Multi-Agent Reinforcement Learning (MARL), where each agent aims to maximize its own total return during the interaction with other agents and the environment. Even one agent receives misleading state information, its action affects both its own return and the other agents' returns (Zhang et al., 2020b) and may result in catastrophic failure.

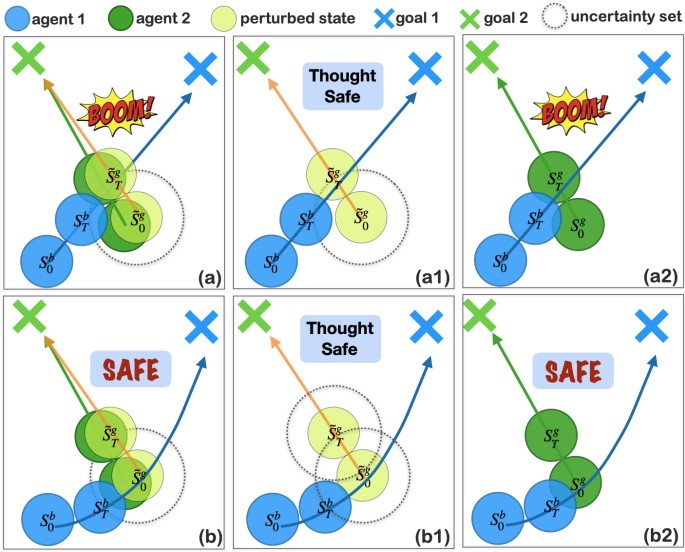

Figure 2: Motivation of considering state uncertainty in MARL.

To better illustrate the effect of state uncertainty in MARL, the path planning problem in Figure 1 is modified such that two agents are trying to reach their individual goals without collision (a penalty or negative reward applied). When the blue agent knows the true position $s_0^g$ (the subscript denotes time, which starts from 0) of the green agent, it will get around the green agent to quickly reach its goal without collision. However, in Figure 2-(a), when the blue agent can only observe the perturbed position $\tilde{s}_0^g$ (yellow circle) of the green agent, it would choose a straight line that it thought safe (Figure 2-(a1)), which eventually leads to a crash (Figure 2-(a2)).

In Figure 2-(b), the blue agent adopts a robust trajectory by considering a state uncertainty set based on its observation. As shown in Figure 2-(b1), there is no overlap between $(s_0^b, \tilde{s}_0^g)$ or $(s_T^b, \tilde{s}_T^g)$. Since the uncertainty sets centered at $\tilde{s}_0^g$ and $\tilde{s}_T^g$ (the dotted circles) include the true state of the green agent, this robust trajectory also ensures no collision between $(s_0^b, s_0^g)$ or $(s_T^b, s_T^g)$. The blue agent considers the interactions with the green agent to ensure no collisions at any time. Therefore, it is necessary to consider state uncertainty in a multi-agent setting where the dynamics of other agents are considered.

In this work, we develop a robust MARL framework that accounts for state uncertainty. Specifically, we model the problem of MARL with state uncertainty as a Markov Game with state perturbation adversaries (MG-SPA), in which each agent is associated with a state perturbation adversary. One state perturbation adversary always plays against its corresponding agent by preventing the agent from knowing the true state accurately. We analyze the MARL problem with adversarial or worst-case state perturbations. Compared to single-agent RL, MARL is more challenging due to the interactions among agents and the necessity of studying equilibrium policies (Nash, 1951; McKelvey & McLennan, 1996; Slantchev, 2008; Daskalakis et al., 2009; Etessami & Yannakakis, 2010). The contributions of this work are summarized as follows.

**Contributions:** To the best of our knowledge, this work is the first attempt to systematically characterize state uncertainties in MARL and provide both theoretical and empirical analysis. First, we formulate the MARL problem with state uncertainty as a Markov Game with state perturbation adversaries (MG-SPA). We define the solution concept of the game as a Robust Equilibrium, where all players including the agents and the adversaries use policies that no one has an incentive to deviate. In an MG-SPA, each agent not only aims to maximize its return when considering other agents' actions but also needs to act against all state perturbation adversaries. Therefore, a Robust Equilibrium policy of one agent is robust to the state uncertainties. Second, we study its fundamental properties and prove the existence of a Robust Equilibrium under certain conditions. We develop a robust multi-agent Q-learning (RMAQ) algorithm with convergence guarantee, and an actor-critic (RMAAC) algorithm for computing a robust equilibrium policy in an MG-SPA. Finally, we conduct experiments in a two-player game to validate the convergence of the proposed Q-learning method RMAQ. We show that our RMAQ and RMAAC algorithms can learn robust policies that outperform baselines under state perturbations in multi-agent environments.

## 2 RELATED WORK

**Robust Reinforcement Learning:** Recent robust reinforcement learning studied different types of uncertainties, such as action uncertainties (Tessler et al., 2019) and transition kernel uncertainties (Sinha et al., 2020; Yu et al., 2021b; Hu et al., 2020; Wang & Zou, 2021; Lim & Autef, 2019; Nisioti et al., 2021; He et al., 2022). Some recent attempts about adversarial state perturbations for single-agent validated the importance of considering state uncertainty and improving the robustness of the learned policy in Deep RL (Huang et al., 2017; Lin et al., 2017; Zhang et al., 2020a; 2021; Everett et al., 2021). The works of Zhang et al. (2020a; 2021) formulate the state perturbation in single agent RL as a modified Markov decision process, then study the robustness of single agent RL policies. The works of Huang et al. (2017) and Lin et al. (2017) show that adversarial state perturbation undermines the performance of neural network policies in single agent reinforcement learning and proposes different single agent attack strategies. In this work, we consider the more challenging problem of adversarial state perturbation for MARL, when the environment of an individual agent is non-stationary with other agents' changing policies during the training process.

**Robust Multi-Agent Reinforcement Learning:** There is very limited literature for the solution concept or theoretical analysis when considering adversarial state perturbations in MARL. Other types of uncertainties have been investigated in the literature, such as uncertainties about training partner's type (Shen & How, 2021) and the other agents' policies (Li et al., 2019; Sun et al., 2021; van der Heiden et al., 2020), and reward uncertainties (Zhang et al., 2020b). The policy considered in these papers relies on the current true state information, hence, the robust MARL considered in this work is fundamentally different since the agents do not know the true state information. Dec-POMDP enables a team of agents to optimize policies with the partial observable states (Oliehoek et al., 2016; Chen et al., 2022). The work of Lin et al. (2020) studies state perturbation in cooperative MARL, and proposes an attack method to attack the state of one single agent in order to decrease the team reward. In contrast, we consider the worst-case scenario that the state of every agent can be perturbed by an adversary and focus on theoretical analysis of robust MARL including the existence of optimal value function and Robust Equilibrium (RE). Our work provides formal definitions of the state uncertainty challenge in MARL, and derives both theoretical analysis and practical algorithms.

**Game Theory and MARL:** MARL shares theoretical foundations with game theory research field and literature review has been provided to understand MARL from a game theoretical perspective (Yang & Wang, 2020). A Markov game, sometimes called a stochastic game models the interaction between multiple agents (Owen, 2013; Littman, 1994). Algorithms to compute the Nash Equilibrium (NE) in Nash Q-learning (Hu & Wellman, 2003), Dec-POMDP (Oliehoek et al., 2016) or POSG (partially observable stochastic game) and analysis assuming that NE exists (Chades et al., 2002; Hansen et al., 2004; Nair et al., 2002) have been developed in the literature without proving the conditions for the existence of NE. The main theoretical contributions of this work include proving conditions under which the proposed MG-SPA has Robust Equilibrium solutions, and convergence analysis of our proposed robust Q-learning algorithm. This is the first attempts to analyze fundamental properties of MARL under adversarial state uncertainties.

## 3 METHODOLOGY

### 3.1 MARKOV GAME WITH STATE PERTURBATION ADVERSARIES (MG-SPA)

**Preliminary:** A Markov Game (MG) $G$ is defined as $(\mathcal{N}, \{S^i\}_{i\in\mathcal{N}}, \{A^i\}_{i\in\mathcal{N}}, \{r^i\}_{i\in\mathcal{N}}, p, \gamma)$, where $\mathcal{N}$ is a set of $N$ agents, $S^i$ and $A^i$ are the state space, action space of agent $i$, respectively (Littman, 1994; Owen, 2013). $\gamma \in [0,1)$ is the discounting factor. $S = S^1 \times \cdots \times S^N$ is the joint state space. $A = A^1 \times \cdots \times A^N$ is the joint action space. The state transition $p : S \times A \to \Delta(S)$ is controlled by the current state and joint action, where $\Delta(S)$ represents the set of all probability distributions over the joint state space $S$. Each agent has a reward function, $r^i : S \times A \to \mathbb{R}$. At time $t$, agent $i$ chooses its action $a_t^i$ according to a policy $\pi^i : S^i \to \Delta(A^i)$. The agents' joint policy $\pi = \prod_{i\in\mathcal{N}} \pi^i : S \to \Delta(A)$.

**Notations:** We use a tuple $\tilde{G} := (\mathcal{N}, \mathcal{M}, \{S^i\}_{i\in\mathcal{N}}, \{A^i\}_{i\in\mathcal{N}}, \{B^{\tilde{i}}\}_{\tilde{i}\in\mathcal{M}}, \{r^i\}_{i\in\mathcal{N}}, p, f, \gamma)$ to denote a Markov game with state perturbation adversaries (MG-SPA). In an MG-SPA, we introduce an additional set of adversaries $\mathcal{M} = \{\tilde{1}, \cdots, \tilde{N}\}$ to a Markov game (MG) with an agent set $\mathcal{N}$. Each agent $i$ is associated with an adversary $\tilde{i}$ and a true state $s^i \in S^i$ if without adversarial perturbation. Each adversary $\tilde{i}$ is associated with an action $b^{\tilde{i}} \in B^{\tilde{i}}$ and the same state $s^i \in S^i$ as agent $i$. We

define the adversaries' joint action as $b = (b^{\tilde{1}}, ..., b^{\tilde{N}}) \in B$, $B = B^{\tilde{1}} \times \cdots \times B^{\tilde{N}}$. At time $t$, adversary $\tilde{i}$ can manipulate the corresponding agent $i$'s state. Once adversary $\tilde{i}$ gets state $s_t^i$, it chooses an action $b_t^{\tilde{i}}$ according to a policy $\rho^{\tilde{i}} : S^i \to \Delta(B^{\tilde{i}})$. According to a perturbation function $f$, adversary $\tilde{i}$ perturbs state $s_t^i$ to $\tilde{s}_t^i = f(s_t^i, b_t^{\tilde{i}}) \in S^i$. Here we define the adversaries' joint policy $\rho = \prod_{\tilde{i} \in \mathcal{M}} \rho^{\tilde{i}} : S \to \Delta(B)$. The definitions of agent action and agents' joint action are the same as their definitions in an MG. Agent $i$ chooses its action $a_t^i$ with $\tilde{s}_t^i$ according to a policy $\pi^i(a_t^i | \tilde{s}_t^i)$, $\pi^i : S^i \to \Delta(A^i)$. Agents execute the agents' joint action $a_t$, then at time $t+1$, the joint state $s_t$ turns to the next state $s_{t+1}$ according to a transition probability function $p : S \times A \times B \to \Delta(S)$. Each agent $i$ gets a reward according to a state-wise reward function $r_t^i : S \times A \times B \to \mathbb{R}$. Each adversary $\tilde{i}$ gets an opposite reward $-r_t^i$. In an MG, the transition probability function and reward function are considered as the model of the game. In an MG-SPA, the perturbation function $f$ is also considered as a part of the model, i.e., the model of an MG-SPA is consisted of $f, p$ and $\{r^i\}_{i \in \mathcal{N}}$. To incorporate realistic settings into our analysis, we restrict the power of each adversary, which is a common assumption for state perturbation adversaries in the RL literature (Zhang et al., 2020a; 2021; Everett et al., 2021). We define perturbation constraints $\tilde{s}^i \in \mathcal{B}_{dist}(\epsilon, s^i) \subset S^i$ to restrict the adversary $\tilde{i}$ to perturb a state only to a predefined set of states. $\mathcal{B}_{dist}(\epsilon, s^i)$ is a $\epsilon$-radius ball measured in metric $dist(\cdot, \cdot)$, which is often chosen to be the $l$-norm distance: $dist(s^i, \tilde{s}^i) = \|s^i - \tilde{s}^i\|_l$. We omit the subscript $dist$ in the following context. For each agent $i$, it attempts to maximize its expected sum of discounted rewards, i.e. its objective function $J^i(\pi, \rho) = \mathbb{E}\left[\sum_{t=1}^{\infty} \gamma^{t-1} r_t^i(s_t, a_t, b_t) | s_1 = s, a_t \sim \pi(\cdot|\tilde{s}_t), b_t \sim \rho(\cdot|s_t)\right]$. Each adversary $\tilde{i}$ aims to minimize the objective function of agent $i$ and is considered as receiving an opposite reward of agent $i$, which also leads to a value function $-J^i(\pi, \rho)$ for adversary $\tilde{i}$. We further define the value functions in an MG-SPA as follows:

**Definition 3.1.** *(Value Functions)* $v^{\pi,\rho} = (v^{\pi,\rho,1}, \cdots, v^{\pi,\rho,N}), q^{\pi,\rho} = (q^{\pi,\rho,1}, \cdots, q^{\pi,\rho,N})$ *are defined as the state-value function or value function for short, and the action-value function, respectively. The $i$th element $v^{\pi,\rho,i}$ and $q^{\pi,\rho,i}$ are defined as* $q^{\pi,\rho,i}(s,a,b) = \mathbb{E}\left[\sum_{t=1}^{\infty} \gamma^{t-1} r_t^i | s_1 = s, a_1 = a, b_1 = b, a_t \sim \pi(\cdot|\tilde{s}_t), b_t \sim \rho(\cdot|s_t), \tilde{s}_t^i = f(s_t^i, b_t^{\tilde{i}})\right]$, $v^{\pi,\rho,i}(s) = \mathbb{E}\left[\sum_{t=1}^{\infty} \gamma^{t-1} r_t^i | s_1 = s, a_t \sim \pi(\cdot|\tilde{s}_t), b_t \sim \rho(\cdot|s_t), \tilde{s}_t^i = f(s_t^i, b_t^{\tilde{i}})\right]$, *respectively.*

We name an equilibrium for an MG-SPA as a Robust Equilibrium (RE) and define it as follows:

**Definition 3.2.** *(Robust Equilibrium) Given a Markov game with state perturbation adversaries $\tilde{G}$, a joint policy $d_* = (\pi_*, \rho_*)$ where $\pi_* = (\pi_*^1, \cdots, \pi_*^N)$ and $\rho_* = (\rho_*^{\tilde{1}}, \cdots, \rho_*^{\tilde{N}})$ is said to be in robust equilibrium, or a robust equilibrium, if and only if, for any $i \in \mathcal{N}$, $s \in S$, $v^{(\pi_*^{-i}, \pi_*^i, \rho_*^{-\tilde{i}}, \rho^{\tilde{i}}), i}(s) \geq v^{(\pi_*^{-i}, \pi_*^i, \rho_*^{-\tilde{i}}, \rho_*^{\tilde{i}}), i}(s) \geq v^{(\pi_*^{-i}, \pi^i, \rho_*^{-\tilde{i}}, \rho_*^{\tilde{i}}), i}(s)$ for all $\pi^i$ and $\rho^{\tilde{i}}$, where $-i/-\tilde{i}$ represents the indices of all agents/adversaries except agent $i$/adversary $\tilde{i}$.*

We seek to characterize the optimal value $v_*(s) = (v_*^1(s), \cdots, v_*^N(s))$ defined by $v_*^i(s) = \max_{\pi^i} \min_{\rho^{\tilde{i}}} v^{(\pi_*^{-i}, \pi^i, \rho_*^{-\tilde{i}}, \rho^{\tilde{i}}), i}(s)$. For notation convenience, we use $v^i(s)$ to denote $v^{(\pi_*^{-i}, \pi^i, \rho_*^{-\tilde{i}}, \rho^{\tilde{i}}), i}(s)$. The Bellman Equations of an MG-SPA are in the forms of (1) and (2). The Bellman Equation is a recursion for expected rewards, which helps us identifying or finding an RE.

$$q_*^i(s,a,b) = r^i + \gamma \sum_{s' \in S} p(s'|s,a,b) \max_{\pi^i} \min_{\rho^{\tilde{i}}} \mathbb{E}\left[q_*^i(s',a',b')|a' \sim \pi(\cdot|\tilde{s}), b' \sim \rho(\cdot|s)\right], \quad (1)$$

$$v_*^i(s) = \max_{\pi^i} \min_{\rho^{\tilde{i}}} \mathbb{E}\left[\sum_{s' \in S} p(s'|s,a,b)[r^i(s,a,b) + \gamma v_*^i(s')]|a \sim \pi(\cdot|\tilde{s}), b \sim \rho(\cdot|s)\right], \quad (2)$$

for all $i \in \mathcal{N}$, where $\pi = (\pi^i, \pi_*^{-i})$, $\rho = (\rho^{\tilde{i}}, \rho_*^{-\tilde{i}})$, $\pi_*^{-i}, \rho_*^{-\tilde{i}}$ are in robust equilibrium for $\tilde{G}$. We prove them in the following subsection.

## 3.2 THEORETICAL ANALYSIS OF MG-SPA

**Vector Notations:** To make the analysis easy to read, we follow and extend the vector notations in Puterman (2014). Let $V$ denote the set of bounded real valued functions on $S$ with component-wise partial order and norm $\|v^i\| := \sup_{s \in S} |v^i(s)|$. Let $V_M$ denote the subspace of $V$ of Borel measurable functions. For discrete state space, all real-valued functions are measurable so that $V = V_M$. But

when $S$ is a continuum, $V_M$ is a proper subset of $V$. Let $v = (v^1, \cdots, v^N) \in \mathbb{V}$ be the set of bounded real valued functions on $S \times \cdots \times S$, i.e. the across product of $N$ state set and norm $\|v\| := \sup_j \|v^j\|$. For discrete $S$, let $|S|$ denote the number of elements in $S$. Let $r^i$ denote a $|S|$-vector, with $s$th component $r^i(s)$ which is the expected reward for agent $i$ under state $s$. And $P$ the $|S| \times |S|$ matrix with $(s, s')$th entry given by $p(s'|s)$. We refer to $r_d^i$ as the reward vector of agent $i$, and $P_d$ as the probability transition matrix corresponding to a joint policy $d = (\pi, \rho)$. $r_d^i + \gamma P_d v^i$ is the expected total one-period discounted reward of agent $i$, obtained using the joint policy $d = (\pi, \rho)$. Let $z$ as a list of joint policy $\{d_1, d_2, \cdots\}$ and $P_z^0 = I$, we denote the expected total discounted reward of agent $i$ using $z$ as $v_z^i = \sum_{t=1}^\infty \gamma^{t-1} P_z^{t-1} r_{d_t}^i = r_{d_1}^i + \gamma P_{d_1} r_{d_2}^i + \cdots + \gamma^{n-1} P_{d_1} \cdots P_{d_{n-1}} r_{t_n}^i + \cdots$. Now, we define the following minimax operator which is used in the rest of the paper.

**Definition 3.3.** *(Minimax Operator) For $v^i \in V, s \in S$, we define the nonlinear operator $L^i$ on $v^i(s)$ by $L^i v^i(s) := \max_{\pi^i} \min_{\rho^{\tilde{i}}} [r_d^i + \gamma P_d v^i](s)$, where $d := (\pi_*^{-i}, \pi^i, \rho_*^{-\tilde{i}}, \rho^{\tilde{i}})$. We also define the operator $Lv(s) = L(v^1(s), \cdots, v^N(s)) = (L^1 v^1(s), \cdots, L^N v^N(s))$. Then $L^i v^i$ is a $|S|$-vector, with $s$th component $L^i v^i(s)$.*

For discrete $S$ and bounded $r^i$, it follows from Lemma 5.6.1 in Puterman (2014) that $L^i v^i \in V$ for all $v^i \in V$. Therefore $Lv \in \mathbb{V}$ for all $v \in \mathbb{V}$. And in this paper, we consider the following assumptions in Markov games with state perturbation adversaries.

**Assumption 3.4.** *(1) Bounded rewards; $|r^i(s, a, b)| \leq M^i < M < \infty$ for all $i \in \mathcal{N}$, $a \in A$, $b \in B$ and $s \in S$. (2) Finite state and action spaces; all $S^i, A^i, B^{\tilde{i}}$ are finite. (3) Stationary transition probability and reward functions. (4) $f$ is a bijection when $s^i$ is fixed. (5) All agents share one common reward function.*

The next two propositions characterize the properties of the minimax operator $L$ and space $\mathbb{V}$. They have been proved in Appendix A.2.

**Proposition 3.5.** *(Contraction mapping) Suppose $0 \leq \gamma < 1$, and Assumption 3.4 holds. Then $L$ is a contraction mapping on $\mathbb{V}$.*

**Proposition 3.6.** *(Complete Space) $\mathbb{V}$ is a complete normed linear space.*

In Theorem 3.7, we show the fundamental theoretical analysis of an MG-SPA. In (1), we show that an optimal value function of an MG-SPA satisfies the Bellman Equations by applying the Squeeze theorem [Theorem 3.3.6, Sohrab (2003)]. Theorem 3.7-(2) shows that the unique solution of the Bellman Equation exists, a consequence of the fixed-point theorem (Smart, 1980). Therefore, the optimal value function of an MG-SPA exists under Assumption 3.4. By introducing (3), we characterize the relationship between the optimal value function and a Robust Equilibrium. However, (3) does not imply the existence of an RE. To this end, in (4), we formally establish the existence of RE when the optimal value function exists. We formulate a $2N$-player Extensive-form game (EFG) (Osborne & Rubinstein, 1994; Von Neumann & Morgenstern, 2007) based on the optimal value function such that its Nash Equilibrium (NE) is equivalent to an RE of the MG-SPA. The full proof of Theorem 3.7 is in Appendix A.3.

**Theorem 3.7.** *Suppose $0 \leq \gamma < 1$ and Assumption 3.4 holds.*

*(1) (Solution of Bellman Equation) A value function $v_* \in \mathbb{V}$ is an optimal value function if for all $i \in \mathcal{N}$, the point-wise value function $v_*^i \in V$ satisfies the corresponding Bellman Equation (2), i.e. $v_*^i = L^i v_*^i$ for all $i \in \mathcal{N}$.*

*(2) (Existence and uniqueness of optimal value function) There exists a unique $v_* \in \mathbb{V}$ satisfying $Lv_* = v_*$, i.e. for all $i \in \mathcal{N}$, $L^i v_*^i = v_*^i$.*

*(3) (Robust Equilibrium (RE) and optimal value function) A joint policy $d_* = (\pi_*, \rho_*)$, where $\pi_* = (\pi_*^1, \cdots, \pi_*^N)$ and $\rho_* = (\rho_*^{\tilde{1}}, \cdots, \rho_*^{\tilde{N}})$, is a robust equilibrium if and only if $v^{d_*}$ is the optimal value function.*

*(4) (Existence of Robust Equilibrium) There exists a mixed RE for an MG-SPA.*

Though the existence of NE in a stochastic game with perfect information has been investigated (Nash, 1951; Wald, 1945), it is still an open and challenging problem when players have no global state or partially observable information (Hansen et al., 2004; Yang & Wang, 2020). There is a bunch of literature developing algorithms trying to find the NE in Dec-POMDP or partially observable

stochastic game (POSG), and conducting algorithm analysis assuming that NE exists (Chades et al., 2002; Hansen et al., 2004; Nair et al., 2002) without proving the conditions for existence of NE. Once established the existence of RE, we design algorithms to find it. We first develop a robust multi-agent Q-learning (RMAQ) algorithm with convergence guarantee. We then propose a robust multi-agent actor-critic (RMAAC) algorithm to handle the case with high-dimensional state-action spaces.

### 3.3 ROBUST MULTI-AGENT Q-LEARNING (RMAQ) ALGORITHM

By solving the Bellman Equation, we are able to get the optimal value function of an MG-SPA as shown in Theorem 3.7. We therefore develop a value iteration (VI)-based method called robust multi-agent Q-learning (RMAQ) algorithm. Recall the Bellman equation using action-value function in (1), the optimal action-value $q_*$ satisfies $q_*^i(s, a, b) := r^i(s, a, b) + \gamma \mathbb{E}\left[\sum_{s' \in S} p(s'|s, a, b)q_*^i(s', a', b')|a' \sim \pi_*(\cdot|\tilde{s}'), b' \sim \rho_*(\cdot|s')\right]$. As a consequence, the tabular-setting RMAQ update can be written as below,

$$q_{t+1}^i(s_t, a_t, b_t) = (1 - \alpha_t)q_t^i(s_t, a_t, b_t)+ \tag{3}$$

$$\alpha_t \left[ r_t^i(s_t, a_t, b_t) + \gamma \sum_{a_{t+1} \in A} \sum_{b_{t+1} \in B} \pi_{*,t}^{q_t}(a_{t+1}|\tilde{s}_{t+1})\rho_{*,t}^{q_t}(b_{t+1}|s_{t+1})q_t^i(s_{t+1}, a_{t+1}, b_{t+1}) \right],$$

where $(\pi_{*,t}^{q_t}, \rho_{*,t}^{q_t})$ is an NE policy by solving the $2N$-player Extensive-form game (EFG) based on a payoff function $(q_t^1, \cdots, q_t^N, -q_t^1, \cdots, -q_t^N)$. The joint policy $(\pi_{*,t}^{q_t}, \rho_{*,t}^{q_t})$ is used in updating $q_t$. All related definitions of the EFG $(q_t^1, \cdots, q_t^N, -q_t^1, \cdots, -q_t^N)$ are introduced in Appendix A.1. How to solve an EFG is out of the scope of this work, algorithms to do this exist in the literature (Čermák et al., 2017; Kroer et al., 2020). Note that, in RMAQ, each agent's policy is related to not only its own value function, but also other agents' value function. This *multi-dependency* structure considers the interactions between agents in a game, which is different from the the Q-learning in single-agent RL that considers optimizing its own value function. Meanwhile, establishing the convergence of a multi-agent Q-learning algorithm is also a general challenge. Therefore, we try to establish the convergence of (3) in Theorem 3.9, motivated from Hu & Wellman (2003). Due to space limitation, in Appendix B.2, we prove that RMAQ is guaranteed to get the optimal value function $q_* = (q_*^1, \cdots, q_*^N)$ by updating $q_t = (q_t^1, \cdots, q_t^N)$ recursively using (3) under Assumptions 3.8.

**Assumption 3.8.** *(1) State and action pairs have been visited infinitely often. (2) The learning rate $\alpha_t$ satisfies the following conditions: $0 \leq \alpha_t < 1$, $\sum_{t \geq 0} \alpha_t^2 \leq \infty$; if $(s, a, b) \neq (s_t, a_t, b_t)$, $\alpha_t(s, a, b) = 0$. (3) An NE of the $2N$-player EFG based on $(q_t^1, \cdots, q_t^N, -q_t^1, \cdots, -q_t^N)$ exists at each iteration $t$.*

**Theorem 3.9.** *Under Assumption 3.8, the sequence $\{q_t\}$ obtained from (3) converges to $\{q_*\}$ with probability $1$, which are the optimal action-value functions that satisfy Bellman equations (1) for all $i = 1, \cdots, N$.*

Assumption 3.8-(1) is a typical ergodicity assumption used in the convergence analysis of Q-learning (Littman & Szepesvári, 1996; Hu & Wellman, 2003; Szepesvári & Littman, 1999; Qu & Wierman, 2020; Sutton & Barto, 1998). And for Q-learning algorithm design papers that the exploration property is not the main focus, this assumption is also a common assumption (Fujimoto et al., 2019). For exploration strategies in RL (McFarlane, 2018), researchers use $\epsilon$-greedy exploration (Gomes & Kowalczyk, 2009), UCB (Jin et al., 2018; Azar et al., 2017), Thompson sampling (Russo et al., 2018), Boltzmann exploration (Cesa-Bianchi et al., 2017), etc. And for assumption 3.8-(3), researchers have found that the convergence is not necessarily so sensitive to the existence of NE for the stage games during training (Hu & Wellman, 2003; Yang et al., 2018). In particular, under Assumption 3.4, an NE of the $2N$-player EFG exists, which has been proved in Lemma A.6 in Appendix A.1. We also provide an example in the experiment part (the two-player game) where assumptions are indeed satisfied, and our RMAQ algorithm successfully converges to the RE of the corresponding MG-SPA,

### 3.4 ROBUST MULTI-AGENT ACTOR-CRITIC (RMAAC) ALGORITHM

According to the above descriptions of a tabular RMAQ algorithm, each learning agent has to maintain $N$ action-value functions. The total space requirement is $N|S||A|^N|B|^N$ if $|A^1| = \cdots = |A^N|, |B^1| = \cdots = |B^N|$. This space complexity is linear in the number of joint states, polynomial in the number of agents' joint actions and adversaries' joint actions, and exponential in the number of agents. The computational complexity is mainly related to algorithms to solve an Extensive-form

game (Čermák et al., 2017; Kroer et al., 2020). However, even for general-sum normal-form games, computing an NE is known to be PPAD-complete, which is still considered difficult in game theory literature (Daskalakis et al., 2009; Chen et al., 2009; Etessami & Yannakakis, 2010). These properties of the RMAQ algorithm motivate us to develop an actor-critic method to handle high-dimensional space-action spaces. Because actor-critic methods can incorporate function approximation into the update (Konda & Tsitsiklis, 1999).

We consider each agent $i$'s policy $\pi^i$ is parameterized as $\pi_{\theta^i}$ for $i \in \mathcal{N}$, and the adversary's policy $\rho^{\tilde{i}}$ is parameterized as $\rho_{\omega^i}$. We denote $\theta = (\theta^1, \cdots, \theta^N)$ as the concatenation of all agents' policy parameters, $\omega$ has the similar definition. For simplicity, we omit the subscript $\theta_i, \omega_i$, since the parameters can be identified by the names of policies. Note that we here parameterize all policies $\pi^i, \rho^{\tilde{i}}$ as deterministic policies. Then the value function $v^i(s)$ under policy $(\pi, \rho)$ satisfies

$$v^{\pi, \rho, i}(s) = \mathbb{E}_{a \sim \pi, b \sim \rho} \left[ \sum_{s' \in S} p(s'|s, a, b)[r^i(s, a, b) + \gamma v^{\pi, \rho, i}(s')] \right] \tag{4}$$

We establish the general policy gradient with respect to the parameter $\theta, \omega$ in the following theorem. Then we propose our robust multi-agent actor-critic algorithm (RMAAC) which adopts a centralized-training decentralized-execution algorithm structure in MARL literature (Li et al., 2019; Lowe et al., 2017; Foerster et al., 2018). We put the pseudo-code of RMAAC in Appendix B.2.2.

**Theorem 3.10.** *(Policy Gradient in RMAAC for MG-SPA). For each agent $i \in \mathcal{N}$ and adversary $\tilde{i} \in \mathcal{M}$, the policy gradients of the objective $J^i(\theta, \omega)$ with respect to the parameter $\theta, \omega$ are:*

$$\nabla_{\theta^i} J^i(\theta, \omega) = \frac{1}{T} \sum_{t=1}^{T} \nabla_{a^i} q^i(s_t, a_t, b_t) \nabla_{\theta^i} \pi^i(\tilde{s}_t^i)|_{a_t^i = \pi^i(\tilde{s}_t^i), b_t^{\tilde{i}} = \rho^{\tilde{i}}(s_t^i)}$$

$$\nabla_{\omega^i} J^i(\theta, \omega) = \frac{1}{T} \sum_{t=1}^{T} \left[ \nabla_{b^{\tilde{i}}} q^i(s_t, a_t, b_t) + reg \right] \nabla_{\omega^i} \rho^{\tilde{i}}(s_t^i)|_{a_t^i = \pi^i(\tilde{s}_t^i), b_t^{\tilde{i}} = \rho^{\tilde{i}}(s_t^i)} \tag{5}$$

*where $reg = \nabla_{b^{\tilde{i}}} f(s_t^i, b_t^{\tilde{i}}) \nabla_{a^i} q^i(s_t, a_t, b_t) \nabla_f \pi^i(f)$.*

*Proof.* Taking gradient with respect to $\theta^i, \omega^i$ for all $i$ on both sides of (4) yields the results. See details in Appendix B.2.1. ⊓⊔

## 4 EXPERIMENT

### 4.1 ROBUST MULTI-AGENT Q-LEARNING (RMAQ)

We show the performance of the proposed RMAQ algorithm by applying it to a two-player game. We first introduce the designed two-player game. Then we investigate the convergence of this algorithm and compare the performance of the Robust Equilibrium policies with other agents' policies under different adversaries' policies.

Figure 3: Two-player game: each player has two states and the same action set with size 2. Under state $s_0$, two players get the same reward 1 when they choose the same action. At state $s_1$, two players get same reward 1 when they choose different actions. One state switches to another state only when two players get reward, i.e. two players always stay in the current state until they get reward.

**Two-player game:** For the game in Figure 3, two players have the same action space $A = \{0, 1\}$ and state space $S = \{s_0, s_1\}$. The two players get the same positive rewards when they choose the same action under state $s_0$ or choose different actions under state $s_1$. The state does not change until these two players get a positive reward. Possible Nash Equilibrium (NE) in this game can be $\pi_1^* = (\pi_1^1, \pi_1^2)$ that player 1 always chooses action 1, player 2 chooses action 1 under state $s_0$ and action 0 under state $s_1$; or $\pi_2^* = (\pi_2^1, \pi_2^2)$ that player 1 always chooses action 0, player 2 chooses action 0 under state $s_0$ and action 1 under state $s_1$. When using the NE policy, these two players always get the same positive rewards. The optimal discounted state value of this game is $v_*^i(s) = 1/(1 - \gamma)$ for all $s \in S, i \in \{1, 2\}$, $\gamma$ is the reward discounted rate. We set $\gamma = 0.99$, then $v_*^i(s) = 100$.

According to the definition of MG-SPA, we add two adversaries, one for each player to perturb the state and get a negative reward of the player. They have a same action space $B = \{0, 1\}$, where 0 means do not disturb, 1 means change the observation to another one. Some times no perturbation

would be a good choice for adversaries. For example, when the true state is $s_0$, players are using $\pi_1^*$, if adversary 1 does not perturb player 1's observation, player 1 will still select action 1. While adversary 2 changes player 2's observation to state $s_1$, player 2 will choose action 0 which is not same to player 1's action 1. Thus, players always fail the game and get no rewards. A Robust Equilibrium for MG-SPA would be $\tilde{d}_* = (\tilde{\pi}_*^1, \tilde{\pi}_*^2, \tilde{\rho}_*^1, \tilde{\rho}_*^2)$ that each player chooses actions with equal probability and so do adversaries. The optimal discounted state value of corresponding MG-SPA is $\tilde{v}_*^i(s) = 1/2(1-\gamma)$ for all $s \in S, i \in \{1, 2\}$ when players use Robust Equilibrium (RE) policies. We use $\gamma = 0.99$, then $\tilde{v}_*^i(s) = 50$. More explanations of this two-player game refer to Appendix C.1.

**The learning process for RE:** We initialize $q^1(s, a, b) = q^2(s, a, b) = 0$ for all $s, a, b$. After observing the current state, adversaries choose their actions to perturb the agents' state. Then players execute their actions based on the perturbed state information. They then observe the next state and rewards. Then every agent updates its $q$ according to (3). In the next state, all agents repeat the process above. The training stops after 7500 steps. When updating the Q-values, the agent applies a NE policy from the Extensive-form game based on $(q^1, q^2, -q^1, -q^2)$.

**Experiment results:** After 7000 steps of training, we find that agents' Q-values stabilize at certain values, though the dimension of $q$ is a bit high as $q \in \mathbb{R}^{32}$. We compare the optimal state value $\tilde{v}_*$ and the total discounted rewards in Table 1. The value of the total discounted reward converges to the optimal state value of the corresponding MG-SPA. This two-player game experiment result validates the convergence of our RMAQ method. We compare the RE policy with other agents' policies under different adversaries' policies in Appendix C.1. This is to verify the robustness of RE policies.

**Discussion:** Even for general-sum normal-form games, computing an NE is known to be PPAD-complete, which is still considered difficult in game theory literature (Conitzer & Sandholm, 2002; Etessami & Yannakakis, 2010). Therefore, we do not anticipate that the RMAQ algorithm can scale to very large MARL problems. In the next experimental subsection, we show RMAAC with function approximation can handle large-scale MARL problems.

## 4.2 ROBUST MULTI-AGENT ACTOR-CRITIC (RMAAC)

We compare our RMAAC algorithm with MAD-DPG (Lowe et al., 2017), which does not consider robustness, and M3DDPG (Li et al., 2019), where robustness is considered with respective to the opponents' policies altering. We run experiments in several benchmark multi-agent environments, based on the multi-agent particle environments (MPE) (Lowe et al., 2017). The host machine adopted in our experiments is a server configured with AMD Ryzen Threadripper 2990WX 32-core processors and four Quadro RTX 6000 GPUs. Our experiments are performed on Python 3.5.4, Gym 0.10.5, Numpy 1.14.5, Tensorflow 1.8.0, and CUDA 9.0.

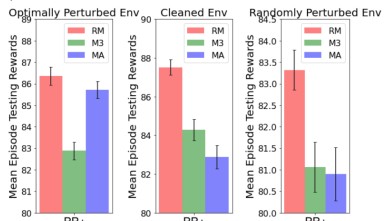

Figure 4: Comparison of episode mean testing rewards using different algorithm in complicated scenarios with a larger number of agents of MPE.

**Experiment procedure:** We first train agents' policies using RMAAC, MADDPG and M3DDPG, respectively. For our RMAAC algorithm, we set the constraint parameter $\epsilon = 0.5$. And we choose two types of perturbation functions to validate the robustness of trained policies under different MG-SPA models. The first one is the linear noise format that $f_1(s^i, b^{\tilde{i}}) := s^i + b^{\tilde{i}}$, i.e. the perturbed state $\tilde{s}^i$ is calculated by adding a random noise $b^{\tilde{i}}$ generated by adversary $\tilde{i}$ to the true state $s^i$. And $f_2(s^i, b^{\tilde{i}}) := s^i + Gaussian(b^{\tilde{i}}, \Sigma)$, where the adversary $\tilde{i}$'s action $b^{\tilde{i}}$ is the mean of the Gaussian distribution. And $\Sigma$ is the covariance, we set it as $I$, i.e. an identity matrix. We call it Gaussian

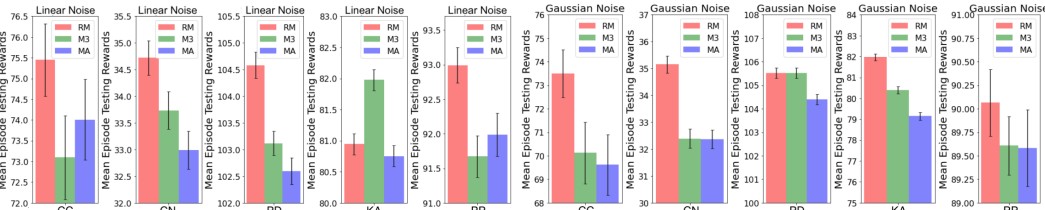

Figure 5: Comparison of episode mean testing rewards using different algorithms and different perturbation functions in MPE.

Table 1: Convergence Values of Total Discounted Rewards when Training Ends

|  | $v^1(s_0)$ | $v^2(s_0)$ | $v^1(s_1)$ | $v^2(s_1)$ | $\tilde{v}^1_*(s_0)$ | $\tilde{v}^2_*(s_0)$ | $\tilde{v}^1_*(s_1)$ | $\tilde{v}^2_*(s_1)$ |
|---|---|---|---|---|---|---|---|---|
| value | 49.99 | 49.99 | 49.99 | 49.99 | 50.00 | 50.00 | 50.00 | 50.00 |

Table 2: Variance of testing rewards

| Perturbation function | Linear noise $f_1$ | | | Gaussian noise $f_2$ | | |
|---|---|---|---|---|---|---|
| Algorithms | RM | M3 | MA | RM | M3 | MA |
| Cooperative communication (CC) | **1.007** | 1.311 | 1.292 | **0.872** | 1.012 | 0.976 |
| Cooperative navigation (CN) | **0.322** | 0.357 | 0.351 | **0.322** | 0.349 | 0.359 |
| Physical deception (PD) | 0.225 | 0.218 | **0.217** | 0.244 | **0.225** | 0.252 |
| Keep away (KA) | **0.161** | 0.168 | 0.175 | **0.167** | 0.17 | 0.167 |
| Predator prey (PP) | 3.213 | **2.812** | 3.671 | **2.304** | 2.711 | 2.811 |

noise format. These two formats $f_1, f_2$ are commonly used in adversarial training (Creswell et al., 2018; Zhang et al., 2020a; 2021). Then we test the well-trained policies in the optimally disturbed environment (injected noise is produced by those adversaries trained with RMAAC algorithm). The testing step is chosen as 10000 and each episode contains 24 steps. All hyperparameters used in experiments for RMAAC, MADDPG and M3DDPG are attached in Appendix C.2.2. Note that since the rewards are defined as negative values in the used multi-agent environments, we add the same baseline (100) to rewards for making them positive. Then it's easier to observe the testing results and make comparisons. Those used MPE scenarios are Cooperative communication (CC), Cooperative navigation (CN), Physical deception (PD), Predator prey (PP) and Keep away (KA). The first two scenarios are cooperative games, the others are mixed games. To investigate the algorithm performance in more complicated situation, we also run experiments in a scenario with more agents, which is called Predator prey+ (PP+). More details of these games are in Append C.2.1.

**Experiment results:** In Figure 5 and Table 2, we report the episode mean testing rewards and variance of 10000 steps testing rewards, respectively. We will use mean rewards and variance for short in the following experimental report and explanations. In the table and figure, we use RM, M3, MA for abbreviations of RMAAC, M3DDPG and MADDPG, respectively. In Figure 5, the left five figures are mean rewards under the linear noise format $f_1$, the right ones are under the Gaussian noise format $f_2$. Under the optimally disturbed environment, agents with RMAAC policies get the highest mean rewards in almost all scenarios no matter what noise format is used. The only exception is when using in Keep away under linear noise. However, our RMAAC still achieves the highest rewards when testing in Keep away under Gaussian noise. In Figure 4, we show the comparison results in a complicated scenario with a larger number of agents. The policies trained with RMAAC get highest reward when testing under optimally perturbed environments. Higher rewards mean agents are performing better. It turns out RMAAC policies are more robust to the worst-case state uncertainty than other two baselines. In Table 2, the left three columns report the variance under the linear noise format $f_1$, and the right ones are under the Gaussian noise format $f_2$. The variance is used to evaluate the stability of the trained policies, i.e. the robustness to system randomness. Because the testing experiments are done in the same environments that are initialized by different random seeds. We can see that, by using our RMAAC method, the agents can get the lowest variance in most of scenarios under these two different perturbation formats. Therefore, our RMAAC algorithm is also more robust to the system randomness, compared with the baselines. Due to the page limits, more experiment results and explanations are in Appendix C.2.

## 5 CONCLUSION

We study the problem of multi-agent reinforcement learning with state uncertainties in this work. We model the problem as a Markov Game with state perturbation adversaries (MG-SPA), where each agent aims to find out a policy to maximize its own total discounted reward and each associated adversary aims to minimize the reward. This problem is challenging with little prior work on theoretical analysis or algorithm design. We provide the first attempt of theoretical analysis and algorithm design for MARL under worst-case state uncertainties. We first introduce Robust Equilibrium as the solution concept for MG-SPA, and prove conditions under which such an equilibrium exists. Then we propose a robust multi-agent Q-learning algorithm (RMAQ) to find such an equilibrium, with convergence guarantees under certain conditions. We also derive the policy gradients and design a robust multi-agent actor-critic (RMAAC) algorithm to handle the more general high-dimensional state-action space MARL problems. We also conduct experiments which validate our methods.

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

# Supplementary Material for "Robust Multi-Agent Reinforcement Learning with State Uncertainty"

## A  THEORY

In this section, we give the full proof of the all propositions and theorems in the theoretical analysis of an MG-SPA.

In section A.1, we construct an extensive-form game (EFG) (Başar & Olsder, 1998; Osborne & Rubinstein, 1994; Von Neumann & Morgenstern, 2007) whose payoff function is related to value functions of an MG-SPA. And, we give certain conditions under which, a Nash Equilibrium for the constructed EFG exists. In section A.2, we prove the propositions 3.5 and 3.6. In section A.3, we give the full proof of Theorem 3.7.

To make the supplemental material self-contained, we re-show the vector notations and assumptions we have presented in section 3.2. Readers can also **skip** the repeated text and go directly to section A.1.

We follow and extend the vector notations in Puterman (2014). Let $V$ denote the set of bounded real valued functions on $S$ with component-wise partial order and norm $\|v^i\| := \sup_{s \in S} |v^i(s)|$. Let $V_M$ denote the subspace of $V$ of Borel measurable functions. For discrete state space, all real-valued functions are measurable so that $V = V_M$. But when $S$ is a continuum, $V_M$ is a proper subset of $V$. Let $v = (v^1, \cdots, v^N) \in \mathbb{V}$ be the set of bounded real valued functions on $S \times \cdots \times S$, i.e. the across product of $N$ state set and norm $\|v\| := \sup_j \|v^j\|$. We also define the set $Q$ and $\mathbb{Q}$ in a similar style such that $q^i \in Q, q \in \mathbb{Q}$.

For discrete $S$, let $|S|$ denote the number of elements in $S$. Let $r^i$ denote a $|S|$-vector, with $s$th component $r^i(s)$ which is the expected reward for agent $i$ under state $s$. And $P$ the $|S| \times |S|$ matrix with $(s, s')$th entry given by $p(s'|s)$. We refer to $r^i_d$ as the reward vector of agent $i$, and $P_d$ as the probability transition matrix corresponding to a joint policy $d = (\pi, \rho)$. $r^i_d + \gamma P_d v^i$ is the expected total one-period discounted reward of agent $i$, obtained using the joint policy $d = (\pi, \rho)$. Let $z$ as a list of joint policy $\{d_1, d_2, \cdots\}$ and $P^0_z = I$, we denote the expected total discounted reward of agent $i$ using $z$ as $v^i_z = \sum_{t=1}^{\infty} \gamma^{t-1} P^{t-1}_z r^i_{d_t} = r^i_{d_1} + \gamma P_{d_1} r^i_{d_2} + \cdots + \gamma^{n-1} P_{d_1} \cdots P_{d_{n-1}} r^i_{t_n} + \cdots$. Now, we define the following minimax operator which is used in the rest of the paper.

**Definition A.1.** *(Minimax Operator, same as definition 3.3) For $v^i \in V, s \in S$, we define the nonlinear operator $L^i$ on $v^i(s)$ by $L^i v^i(s) := \max_{\pi^i} \min_{\rho^{\tilde{i}}} [r^i_d + \gamma P_d v^i](s)$, where $d := (\pi^{-i}_*, \pi^i, \rho^{-\tilde{i}}_*, \rho^{\tilde{i}})$. We also define the operator $Lv(s) = L(v^1(s), \cdots, v^N(s)) = (L^1 v^1(s), \cdots, L^N v^N(s))$. Then $L^i v^i$ is a $|S|$-vector, with $s$th component $L^i v^i(s)$.*

For discrete $S$ and bounded $r^i$, it follows from Lemma 5.6.1 in Puterman (2014) that $L^i v^i \in V$ for all $v^i \in V$. Therefore $Lv \in \mathbb{V}$ for all $v \in \mathbb{V}$. And in this paper, we consider the following assumptions in Markov games with state perturbation adversaries.

**Assumption A.2.** *(same as assumption 3.4)*

*(1) Bounded rewards; $|r^i(s, a, b)| \leq M^i < M < \infty$ for all $i \in \mathcal{N}$, $a \in A$, $b \in B$ and $s \in S$.*

*(2) Finite state and action spaces; all $S^i, A^i, B^{\tilde{i}}$ are finite.*

*(3) Stationary transition probability and reward functions.*

*(4) $f$ is a bijection when $s^i$ is fixed.*

*(5) All agents share one common reward function.*

### A.1  EXTENSIVE-FORM GAME

An extensive-form game (EFG) (Başar & Olsder, 1998; Osborne & Rubinstein, 1994; Von Neumann & Morgenstern, 2007) basically involves a tree structure with several nodes and branches, providing an explicit description of the order of players and the information available to each player at the time of his decision.

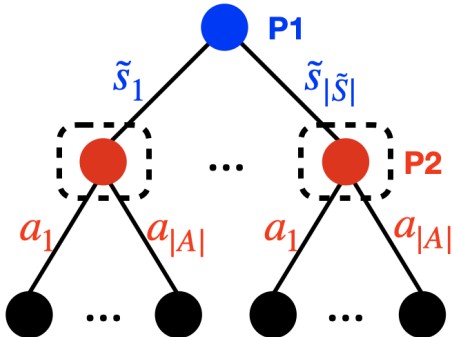

Figure 6: a team extensive-form game

Look at Figure 6, an EFG involves from the top of the tree to the tip of one of its branches. And a centralized nature player ($P1$) has $|\tilde{S}|$ alternatives (branches) to choose from, whereas a centralized agent ($P2$) has $|A|$ alternatives, and the order of play is that the centralized nature player acts before the centralized agent does. The set $A$ is same as the agents' joint action set in an MG-SPA, set $\tilde{S}$ is a set of perturbed state constrained by a constrained parameter $\epsilon$. At the end of lower branches, some numbers will be given. These numbers represent the playoffs to the centralized agent (or equivalently, losses incurred to the centralized nature player) if the corresponding paths are selected by the players. We give the formal definition of an EFG we will use in the proof and the main text as follows:

**Definition A.3.** *An extensive-form game based on $(v^1, \cdots, v^N, -v^1, \cdots, -v^N)$ under $s \in S$ is a finite tree structure with:*

1. *A player $P1$ has a action set $\tilde{S} = \mathcal{B}(\epsilon, s) = \mathcal{B}(\epsilon, s^1) \times \cdots \times \mathcal{B}(\epsilon, s^N)$, with a typical element designed as $\tilde{s}$. And $P1$ moves first,*

2. *Another player $P2$ has a action set $A$, with a typical element designed as $a$. And $P2$ which moves after $P1$,*

3. *a specific vertex indicating the starting point of the game,*

4. *a payoff function $g_s(\tilde{s}, a) = (g_s^1(\tilde{s}, a), \cdots, g_s^N(\tilde{s}, a))$ where $g_s^i(\tilde{s}, a) = r^i(s, a, f_s^{-1}(\tilde{s})) + \sum_{s'} p(s'|a, f_s^{-1}(\tilde{s}))v^i(s')$ assigns a real number to each terminal vector of the tree. Player $P1$ gets $-g_s(\tilde{s}, a)$ while player $P2$ gets $g_s(\tilde{s}, a)$,*

5. *a partition of the nodes of the tree into two player sets (to be denoted by $\bar{N}^1$ and $\bar{N}^2$ for $P1$ and $P2$, respectively),*

6. *a sub-partition of each player set $\bar{N}^i$ into information set $\{\eta_j^i\}$, such that the same number of immediate branches emanates from every node belonging to the same information set, and no node follows another node in the same information set.*

Note that $f_s(b) := f(s, b) = (f(s^1, b^{\tilde{1}}), \cdots, f(s^N, b^{\tilde{N}}))$ is the vector version of the perturbation function $f$ in an MG-SPA. Since in an MG-SPA, $q^i(s, a, b) = r^i(s, a, b) + \sum_{s'} p(s'|s, a, b)v^i(s')$ for all $i = 1, \cdots, N$, $g_s^i(\tilde{s}, a) = q^i(s, a, f_s^{-1}(\tilde{s}))$ as well. We can also use $(q^1, \cdots, q^N, -q^1, \cdots, -q^N)$ to denote an extensive-form game based on $(v^1, \cdots, v^N, -v^1, \cdots, -v^N)$. Then we define the behavioral strategies for $P1$ and $P2$, respectively in the following definition.

**Definition A.4.** *(Behavioral strategy) Let $I^i$ denote the class of all information sets of $Pi$, with a typical element designed as $\eta^i$. Let $U_{\eta^i}^i$ denote the set of alternatives of $Pi$ at the nodes belonging to the information set $\eta^i$. Define $U^i = \cup U_{\eta^i}^i$ where the union is over $\eta^i \in I^i$. Let $Y_{\eta^1}$ denote the set of all probability distributions on $U_{\eta^1}^1$, where the latter is the set of all alternatives of $P1$ at the nodes belonging to the information set $\eta^1$. Analogously, let $Z_{\eta^2}$ denote the set of all probability distributions on $U_{\eta^2}^2$. Further define $Y = \cup_{I^1} Y_{\eta^1}, Z = \cup_{I^2} Z_{\eta^2}$. Then, a behavioral strategy $\lambda$ for $P1$ is a mapping from the class of all his information sets $I^1$ into $Y$, assigning one element in $Y$ for*

each set in $I^1$, such that $\lambda(\eta^1) \in Y_{\eta^1}$ for each $\eta^1 \in I^1$. A typical behavioral strategy $\chi$ for P2 is defined, analogously, as a restricted mapping from $I^2$ into $Z$. The set of all behavioral strategies for $Pi$ is called his behavioral strategy set, and it is denoted by $\Gamma^i$.

The information available to the centralized agent ($P2$) at the time of his play is indicated on the tree diagram in Figure 6 by dotted lines enclosing an area (i.e. the information set) including the relevant nodes. This means the centralized agent is in a position to know exactly how the centralized nature player acts. In this case, a strategy for the centralized agent is a mapping from the collection of his information sets into the set of his actions.

And the behavioral strategy $\lambda$ for $P1$ is a mapping from his information sets and action space into a probability simplex, i.e. $\lambda(\tilde{s}|s)$ is the probability of choosing $\tilde{s}$ given $s$. Similarly, the behavioral strategy $\chi$ for $P2$ is $\chi(a|\tilde{s})$, i.e. the probability of choosing action $a$ when $\tilde{s}$ is given. Note that every behavioral strategy is a mixed strategy. We then give the definition of Nash Equilibrium in behavioral strategies for an EFG.

**Definition A.5.** *(Nash Equilibrium in behavioral strategies) A pair of strategies $\{\lambda_* \in \Gamma^1, \chi_* \in \Gamma^2\}$ is said to constitute a Nash Equilibrium in behavioral strategies if the following inequalities are satisfied that for all $i = 1, \cdots, N, \lambda \in \Gamma^1, \chi \in \Gamma^2, s \in S$:*

$$J^i(\lambda_*^i, \lambda_*^{-i}, \chi^i, \chi_*^{-i}) \geq J^i(\lambda_*^i, \lambda_*^{-i}, \chi_*^i, \chi_*^{-i}) \geq J^i(\lambda^i, \lambda_*^{-i}, \chi_*^i, \chi_*^{-i}) \tag{6}$$

*where $J^i(\lambda, \chi)$ is the expected payoff i.e. $-\mathbb{E}_{\lambda,\chi}[g_s^i]$ when P1 takes $\lambda$, P2 takes $\chi$, $\chi(\tilde{s}|s) = \prod_{i=1}^N \chi^i(\tilde{s}^i|s^i)$, $\pi(a|\tilde{s}) = \prod_{i=1}^N \pi^i(a^i|\tilde{s}^i)$.*

In the following parts as well as the main text, when we mention a Nash Equilibrium for an EFG, it refers to a Nash Equilibrium in behavioral strategies. How to solve an EFG is out of our scope since it has been investigated in many literature (Başar & Olsder, 1998; Schipper, 2017; Slantchev, 2008). And the single policies $\lambda^i$ and $\chi^i$ can be attained through the marginal probabilities calculation with chain rules (Devore et al., 2012; Mémoli, 2012).

**Lemma A.6.** *Suppose $v^1 = \cdots = v^N$, and $S, A$ are finite. An NE $(\lambda_*, \chi_*)$ of the EFG based on $(v^1, \cdots, v^N, -v^1, \cdots, -v^N)$ exists.*

*Proof.* Since $\tilde{S}$ is a subset of $S$, $\tilde{S}$ is finite when $S$ is finite. When $v^1 = \cdots = v^N$, and $\tilde{S}, A$ are finite, an EFG based on $(v^1, \cdots, v^N, -v^1, \cdots, -v^N)$ degenerates to a zero-sum two-person extensive-form game with finite strategies and perfect recall. Thus, an NE of this EFG exists (Başar & Olsder, 1998; Schipper, 2017; Slantchev, 2008). $\square$

**Lemma A.7.** *Suppose $f$ is a bijection when $s^i$ is fixed for all $i = 1, \cdots, N$. For an EFG $(v^1, \cdots, v^N, -v^1, \cdots, -v^N)$ with an NE $(\lambda_*, \chi_*)$, we call a joint policy $(\pi_*^v, \rho_*^v)$ as the joint policy implied from the NE $(\lambda_*, \chi_*)$, where $\rho_*^v(b|s) = \lambda_*(\tilde{s} = f_s(b)|s), \pi_*^v(a|\tilde{s} = f_s(b)) = \chi_*(a|\tilde{s})$. The joint policy $(\pi_*^v, \rho_*^v)$ satisfies $L^i v^i(s) = r_{(\pi_*^v, \rho_*^v)}^i(s) + \gamma \sum_{s' \in S} p_{(\pi_*^v, \rho_*^v)}(s'|s) v^i(s')$ for all $s \in S$.*

*Proof.* The NE of the extensive-form game $(\lambda_*, \chi_*)$ implies that for all $i = 1, \cdots, N, s \in S, \lambda \in \Gamma^1, \chi \in \Gamma^2$, we have

$$J^i(\lambda_*, \chi) \geq J^i(\lambda_*, \chi_*) \geq J^i(\lambda, \chi_*),$$

where $J^i(\lambda, \chi) = -\mathbb{E}[r^i(s, a, f_s^{-1}(\tilde{s})) + \sum_{s'} p(s'|s, a, f_s^{-1}(\tilde{s})) v^i(s')|\tilde{s} \sim \lambda(\cdot|s), a \sim \chi(\cdot|\tilde{s})]$ according to Definition A.5. Let $b$ denote $f_s^{-1}(\tilde{s})$, because $f$ is a bijection when $s^i$ is fixed for all $i = 1, \cdots, N$, $f_s(b) = (f_{s^1}(b^{\tilde{1}}), \cdots, f_{s^N}(b^{\tilde{N}}))$ is a bijection, and the inverse function $f_s^{-1}(\tilde{s}) = (f_{s^1}^{-1}(\tilde{s}^1), \cdots, f_{s^N}^{-1}(\tilde{s}^N))$ exists and is a bijection as well, then we have

$$-J^i(\lambda_*, \chi_*) = \mathbb{E}\left[r^i(s, a, f_s^{-1}(\tilde{s})) + \sum_{s'} p(s'|s, a, f_s^{-1}(\tilde{s})) v^i(s')|\tilde{s} \sim \lambda_*(\cdot|s), a \sim \chi_*(\cdot|\tilde{s})\right]$$

$$= \mathbb{E}\left[r^i(s, a, b) + \sum_{s'} p(s'|s, a, b) v^i(s')|b \sim \lambda_*(f_s(b)|s), a \sim \chi_*(\cdot|f_s(b))\right]$$

$$= \mathbb{E}\left[r^i(s, a, b) + \sum_{s'} p(s'|s, a, b) v^i(s')|b \sim \rho_*^v(\cdot|s), a \sim \pi_*^v(\cdot|\tilde{s})\right]$$

Similarly, we have

$$-J^i(\lambda_*, \chi) = \mathbb{E}\left[r^i(s, a, b) + \sum_{s'} p(s'|s, a, b)v^i(s')|b \sim \rho_*^v(\cdot|s), a \sim \pi^v(\cdot|\tilde{s})\right],$$

$$-J^i(\lambda, \chi_*) = \mathbb{E}\left[r^i(s, a, b) + \sum_{s'} p(s'|s, a, b)v^i(s')|b \sim \rho^v(\cdot|s), a \sim \pi_*^v(\cdot|\tilde{s})\right].$$

Recall the definition of the minimax operator of $L^i v^i(s)$, we have, for all $s \in S$,

$$L^i v^i(s) = r_{(\pi_*^v, \rho_*^v)}^i(s) + \gamma \sum_{s' \in S} p_{(\pi_*^v, \rho_*^v)}(s'|s)v^i(s')$$

$\square$

Based on the proof, we also denote $(\pi_*^v, \rho_*^v)$ as an NE policy for the EFG $(v^1, \cdots, v^N, -v^1, \cdots, -v^N)$ for convenience, instead of calling it the joint policy derived from an NE for the EFG $(v^1, \cdots, v^N, -v^1, \cdots, -v^N)$.

## A.2 PROOF OF TWO PROPOSITIONS

**Proposition A.8.** *(Contraction mapping, same as proposition 3.5 in the main text.) Suppose $0 \leq \gamma < 1$ and Assumption 3.4 hold. Then $L$ is a contraction mapping on $\mathbb{V}$.*

*Proof.* Let $u$ and $v$ be in $\mathbb{V}$. Given Assumption 3.4, these two EFGs $(u^1, \cdots, u^N, -u^1, \cdots, -u^N)$, $(v^i, \cdots, v^N, -v^i, \cdots, -v^N)$ both have at least one mixed Nash Equilibrium according to Lemma A.6. And let $(\pi_*^u, \rho_*^u)$ and $(\pi_*^v, \rho_*^v)$ be two Nash Equilibrium for these two games, respectively. According to Lemma A.7, we have the following equations hold for all $s \in S$,

$$L^i v^i(s) = r_{(\pi_*^v, \rho_*^v)}^i(s) + \gamma \sum_{s' \in S} p_{(\pi_*^v, \rho_*^v)}(s'|s)v^i(s')$$

$$L^i u^i(s) = r_{(\pi_*^u, \rho_*^u)}^i(s) + \gamma \sum_{s' \in S} p_{(\pi_*^u, \rho_*^u)}(s'|s)u^i(s')$$

Then we have

$$r_{(\pi_*^u, \rho_*^v)}^i(s) + \gamma \sum_{s' \in S} p_{(\pi_*^u, \rho_*^v)}(s'|s)v^i(s') \leq L^i v^i(s) \leq r_{(\pi_*^v, \rho_*^u)}^i(s) + \gamma \sum_{s' \in S} p_{(\pi_*^v, \rho_*^u)}(s'|s)v^i(s'),$$

$$r_{(\pi_*^v, \rho_*^u)}^i(s) + \gamma \sum_{s' \in S} p_{(\pi_*^v, \rho_*^u)}(s'|s)u^i(s') \leq L^i u^i(s) \leq r_{(\pi_*^u, \rho_*^v)}^i(s) + \gamma \sum_{s' \in S} p_{(\pi_*^u, \rho_*^v)}(s'|s)u^i(s'),$$

since $(\pi_*^u, \rho_*^v)$ and $(\pi_*^v, \rho_*^u)$ are derived from the Nash Equilibrium of the EFG $(v^i, \cdots, v^N, -v^i, \cdots, -v^N)$, and $(\pi_*^u, \rho_*^v)$ and $(\pi_*^v, \rho_*^u)$ are also derived from the Nash Equilibrium of the EFG $(u^i, \cdots, u^N, -u^i, \cdots, -u^N)$. We assume that $L^i v^i(s) \leq L^i u^i(s)$, then we have

$$0 \leq L^i u^i(s) - L^i v^i(s)$$

$$\leq \left[r_{(\pi_*^u, \rho_*^v)}^i(s) + \gamma \sum_{s' \in S} p_{(\pi_*^u, \rho_*^v)}(s'|s)u^i(s')\right] - \left[r_{(\pi_*^u, \rho_*^v)}^i(s) + \gamma \sum_{s' \in S} p_{(\pi_*^u, \rho_*^v)}(s'|s)v^i(s')\right]$$

$$\leq \gamma \sum_{s' \in S} p_{(\pi_*^u, \rho_*^v)}(s'|s)(u^i(s') - v^i(s'))$$

$$\leq \gamma||v^i - u^i||$$

Repeating this argument in the case that $L^i u^i(s) \leq L^i v^i(s)$ implies that

$$||L^i v^i(s) - L^i u^i(s)|| \leq \gamma||v^i - u^i||$$

for all $s \in S$, i.e. $L^i$ is a contraction mapping on $V$. Recall that $||v|| = \sup_j ||v^j||$, then we have

$$||Lv - Lu|| = \sup_j ||L^j v^j - L^j u^j|| \leq \gamma \sup_j ||v^j - u^j|| = \gamma||v - u||$$

$L$ is a contraction mapping on $\mathbb{V}$.

$\square$

**Proposition A.9.** *(Complete Space, same as proposition 3.6 in the main text.)* $\mathbb{V}$ *is a complete normed linear space.*

*Proof.* Recall that $\mathbb{V}$ denote the set of bounded real valued functions on $S \times \cdots \times S$, i.e. the across product of $N$ state set with component-wise partial order and norm $||v|| := \sup_{s \in S} \sup_j |v^i(s)|$. Since $\mathbb{V}$ is closed under addition and scalar multiplication and is endowed with a norm, it is a normed linear space. Since every Cauchy sequence contains a limit point in $\mathbb{V}$, $\mathbb{V}$ is a complete space. $\qquad\square$

## A.3 Proof of Theorem 3.7

In this section, our goal is to prove Theorem 3.7. We first prove (1) the optimal value function of an MG-SPA satisfies the Bellman Equation by applying the Squeeze theorem [Theorem 3.3.6, Sohrab (2003)] in A.3.1. Then we prove that a unique solution of the Bellman Equation exists using fixed-point theorem (Smart, 1980) in A.3.2. Thereby, the existence of the optimal value function gets proved. By introducing (3), we characterize the relationship between the optimal value function and a Robust Equilibrium. The proof of (3) can be found in A.3.3. However, (3) does not imply the existence of an RE. To this end, in (4), we formally establish the existence of RE when the optimal value function exists. We formulate a $2N$-player Extensive-form game (EFG) (Osborne & Rubinstein, 1994; Von Neumann & Morgenstern, 2007) based on the optimal value function such that its Nash Equilibrium (NE) is equivalent to an RE of the MG-SPA. The details are in A.3.4.

**Theorem A.10.** *(Same as theorem 3.7 in the main text.) Suppose $0 \leq \gamma < 1$ and Assumption 3.4 holds.*

*(1) (Solution of Bellman Equation) A value function $v_* \in \mathbb{V}$ is an optimal value function if for all $i \in \mathcal{N}$, the point-wise value function $v_*^i \in V$ satisfies the corresponding Bellman Equation (2), i.e. $v_*^i = L^i v_*^i$ for all $i \in \mathcal{N}$.*

*(2) (Existence and uniqueness of optimal value function) There exists a unique $v_* \in \mathbb{V}$ satisfying $L v_* = v_*$, i.e. for all $i \in \mathcal{N}$, $L^i v_*^i = v_*^i$.*

*(3) (Robust Equilibrium (RE) and optimal value function) A joint policy $d_* = (\pi_*, \rho_*)$, where $\pi_* = (\pi_*^1, \cdots, \pi_*^N)$ and $\rho_* = (\rho_*^{\tilde{1}}, \cdots, \rho_*^{\tilde{N}})$, is a robust equilibrium if and only if $v^{d_*}$ is the optimal value function.*

*(4) (Existence of Robust Equilibrium) There exists a mixed RE for an MG-SPA.*

### A.3.1 (1) Solution of Bellman Equation

*Proof.* First, we prove that if there exists a $v^i \in V$ such that $v^i \geq L v^i$ then $v^i \geq v_*^i$. $v^i \geq L v^i$ implies $v^i \geq \max \min[r^i + \gamma P v^i] = r_d^i + \gamma P_d v^i$, where $d = (\pi_*^{v,-i}, \pi_*^{v,i}, \rho_*^{v,-i}, \rho_*^{v,i})$ is a Nash Equilibrium for the EFG $v = (v^1, \cdots, v^N, -v^1, \cdots, -v^N)$. We omit the superscript $v$ for convenience when there is no confusion. We choose a list of policy i.e. $z = (d_1, d_2, \cdots)$ where $d_j = (\pi_*^{-i}, \pi_j^i, \rho_*^{-\tilde{i}}, \rho^{\tilde{i}}_*)$. Then we have

$$v^i \geq r_{d_1} + \gamma P_{d_1} v^i \geq r_{d_1}^i + \gamma P_{d_1}(r_{d_2}^i + \gamma P_{d_2} v^i) = r_{d_1}^i + \gamma P_{d_1} r_{d_2}^i + \gamma P_{d_1} P_{d_2} v^i$$

By induction, it follows that, for $n \geq 1$,

$$v^i \geq r_{d_1}^i + \gamma P_{d_1} r_{d_2}^i + \cdots + \gamma^{n-1} P_{d_1} \cdots P_{d_{n-1}} r_{d_n}^i + \gamma^n P_z^n v^i$$

$$v^i - v_z^i \geq \gamma^n P_z^n v^i - \sum_{t=n}^{\infty} \gamma^t P_z^t r_{d_{t+1}}^i \tag{7}$$

Since $||\gamma^n P_z^n v^i|| \leq \gamma^n ||v^i||$ and $\gamma \in [0,1)$, for $\epsilon > 0$, we can find a sufficiently large $n$ such that

$$\epsilon e/2 \geq \gamma^n P_z^n v^i \geq -\epsilon e/2 \tag{8}$$

where $e$ denotes a vector of 1's. And as a result of Assumption 3.4-(1), we have

$$-\sum_{t=n}^{\infty} \gamma^t P_z^t r_{d_{t+1}}^i \geq -\frac{\gamma^n M e}{1 - \gamma} \tag{9}$$

Then we have

$$v^i(s) - v^i_z(s) \geq -\epsilon \tag{10}$$

for all $s \in S$ and $\epsilon > 0$. Let all $d_j$ the same, since $\epsilon$ was arbitrary, we have

$$v^i(s) \geq \max_{\pi^i} \min_{\rho^{\bar{i}}} v^i_z(s) = v^i_*(s) \tag{11}$$

Then we prove that if there exists a $v^i \in V$ such that $v^i \leq Lv^i$ then $v^i \leq v^i_*$. For arbitrary $\epsilon > 0$ there exists a joint policy $d' = (\pi^{-i}_*, \pi^i_*, \rho^{-\bar{i}}_*, \rho^{\bar{i}})$ and a list of policy $z = (d', d', \cdots)$ such that

$$
\begin{aligned}
v^i &\leq r^i_{d'} + \gamma P_{d'} v^i + \epsilon \\
(I - \gamma P_{d'})v^i &\leq r^i_{d'} + \epsilon \\
&\leq (I - \gamma P_{d'})^{-1} r^i_{d'} + (1-\gamma)^{-1}\epsilon e = v^i_z + (1-\gamma)^{-1}\epsilon e \\
&\leq v^i_* + (1-\gamma)^{-1}\epsilon e
\end{aligned}
$$

The equality holds because the Theorem 6.1.1 in Puterman (2014). Since $\epsilon$ was arbitrary, we have

$$v^i \leq v^i_* \tag{12}$$

So if there exists a $v^i \in V$ such that $v^i = L^i v^i$ i.e. $v^i \leq L^i v^i$ and $v^i \geq L^i v^i$, we have $v^i = v^i_*$, i.e. if $v^i$ satisfies the Bellman Equation, $v^i$ is an optimal value function.

$\square$

### A.3.2 (2) EXISTENCE OF OPTIMAL VALUE FUNCTION

*Proof.* Proposition 3.5 and 3.6 establish that $\mathbb{V}$ is a complete normed linear space and $L$ is a contraction mapping, so that the hypothesis of Banach Fixed-Point Theorem are satisfied (Smart, 1980). Therefore there exists a unique solution $v_* \in \mathbb{V}$ to $Lv = v$. From (1), we know if $v_*$ satisfies the Bellman Equation, it is an optimal value function. Therefore, the existence of the optimal value function is proved. $\square$

### A.3.3 (3) ROBUST EQUILIBRIUM AND OPTIMAL VALUE FUNCTION

*Proof.* (i) Robust Equilibrium $\rightarrow$ Optimal value function.

Suppose $d^*$ is a robust equilibrium. Then $v^{d^*} = v^*$. From (2), it follows that $v^{d^*}$ satisfies $Lv = v$. Thus $v^{d^*}$ is the optimal value function.

(ii) Optimal value function $\rightarrow$ Robust Equilibrium.

Suppose $v^{d^*}$ is the optimal value function, i.e., $Lv^{d^*} = v^{d^*}$. The proof of (1) implies that $v^{d^*} = v^*$, so $d^*$ is in robust equilibrium. $\square$

### A.3.4 (4) EXISTENCE OF ROBUST EQUILIBRIUM

*Proof.* From (2), we know that there exists a solution $v_* \in \mathbb{V}$ to Bellman Equation $Lv = v$. Now, we consider an EFG based on $(v^1_*, \cdots, v^N_*, -v^1_*, \cdots, -v^N_*)$. Under Assumption 3.4, we can get an NE policy $(\pi^{v_*}_*, \rho^{v_*}_*)$ by solving the EFG as a consequence of Lemma A.6. According to Lemma A.7, $(\pi^{v_*}_*, \rho^{v_*}_*)$ satisfies

$$L^i v^i_*(s) = r^i_{(\pi^{v_*}_*, \rho^{v_*}_*)}(s) + \gamma \sum_{s' \in S} p_{(\pi^{v_*}_*, \rho^{v_*}_*)}(s'|s) v^i_*(s'),$$

for all $s \in S$. According to (3), $(\pi^{v_*}_*, \rho^{v_*}_*)$ is a Robust Equilibrium. $\square$

# B   ALGORITHM

## B.1   ROBUST MULTI-AGENT Q-LEARNING (RMAQ)

In this section, we prove the convergence of RMAQ under certain conditions. First, let's recall the convergence theorem and certain assumptions.

**Assumption B.1.** *(Same as assumption 3.8) (1) State and action pairs have been visited infinitely often. (2) The learning rate $\alpha_t$ satisfies the following conditions: $0 \le \alpha_t < 1$, $\sum_{t \ge 0} \alpha_t^2 \le \infty$; if $(s, a, b) \ne (s_t, a_t, b_t)$, $\alpha_t(s, a, b) = 0$. (3) An NE of the EFG based on $(q_t^1, \cdots, q_t^N, -q_t^1, \cdots, -q_t^N)$ exists at each iteration $t$.*

**Theorem B.2.** *(Same as theorem 3.9) Under Assumption B.1, the sequence $\{q_t\}$ obtained from (13) converges to $\{q_*\}$ with probability $1$, which are the optimal action-value functions that satisfy Bellman equations (1) for all $i = 1, \cdots, N$.*

$$q_{t+1}^i(s_t, a_t, b_t) = (1 - \alpha_t)q_t^i(s_t, a_t, b_t) + \tag{13}$$

$$\alpha_t \left[ r_t^i(s_t, a_t, b_t) + \gamma \sum_{a_{t+1} \in A} \sum_{b_{t+1} \in B} \pi_{*,t}^{q_t}(a_{t+1}|\tilde{s}_{t+1}) \rho_{*,t}^{q_t}(b_{t+1}|s_{t+1}) q_t^i(s_{t+1}, a_{t+1}, b_{t+1}) \right],$$

*Proof.* Define the operator $Tq_t = T(q_t^1, \cdots, q_t^N) = (T^1 q_t^1, \cdots, T^N q_t^N)$ where the operator $T^i$ is defined as below:

$$T^i q_t^i(s, a, b) = r_t^i + \gamma \sum_{a' \in A} \sum_{b' \in B} \pi_*^{q_t}(a'|\tilde{s}') \rho_*^{q_t}(b'|s') q_t^i(s', a', b') \tag{14}$$

for $i \in \mathcal{N}$, where $(\pi_*^{q_t}, \rho_*^{q_t})$ is the tuple of Nash Equilibrium policies for the EFG based on $(q_t^1, \cdots, q_t^N, -q_t^1, \cdots, -q_t^N)$ obtained from (13). Because of proposition B.3 and proposition B.4 the Lemma 8 in Hu & Wellman (2003) or Corollary 5 in Szepesvári & Littman (1999) tell that $q_{t+1} = (1 - \alpha_t)q_t + \alpha_t Tq_t$ converges to $q_*$ with probability 1. $\qquad\square$

**Proposition B.3.** *(contraction mapping) $Tq_t = (T^1 q_t^1, \cdots, T^N q_t^N)$ is a contraction mapping.*

*Proof.* We omit the subscript $t$ when there is no confusion. Assume $T^i p^i \ge T^i q^i$, we have

$$0 \le T^i p^i - T^i q^i$$

$$= \gamma \left\| \sum_{a' \in A} \sum_{b' \in B} \pi_*^p(a'|\tilde{s}') \rho_*^p(b'|s') p^i(s', a', b') - \sum_{a' \in A} \sum_{b' \in B} \pi_*^q(a'|\tilde{s}') \rho_*^q(b'|s') q^i(s', a', b') \right\|$$

$$\le \gamma \left\| \sum_{a' \in A} \sum_{b' \in B} \pi_*^q(a'|\tilde{s}') \rho_*^q(b'|s') p^i(s', a', b') - \sum_{a' \in A} \sum_{b' \in B} \pi_*^p(a'|\tilde{s}') \rho_*^p(b'|s') q^i(s', a', b') \right\|$$

$$\le \gamma \left\| p^i - q^i \right\|. \tag{15}$$

Repeating the case $T^i p^i \le T^i q^i$ implies that $T^i$ is a contraction mapping such that $||T^i p^i - T^i q^i|| \le \gamma ||p^i - q^i||$ for all $p^i, q^i \in Q$. Recall that $||p - q|| = \sup_j ||p^j - q^j||$

$$||Tp - Tq|| = \sup_j ||T^j p^j - T^j q^j|| \le \gamma \sup_j ||p^j - q^j|| = \gamma ||p - q||$$

$T$ is a contraction mapping such that $||Tp - Tq|| \le \gamma ||p - q||$ for all $p, q \in \mathbb{Q}$. $\qquad\square$

**Proposition B.4.** *(a condition of Lemma 8 in Hu & Wellman (2003) also Corollary 5 in Szepesvári & Littman (1999))*

$$q_* = \mathbb{E}[Tq_*] \tag{16}$$

*Proof.*

$$\mathbb{E}\left[T^i q_*^i(s,a,b)\right] = \mathbb{E}\left[r^i + \gamma \sum_{a' in A} \sum_{b' in B} \pi_*(a'|\tilde{s}')\rho_*(b'|s')q_*^i(s',a',b')\right]$$

$$= r^i + \gamma \sum_{s' \in S} p(s'|s,a,b) \sum_{a' \in A} \sum_{b' \in B} \pi_*(a'|\tilde{s}')\rho_*(b'|s')q_*^i(s',a',b')$$

$$= q_*^i(s,a,b) \tag{17}$$

Therefore $q_* = \mathbb{E}[Tq_*]$. □

### B.2 ROBUST MULTI-AGENT ACTOR-CRITIC (RMAAC)

In this section, we first give the details of policy gradients proof in MG-SPA and then list the Pseudo code of RMAAC.

#### B.2.1 PROOF OF POLICY GRADIENTS

Recall the policy gradient in RMAAC for MG-SPA in the follows:

**Theorem B.5.** *(Policy Gradient in RMAAC for MG-SPA, same as the theorem 3.10). For each agent and adversary $i = 1, \cdots, N$, the policy gradients of the objective $J^i(\theta, \omega)$ with respect to the parameter $\theta, \omega$ are:*

$$\nabla_{\theta^i} J^i(\theta, \omega) = \frac{1}{T} \sum_{t=1}^{T} \nabla_{a^i} q^i(s_t, a_t, b_t) \nabla_{\theta^i} \pi^i(\tilde{s}_t^i)|_{a_t^i = \pi^i(\tilde{s}_t^i), b_t^i = \rho^i(s_t^i)} \tag{18}$$

$$\nabla_{\omega^i} J^i(\theta, \omega) = \frac{1}{T} \sum_{t=1}^{T} \left[\nabla_{b^i} q^i(s_t, a_t, b_t) + reg\right] \nabla_{\omega^i} \rho^i(s_t^i)|_{a_t^i = \pi^i(\tilde{s}_t^i), b_t^i = \rho^i(s_t^i)} \tag{19}$$

*where $reg = \nabla_{b_i} f(s_t^i, b_t^i) \nabla_{a^i} q^i(s_t, a_t, b_t) \nabla_f \pi^i(f)$.*

*Proof.*

$$\nabla_{\theta^i} J^i(\theta, \omega) = \mathbb{E}_{s \sim p_{(\pi,\rho)}} \left[\nabla_{\theta^i} q^i(s,a,b)\right]$$

$$= \mathbb{E}_{s \sim p_{(\pi,\rho)}} \left[\nabla_{a^i} q^i(s,a,b) \nabla_{\theta^i} \pi^i(\tilde{s}^i)\right], \tag{20}$$

$$\nabla_{\omega^i} J^i(\theta, \omega) = \mathbb{E}_{s \sim p_{(\pi,\rho)}} \left[\nabla_{\omega^i} q^i(s,a,b)\right]$$

$$= \mathbb{E}_{s \sim p_{(\pi,\rho)}} \left[\nabla_{a^i} q^i(s,a,b) \nabla_{\tilde{s}^i} \pi^i(\tilde{s}^i) \nabla_{b^i} f(s^i, b^i) \nabla_{\omega^i} \rho^i(s^i) + \nabla_{b^i} q^i(s,a,b) \nabla_{\omega^i} \rho^i(s^i)\right]$$

$$= \mathbb{E}_{s \sim p_{(\pi,\rho)}} \left[\nabla_{\omega^i} \rho^i(s^i) \left[\nabla_{b^i} q^i(s,a,b) + reg\right]\right], \tag{21}$$

where $reg = \nabla_{a^i} q^i(s,a,b) \nabla_{\tilde{s}^i} \pi^i(\tilde{s}^i) \nabla_{b^i} f(s^i, b^i)$. When the actors are updated in a mini-batch fashion (Mnih et al., 2015; Li et al., 2014), (18) and (19) approximate (20) and (21), respectively. □

#### B.2.2 PSEUDO CODE OF RMAAC

We provide the Pseudo code of RMMAC in Algorithm 1.

---

**Algorithm 1:** RMAAC

---

1 Randomly initialize the critic network $q^i(s, a, b|\eta^i)$, the actor network $\pi^i(s^i|\theta^i)$, and the adversary network $\rho^i(s^i|\omega^i)$ for agent $i$. Initialize target networks $q^{i\prime}, \pi^{i\prime}, \rho^{i\prime}$;

2 **for** *each episode* **do**

3 $\quad$ Initialize a random process $\mathcal{N}$ for action exploration;

4 $\quad$ Receive initial state $s$;

5 $\quad$ **for** *each time step* **do**

6 $\quad\quad$ For each adversary $i$, select action $b^i = \rho^i(s^i) + \mathcal{N}$ w.r.t the current policy and exploration. Compute the perturbed state $\tilde{s}^i = f(s^i, b^i)$. Execute actions $a^i = \pi(\tilde{s}^i) + \mathcal{N}$ and observe the reward $r = (r^1, ..., r^n)$ and the new state information $s'$ and store$(s, a, b, \tilde{s}, r, s')$ in replay buffer $\mathcal{D}$. Set $s' \to s$;

7 $\quad\quad$ **for** *agent i=1 to n* **do**

8 $\quad\quad\quad$ Sample a random minibatch of $K$ samples $(s_k, a_k, b_k, r_k, s'_k)$ from $\mathcal{D}$;

9 $\quad\quad\quad$ Set $y_k^i = r_k^i + \gamma q^{i\prime}(s'_k, a'_k, b'_k)|_{a_k^{i\prime} = \pi^{i\prime}(\tilde{s}_k^i), b_k^{i\prime} = \rho^{i\prime}(s_k^i)}$;

10 $\quad\quad\quad$ Update critic by minimizing the loss $\mathcal{L} = \frac{1}{K} \sum_k \left[ y_k^i - q^i(s_k, a_k, b_k) \right]^2$;

11 $\quad\quad\quad$ **for** *each iteration step* **do**

12 $\quad\quad\quad\quad$ Update actor $\pi^i(\cdot|\theta^i)$ and adversary $\rho^i(\cdot|\omega^i)$ using the following gradients

13 $\quad\quad\quad\quad$ $\theta^i \leftarrow \theta^i + \alpha_a \frac{1}{K} \sum_k \nabla_{\theta^i} \pi^i(\tilde{s}_k^i) \nabla_{a^i} q^i(s_k, a_k, b_k)$ where $a_k^i = \pi^i(\tilde{s}_k^i)$, $b_k^i = \rho^i(s_k^i)$;

14 $\quad\quad\quad\quad$ $\omega^i \leftarrow \omega^i - \alpha_b \frac{1}{K} \sum_k \nabla_{\omega^i} \rho^i(s_k^i) \left[ \nabla_{b^i} q^i(s_k, a_k, b_k) + reg \right]$ where $reg = \nabla_{a_k^i} q^i(s_k, a_k, b_k) \nabla_{\tilde{s}_k^i} \pi^i(\tilde{s}_k^i), a_k^i = \pi^i(\tilde{s}_k^i), b_k^i = \rho^i(s_k^i)$;

15 $\quad\quad\quad$ **end**

16 $\quad\quad$ **end**

17 $\quad\quad$ Update all target networks: $\theta^{i\prime} \leftarrow \tau\theta^i + (1-\tau)\theta^{i\prime}, \omega^{i\prime} \leftarrow \tau\omega^i + (1-\tau)\omega^{i\prime}$.

18 $\quad$ **end**

19 **end**

---

## C  EXPERIMENTS

### C.1  ROBUST MULTI-AGENT Q-LEARNING (RMAQ)

In this section, we first further introduce the designed two-player game that the reward function, transition probability function are formally defined. The MG-SPA based on the two-player game is also further explained. Then we show more experimental results about the proposed robust multi-agent Q-learning (RMAQ) algorithm, including the training process of the RMAQ algorithm in terms of the total discounted rewards, the comparison of testing total discounted rewards when using different policies with different adversaries.

#### C.1.1  TWO-PLAYER GAME

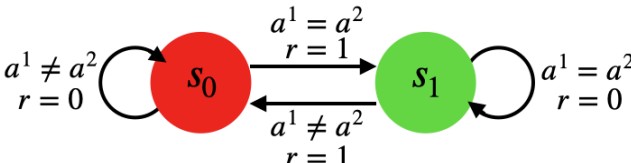

Figure 7: Two-player game: each player has two states and the same action set with size 2. Under state $s_0$, two players get the same reward 1 when they choose the same action. At state $s_1$, two players get same reward 1 when they choose different actions. One state switches to another state only when two players get reward, i.e. two players always stay in the current state until they get reward.

Look at Figure 7 (same as Figure 3 in the main context.), this is how the designed two-player game run. The reward function $r$ and transition probability function $p$ are defined in follows.

These two players get same rewards all the time, i.e. they share a reward function $r$.

$$r^i(s, a^1, a^2) = \begin{cases} 1, & a^1 = a^2, \text{and } s = s_0 \\ 1, & a^1 \neq a^2, \text{and } s = s_1 \\ 0, & a^1 \neq a^2, \text{and } s = s_0 \\ 0, & a^1 = a^2, \text{and } s = s_1 \end{cases} \tag{22}$$

The state does not change until these two players get a positive reward. So the transition probability function $p$ is

$$p(s_1|s, a^1, a^2) = \begin{cases} 1, & a^1 = a^2, \text{and } s = s_0 \\ 0, & a^1 \neq a^2, \text{and } s = s_0 \\ 1, & a^1 = a^2, \text{and } s = s_1 \\ 0, & a^1 \neq a^2, \text{and } s = s_1 \end{cases} \quad p(s_0|s, a^1, a^2) = \begin{cases} 0, & a^1 = a^2, \text{and } s = s_0 \\ 1, & a^1 \neq a^2, \text{and } s = s_0 \\ 0, & a^1 = a^2, \text{and } s = s_1 \\ 1, & a^1 \neq a^2, \text{and } s = s_1 \end{cases} \tag{23}$$

Possible Nash Equilibrium can be $\pi_1^* = (\pi_1^1, \pi_1^2)$ or $\pi_2^* = (\pi_2^1, \pi_2^2)$ where

$$\pi_1^1(a^1|s) = \begin{cases} 1, & a^1 = 1, \text{and } s = s_0 \\ 0, & a^1 = 0, \text{and } s = s_0 \\ 1, & a^1 = 1, \text{and } s = s_1 \\ 0, & a^1 = 0, \text{and } s = s_1 \end{cases} \quad \pi_1^2(a^2|s) = \begin{cases} 1, & a^2 = 1, \text{and } s = s_0 \\ 0, & a^2 = 0, \text{and } s = s_0 \\ 0, & a^2 = 1, \text{and } s = s_1 \\ 1, & a^2 = 0, \text{and } s = s_1 \end{cases} \tag{24}$$

$$\pi_2^1(a^1|s) = \begin{cases} 0, & a^1 = 1, \text{and } s = s_0 \\ 1, & a^1 = 0, \text{and } s = s_0 \\ 0, & a^1 = 1, \text{and } s = s_1 \\ 1, & a^1 = 0, \text{and } s = s_1 \end{cases} \quad \pi_2^2(a^2|s) = \begin{cases} 0, & a^2 = 0, \text{and } s = s_0 \\ 1, & a^2 = 1, \text{and } s = s_0 \\ 1, & a^2 = 0, \text{and } s = s_1 \\ 0, & a^2 = 1, \text{and } s = s_1 \end{cases} \tag{25}$$

NE $\pi_1^*$ means player 1 always selects action 1, player 2 selects action 1 under state $s_0$ and action 0 under state $s_1$. NE $\pi_2^*$ means player 1 always selects action 0, player 2 selects action 0 under state $s_0$ and action 0 under state $s_1$.

According to the definition of MG-SPA, we add two adversaries for each player to perturb the player's observations. And adversaries get negative rewards of players. We let adversaries share a same action space $B^1 = B^2 = \{0, 1\}$, where 0 means do not disturb, 1 means change the observation to the opposite one. Therefore, the perturbed function $f$ in this MG-SPA is defined as:

$$\begin{cases} f(s_0, b = 0) = s_0 \\ f(s_1, b = 0) = s_1 \\ f(s_0, b = 1) = s_1 \\ f(s_1, b = 1) = s_0 \end{cases} \tag{26}$$

Obviously, $f$ is a bijective function when $s^i$ is given. And the constraint parameter $\epsilon = ||S||$, where $||S|| := \max |s - s'|_{\forall s, s' \in S}$, i.e. no constraints for adversaries' power.

A Robust Equilibrium (RE) of this MG-SPA would be $\tilde{d}^* = (\tilde{\pi}_*^1, \tilde{\pi}_*^2, \tilde{\rho}_*^1, \tilde{\rho}_*^2)$, where

$$\begin{cases} \tilde{\pi}_*^1(a^1|s) = 0.5, & \forall s \in S \\ \tilde{\pi}_*^2(a^2|s) = 0.5, & \forall s \in S \\ \tilde{\rho}_*^1(b^1|s) = 0.5, & \forall s \in S \\ \tilde{\rho}_*^2(b^2|s) = 0.5, & \forall s \in S \end{cases} \tag{27}$$

### C.1.2 TRAINING PROCEDURE

In Figure 8, we show the total discounted rewards in the function of training episodes. We set learning rate as 0.1 and train our RMAQ algorithm for 400 episodes. And each episode contains 25 training steps. We can see the total discounted rewards converges to 50, i.e. the optimal value in the MG-SPA, after about 280 episodes or 7000 steps.

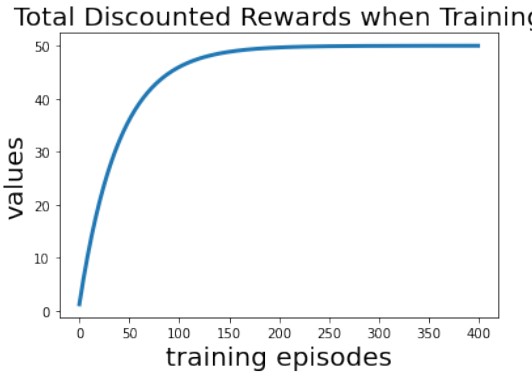

Figure 8: The total discounted rewards converges to the optimal value after about 280 training episodes.

### C.1.3 TESTING COMPARISON

We further test well-trained RE policy when 'strong' adversaries exists. 'Strong' adversary means its probability of modifying agents' observations is larger than the probability of no perturbations in state information. We make two agents play the game using 3 different policies for 1000 steps under different adversaries. And the accumulated rewards, total discounted rewards are calculated. We use the Robust Equilibrium (of the MG-SPA), the Nash Equilibrium (of the original game) and a baseline policy and report the result in Figure 9. The vertical axis is the accumulated/discounted reward, and the horizon axis is the probability that the adversary will attack/perturb the state. And we let these two adversaries share a same policy. We can see as the probability increase, the accumulated and

discounted rewards of RE agents are stable but those rewards of NE agents and baseline agents are keep decreasing. This experiment is to validate the necessity of RE policy which is not only robust to the worst-case or adversarial state uncertainties, but also robust to some worse but note the worst cases.

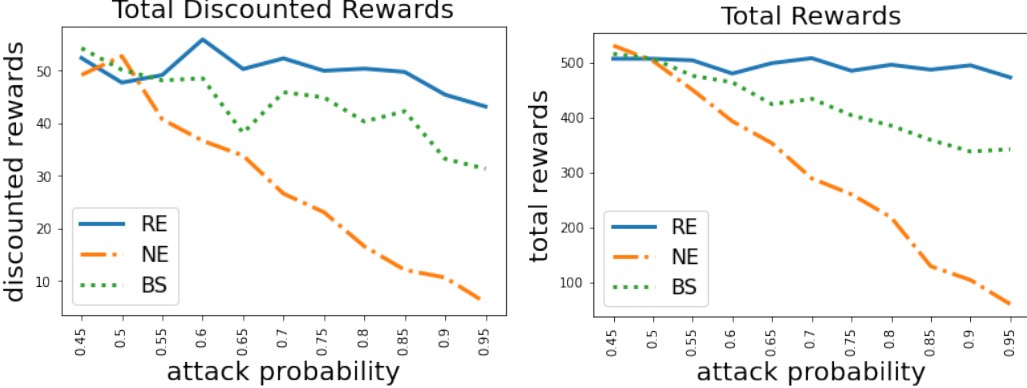

Figure 9: RE policy outperforms other polices in terms of total discounted rewards and total accumulated rewards when strong adversaries exist.

## C.2 ROBUST MULTI-AGENT ACTOR-CRITIC (RMAAC)

In this section, we first briefly introduce the multi-agent environments we use in our experiments. Then we provide more experimental results and explanations, such as the testing results under a cleaned environment (accurate state information can be attained) and a randomly perturbed environment (injecting standard Gaussian noise in agents' observations). In the last subsection, we list all hyperparameters we used in the experiments, as well as the baseline source code.

### C.2.1 MULTI-AGENT ENVIRONMENTS

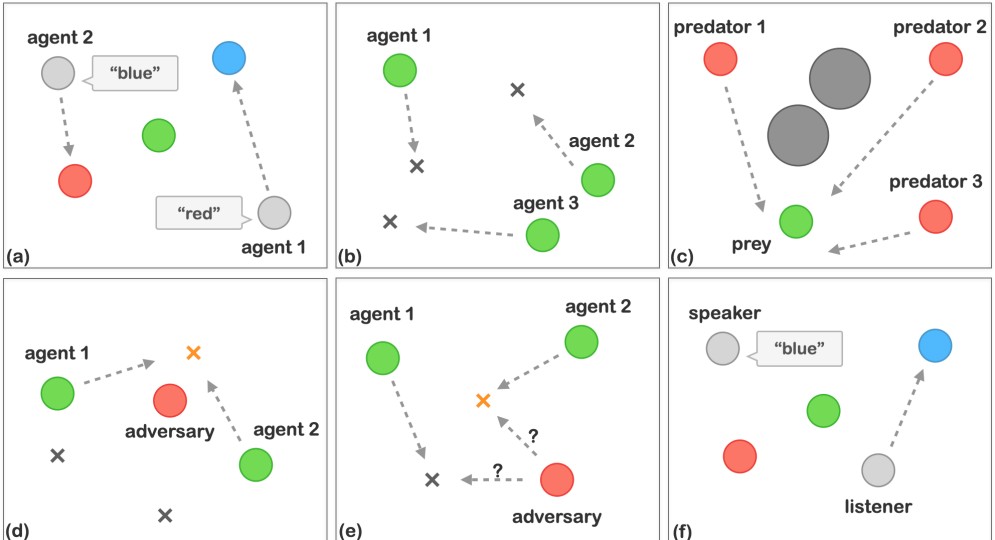

Figure 10: Illustrations of the experimental scenarios and some games we consider, including a) *Cooperative communication* b) *Cooperative navigation* c) *Predator prey* d) *Keep away* e) *Physical deception* f) *Navigate communication*

**Cooperative communication (CC):** This is a cooperative game. There are 2 agents and 3 landmarks of different colors. Each agent wants to get to their target landmark, which is known only by other agent. Reward is collective. So agents have to learn to communicate the goal of the other agent, and navigate to their landmark.

**Cooperative navigation (CN):** This is a cooperative game. There are 3 agents and 3 landmarks. Agents are rewarded based on how far any agent is from each landmark. Agents are penalized if they collide with other agents. So, agents have to learn to cover all the landmarks while avoiding collisions.

**Physical deception (PD):** This is a mixed cooperative and competitive task. There are 2 collaborative agents, 2 landmarks, and 1 adversary. Both the collaborative agents and the adversary want to reach the target, but only collaborative agents know the correct target. The collaborative agents should learn a policy to cover all landmarks so that the adversary does not know which one is the true target.

**Keep away (KA):** This is a competitive task. There is 1 agent, 1 adversary, and 1 landmark. The agent knows the position of the target landmark and wants to reach it. Adversary is rewarded if it is close to the landmark, and if the agent is far from the landmark. Adversary should learn to push agent away from the landmark.

**Predator prey (PP):** This is a mixed game known as predator-prey. Prey agents (green) are faster and want to avoid being hit by adversaries (red). Predator are slower and want to hit good agents. Obstacles (large black circles) block the way.

**Navigate communication (NC):** This is a cooperative game which is similar to Cooperative communication. There are 2 agents and 3 landmarks of different colors. A agent is the 'speaker' that does not move but observes goal of other agent. Another agent is the listener that cannot speak, but must navigate to correct landmark.

**Predator prey+ (PP+):** This is an extension of the Predator prey environment by adding more agents. There are 2 preys, 6 adversaries, and 4 landmarks. Prey agents are faster and want to avoid being hit by adversaries. Predator are slower and want to hit good agents. Obstacles block the way.

### C.2.2   EXPERIMENTS HYPER-PARAMETERS

In Table 3, we show all hyper-parameters we use to train our policies and baselines. We also provide our source code in the supplementary material. The source code of M3DDPG (Li et al., 2019) and MADDPG (Lowe et al., 2017) accept the MIT License which allows any person obtaining them to deal in the code without restriction, including without limitation the rights to use, copy, modify, etc. More information about this license refers to `https://github.com/openai/maddpg` and `https://github.com/dadadidodi/m3ddpg`.

Table 3: Hyper-parameters

| Parameter | RMAAC | M3DDPG | MADDPG |
|---|---|---|---|
| optimizer | Adam | Adam | Adam |
| learning rate | 0.01 | 0.01 | 0.01 |
| adversarial learning rate | 0.005 | / | / |
| discount factor | 0.95 | 0.95 | 0.95 |
| replay buffer size | $10^6$ | $10^6$ | $10^6$ |
| number of hidden layers | 2 | 2 | 2 |
| activation function | Relu | Relu | Relu |
| number of hidden unites per layer | 64 | 64 | 64 |
| number of samples per minibatch | 1024 | 1024 | 1024 |
| target network update coefficient $\tau$ | 0.01 | 0.01 | 0.01 |
| iteration steps | 20 | 20 | 20 |
| constraint parameter $\epsilon$ | 0.5 | / | / |
| episodes in training | 10k | 10k | 10k |
| time steps in one episode | 25 | 25 | 25 |

### C.2.3   MORE TESTING RESULTS

In this subsection, we provide the testing results under a cleaned environment (accurate state information can be attained) and a randomly disturbed environment (injecting standard Gaussian noise into agents' observations).

In Figure 11, we show the comparison of mean episode testing rewards under a cleaned environment by using 4 different methods, RM1 denotes our RMAAC policy trained with the linear noise format $f_1$, RM2 denotes our RMAAC policy trained with the Gaussian noise format $f_2$, MA denotes MADDPG (`https://github.com/openai/maddpg`), M3 denotes M3DDPG (`https://github.com/dadadidodi/m3ddpg`). We can see only in the Predator prey scenario, our method outperforms others under a cleaned environment. In Figure 12, we can see our method outperforms others in the Cooperative communication, Keep away and Predator prey scenarios, and achieves a similar performance as others in the Cooperative navigation scenario under a randomly perturbed environment. In Table 4 and 5, we also report the variances of testing rewards in different scenarios under different environment settings. Our method has lower variance in three of five scenarios.

This kind of performance also happens in robust optimization (Beyer & Sendhoff, 2007; Boyd & Vandenberghe, 2004) and distributionally robust optimization (Delage & Ye, 2010; Rahimian & Mehrotra, 2019) that the robust solution outperforms other non-robust solutions in the worst-case scenario. Similarly, for single-agent RL with state perturbations, robust policies perform better compared with baselines under state perturbations Zhang et al. (2020b). But the robust solutions may get relatively poor performance compared with other non-robust solutions when there is no uncertainty or perturbation in the environment even in a single agent RL problem Zhang et al. (2020b). Improving the robustness of the trained policy may sacrifice the performance of the decisions when the perturbations or uncertainties do not happen. That's why our RMAAC policies only beat all baselines in one scenario when the state uncertainty is eliminated. However, for many real-world systems we

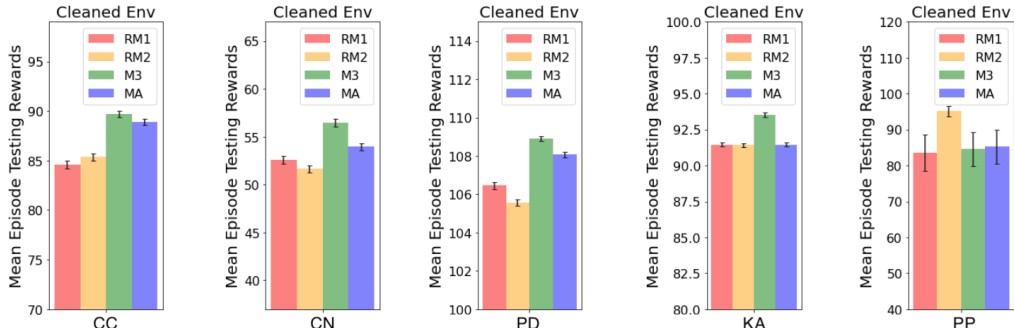

Figure 11: Comparison of episode mean testing rewards using different algorithms and different perturbation functions, under cleaned environments.

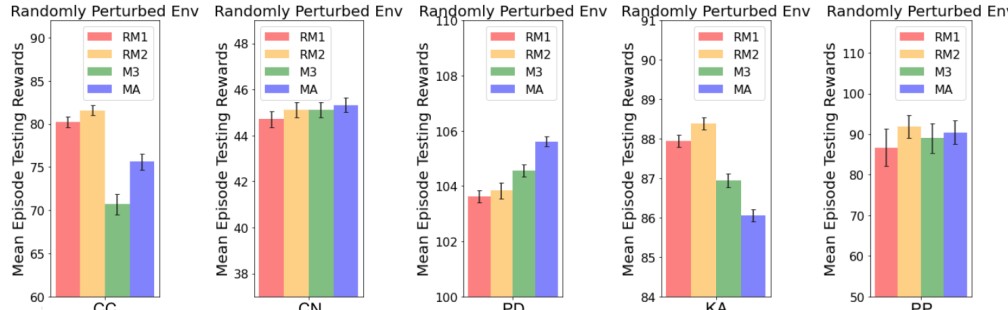

Figure 12: Comparison of episode mean testing rewards using different algorithms and different perturbation functions, under randomly perturbed environments.

can not assume the agents always have accurate information of the states. Hence, improving the robustness of the policies is very important for MARL as we explained in the introduction of this work. It is worth noting that our RMAAC policies also work well in environments with random perturbations instead of worst-case perturbations. As shown in Fig. 12, the performance of our RMAAC policies outperforms the baselines in most scenarios when random noise is introduced into the state.

MAPPO is a multi-agent reinforcement learning algorithm which performs well in cooperative multi-agent settings (Yu et al., 2021a). We use MP to denote MAPPO (`https://github.com/marlbenchmark/on-policy.`). In Figure 13, we compare its performance with our RMAAC algorithm in two cooperative scenarios of MPE. The details of scenarios such as Cooperative navigation, Navigate communication can be found in the last section. We can see that under the optimally perturbed environment, RMAAC outperforms MAPPO in all scenarios.

In Figure 14 and Table 6, we compare the episode mean testing rewards and variances under different environments in the complicated scenario with a large number of agents between different algorithms. We adopt Gaussian noise format in training RMAAC polices. We can see our method has lower variance under two of three environments and has the highest rewards under all environments.

Table 4: Variance of testing rewards under cleaned environment

| Algorithms | RM with $f_1$ | RM with $f_2$ | M3 | MA |
|---|---|---|---|---|
| Cooperative communication (CC) | 0.383 | 0.376 | **0.295** | 0.328 |
| Cooperative navigation (CN) | 0.413 | **0.361** | 0.416 | 0.376 |
| Physical deception (PD) | 0.175 | 0.165 | **0.133** | 0.143 |
| Keep away (KA) | 0.137 | **0.134** | 0.17 | 0.145 |
| Predator prey (PP) | 5.139 | **1.450** | 4.681 | 4.725 |

Table 5: Variance of testing rewards under randomly perturbed environment

| Algorithms | RM with $f_1$ | RM with $f_2$ | M3 | MA |
|---|---|---|---|---|
| Cooperative communication (CC) | 0.592 | **0.547** | 1.187 | 0.937 |
| Cooperative navigation (CN) | 0.336 | 0.33 | 0.328 | **0.321** |
| Physical deception (PD) | 0.222 | 0.292 | 0.209 | **0.184** |
| Keep away (KA) | **0.155** | **0.155** | 0.166 | 0.161 |
| Predator prey (PP) | 4.629 | **2.752** | 3.644 | 2.9 |

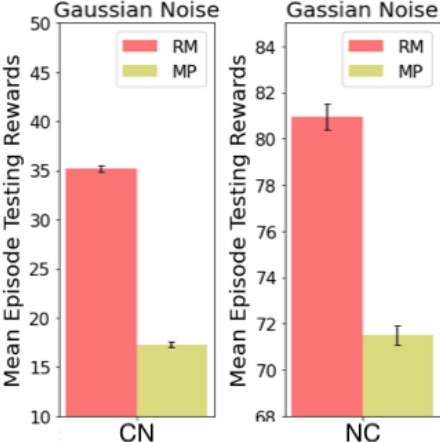

Figure 13: Comparison of episode mean testing rewards using MAPPO and RMAAC under optimally perturbed environments.

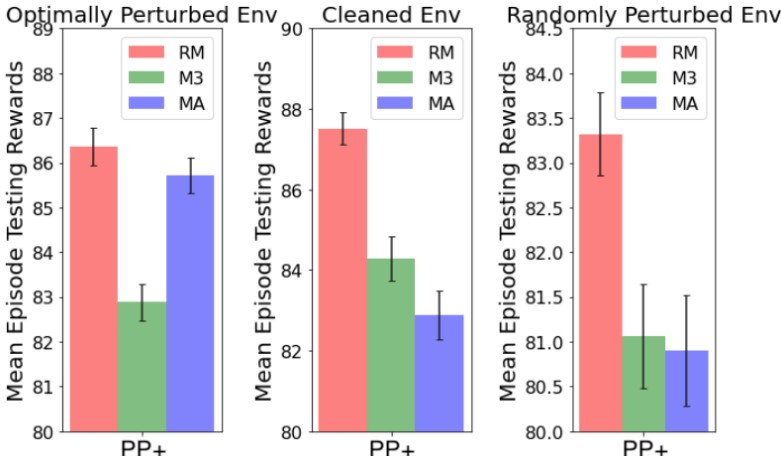

Figure 14: Comparison of episode mean testing rewards using different algorithms under different environments in Predator prey+.

Table 6: Variance of testing rewards under different environments in Predator prey+.

| Algorithm | RM | M3 | MA |
|---|---|---|---|
| Optimally Perturbed Env | 4.199 | 4.046 | **3.924** |
| Randomly Perturbed Env | **4.664** | 5.774 | 6.191 |
| Cleaned Env | **3.928** | 5.521 | 6.006 |

### C.2.4 TRAINING RESULTS USING LINEAR NOISE WITH DIFFERENT CONSTRAINT PARAMETERS

**Training Setup:** In this subsection, we train several RMAAC policies using linear noise format as the state perturbation function, i.e. $f_1(s^i, b^{\tilde{i}}) = s^i + b^{\tilde{i}}$. The constraint parameter $\epsilon$ is respectively set as $0.01, 0.05, 0.1, 0.5, 1$ and $2$, given other hyper-parameters unchanged. Other used hyper-parameters can be found in Table 3.

**Training Results:** In Figure 15, 16, and 17, we show the training process in three scenarios: Cooperative communication (CC), Cooperative navigation (CN), Predator Prey (PP), respectively. The y-axis denotes the mean episode reward of the agents and the x-axis denotes the training episodes.

From these figures we can see that, in general, the smaller the used variance, the higher the mean episode rewards RMAAC can achieve. However, RMAAC has different sensitivities to the value of variance in different scenarios. When we use $\epsilon = 2$, the RMAAC policies have the lowest mean episode rewards in all three scenarios. Nevertheless, when we use the smallest constraint parameter $\epsilon = 0.01$, the trained RMAAC policies do not achieve the highest mean episode rewards in all three scenarios. In these three scenarios, it is clear to see the performance of RMAAC using $\epsilon = 0.5$ is better than or similar to the performance of RMAAC using $\epsilon = 1$, and better than the performance of RMAAC using $\epsilon = 2$, i.e. Performance$(\epsilon = 0.5) \geq$ Performance$(\epsilon = 1) >$ Performance$(\epsilon = 2)$. The performance of the RMAAC policies is close when the constraint parameters are less or equal to than $0.1$.

### C.2.5 TESTING RESULTS USING LINEAR NOISE WITH DIFFERENT CONSTRAINT PARAMETERS

In this subsection, we test well-trained RMAAC policies in perturbed environments where adversaries adopt linear noise format and different constraint parameters.

**Testing Setup:** The tested policy $\pi_{test}$ is trained with the linear noise format $f_1(s^i, b^{\tilde{i}}) = s^i + b^{\tilde{i}}$, constraint parameter is $0.5$, where $b^{\tilde{i}} = \rho_{test}^{\tilde{i}}(s^i|\epsilon = 0.5)$. The policy $\rho_{test}^{\tilde{i}}$ is adversary $\tilde{i}$'s policy which is trained with $\pi_{test}$ in RMAAC, for all $\tilde{i} = \tilde{1}, \cdots, \tilde{N}$. We use $\rho_{test}$ to denote the joint policy of adversaries which is used in the testing. In summary, we test agents' joint policy $\pi_{test}^{scenario}(\tilde{s})$ when adversaries adopt the joint policy $\rho_{test}^{scenario}(s|\epsilon)$, and $\epsilon = 0.01, 0.05, 0.1, 0.5, 1, 2$, $scenario =$ Cooperative communication (CC), Cooperative navigation (CN), Predator Prey (PP), respectively. The testing is conducted over 400 episodes, which each episode has 25 time-steps.

**Testing Results:** In Figures 18, 19 and 20, we compare the performance of RMAAC, M3DDPG, and MADDPG in scenarios CC, CN, and PP with different values of constraint parameters. MADDPG is a MARL baseline algorithm. M3DDPG is a robust MARL baseline algorithm. The y-axis denotes the mean episode reward of the agents.

From these figures, we can see that in all three scenarios, our RMAAC policies outperform the baseline MARL and robust MARL policies in terms of mean episode testing rewards under the attacks of linear noise format with different constraint parameters $\epsilon$. Our proposed RMAAC algorithm is robust to the state information attacks of linear noise format with different constraint parameters.

### C.2.6 TRAINING RESULTS USING GAUSSIAN NOISE WITH DIFFERENT VARIANCE

**Training Setup:** In this subsection, we train several RMAAC policies using Gaussian noise format as the state perturbation function, i.e. $f_2(s^i, b^{\tilde{i}}) = s^i + \mathcal{N}(b^{\tilde{i}}, \sigma)$. The variance $\sigma$ is respectively set as $0.001, 0.05, 0.1, 0.5, 1, 2$ and $3$, given other hyper-parameters unchanged. Other used hyper-parameters can be found in Table 3.

**Training Results:** In Figure 21, 22 and 23, we show the training process of RMAAC in three scenarios: Cooperative communication (CC), Cooperative navigation (CN), Predator Prey (PP). The y-axis denotes the mean episode reward of the agents and the x-axis denotes the training episodes.

From the figures we can see that, in general, the smaller the value of variance is used, the higher the mean episode rewards RMAAC can achieve. However, RMAAC has different sensitivities to the value of variance in different scenarios. When we use $\sigma = 3$, the RMAAC policies have the lowest mean episode rewards in all three scenarios. Nevertheless, when we use the smallest magnitude $0.001$,

the trained RMAAC policies do not always achieve the highest mean episode rewards. In these three scenarios, it is clear to see the performance of RMAAC using $\sigma = 1$ is better than or close to that of using $\sigma = 2$, and better than that of using $\sigma = 3$, i.e. Performance($\sigma = 1$) $\geq$ Performance($\sigma = 2$) $>$ Performance($\sigma = 3$). The performance of the RMAAC policies is close when the constraint parameters are less than or equal to 0.5.

### C.2.7 TESTING RESULTS USING GAUSSIAN NOISE WITH DIFFERENT VARIANCE

In this subsection, we test well-trained RMAAC policies in perturbed environments where adversaries adopt Gaussian noise format and different variances.

**Testing Setup:** The tested policy $\pi_{test}$ is trained with Gaussian noise format $f_2(s^i, b^{\tilde{i}}) = s^i + \mathcal{N}(b^{\tilde{i}}, \sigma = 1)$, constraint parameter is 0.5, where $b^{\tilde{i}} = \rho_{test}^{\tilde{i}}(s^i | \epsilon = 0.5)$. $\rho^{\tilde{i}}$ is adversary $i$'s policy which is trained with $\pi_{test}$ in RMAAC, for all $i = 1, \cdots, N$. We use $\rho_{test}$ to denote the joint policy of adversaries. In summary, we test agents' joint policy $\pi_{test}^{scenario}(\tilde{s})$ when adversaries adopt the joint policy $\rho_{test}^{scenario}(s | \epsilon = 0.5)$ and Gaussian noise format $f_2(s^i, b^{\tilde{i}}) = s^i + \mathcal{N}(b^{\tilde{i}}, \sigma)$, where $\sigma = 0.001, 0.05, 0.1, 0.5, 1, 2, 3$, $scenario =$ Cooperative communication (CC), Cooperative navigation (CN), Predator Prey (PP). The testing is conducted over 400 episodes, which each episode has 25 time-steps.

**Testing Results:** In Figures 24, 25 and 26, we respectively compare the performance of RMAAC, M3DDPG, and MADDPG in scenarios Cooperative communication, Cooperative navigation and Predator Prey with different values of constraint parameters. MADDPG is a MARL baseline algorithm. M3DDPG is a robust MARL baseline algorithm. The y-axis denotes the mean episode reward of the agents.

From these figures, we can see that in all three scenarios with all different values of constraint parameters, our RMAAC policies outperform the MARL and robust MARL baseline policies in terms of mean episode rewards under the attacks of Gaussian noise format with different variance. Our proposed RMAAC algorithm is robust to the state information attacks of Gaussian noise format with different values of variance.

### C.2.8 TESTING RESULTS UNDER DIFFERENT STATE PERTURBATION FUNCTIONS

In this subsection, we test the well-trained RMAAC policies in perturbed environments where adversaries adopt different noise formats and policies.

**Testing Setup:** The tested agents' joint policy $\pi_{test}$ is trained with Gaussian noise format $f_2(s^i, b^{\tilde{i}}) = s^i + Gaussian(b^{\tilde{i}}, \sigma = 1)$, constraint parameter is 0.5, where $b^{\tilde{i}} = \rho_{test}^{\tilde{i}}(s^i | \epsilon = 0.5)$. $\rho_{test}^{\tilde{i}}$ is adversary $\tilde{i}$'s policy which is trained with $\pi_{test}$ in RMAAC, for all $\tilde{i} = \tilde{1}, \cdots, \tilde{N}$. We use $\rho_{test}$ to denote the joint policy of adversaries. In a summary, we test agents' joint policy $\pi_{test}^{scenario}(\tilde{s})$ when adversaries adopt the joint policy $\rho_{test}^{scenario}(s | \epsilon = 0.5)$, in three scenarios $scenario =$ Cooperative communication (CC), Cooperative navigation (CN), Predator Prey (PP), under non-optimal Gaussian format $f_3$, Uniform noise format $f_4$, fixed Gaussian noise format $f_5$ and Laplace noise format $f_6$, respectively. These noise formats are defined in the following:

$$
\begin{aligned}
f_3(s^i, b^{\tilde{i}}) &= s^i + Gaussian(b^{\tilde{i}}, 1) \quad \text{where} \quad b^{\tilde{i}} = \rho_{non-optimal}^{\tilde{i}}(s^i | \epsilon), \\
f_4(s^i, b^{\tilde{i}}) &= s^i + Uniform(-\epsilon, +\epsilon), \\
f_5(s^i, b^{\tilde{i}}) &= s^i + Gaussian(0, 1), \\
f_6(s^i, b^{\tilde{i}}) &= s^i + Laplace(b^{\tilde{i}}, 1) \quad \text{where} \quad b^{\tilde{i}} = \rho_{test}^{\tilde{i}}(s^i | \epsilon),
\end{aligned}
\tag{28}
$$

where $\rho_{non-optimal}^{\tilde{i}}$ is a non-optimal policy of adversary $\tilde{i}$. $\rho_{non-optimal}^{\tilde{i}}$ is randomly chosen from the training process. As we can see that $f_3$ and $f_6$ are independent of the optimal joint policy of adversaries, but $f_4$ and $f_6$ are not. The testing is conducted over 400 episodes, which each episode has 25 time-steps.

**Testing Results:** In Figures 27, 28, and 29, we compare the performance of RMAAC, M3DDPG and MADDPG in scenarios Cooperative communication, Cooperative navigation and Predator Prey under

4 different noise formats, respectively. MADDPG is a MARL baseline algorithm. M3DDPG is a robust MARL baseline algorithm. The y-axis denotes the mean episode reward of the agents.

As we can see from these figures, most of the time, our RMAAC policy outperforms the MARL (MADDPG) and robust MARL (M3DDPG) baseline policies. In Cooperative communication and Predator Prey, under all 4 different noise formats, RMAAC policies achieve the highest mean episode rewards. In Cooperative navigation, RMAAC policies have the highest mean episode rewards when the non-optimal Gaussian noise format and Laplace noise format are used. The only exceptions happen in Cooperative navigation when Uniform noise format and fixed Gaussian noise format are used. However, we can find that the performance of RMAAC policies is close to that of the baseline policies in terms of mean episode testing rewards. In general, our RMAAC algorithm is robust to different types of state information attacks.

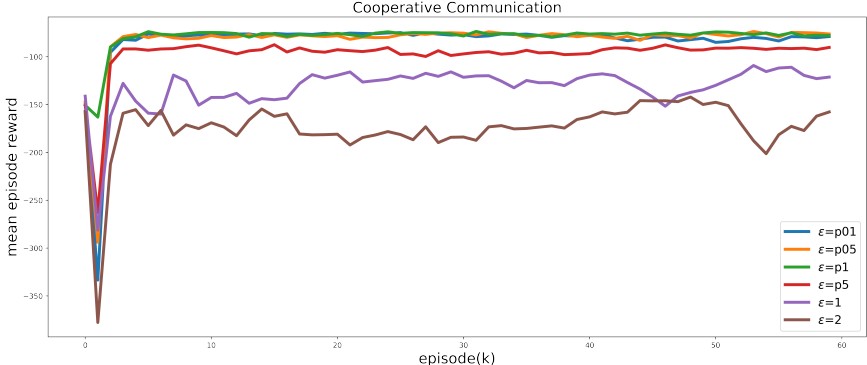

Figure 15: We train RMAAC policies using different values of constraint parameters in the scenario Cooperative Communication. In general, the smaller the constraint parameter is used, the higher the mean episode rewards RMAAC can achieve.

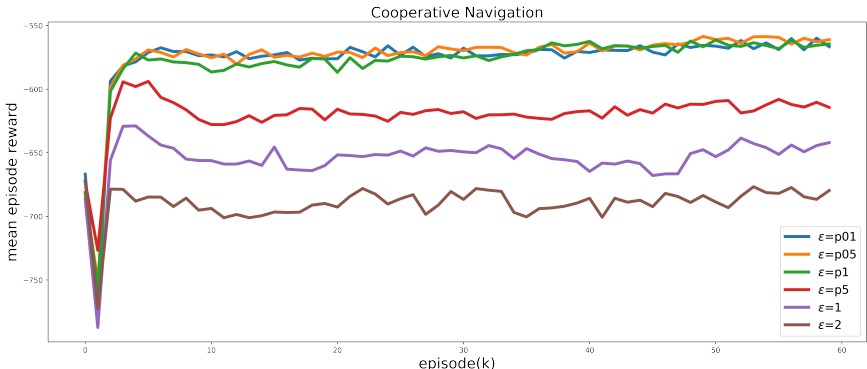

Figure 16: We train RMAAC policies using different values of constraint parameters in the scenario Cooperative Navigation. In general, the smaller the constraint parameter is used, the higher the mean episode rewards RMAAC can achieve.

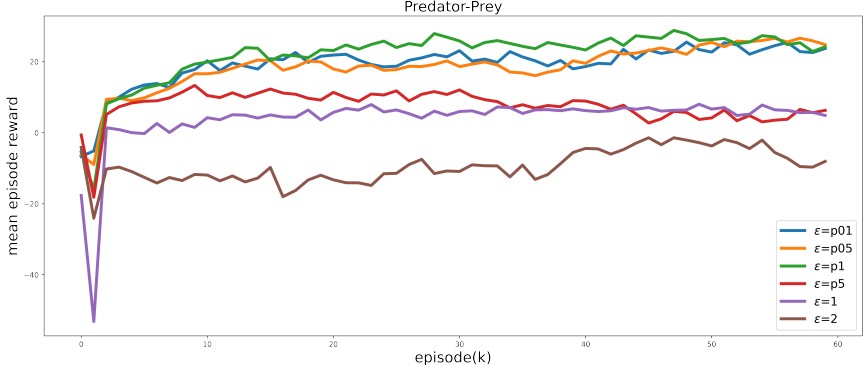

Figure 17: We train RMAAC policies using different values of constraint parameters in the scenario Predator-Prey. In general, the smaller the constraint parameter is used, the higher the mean episode rewards RMAAC can achieve.

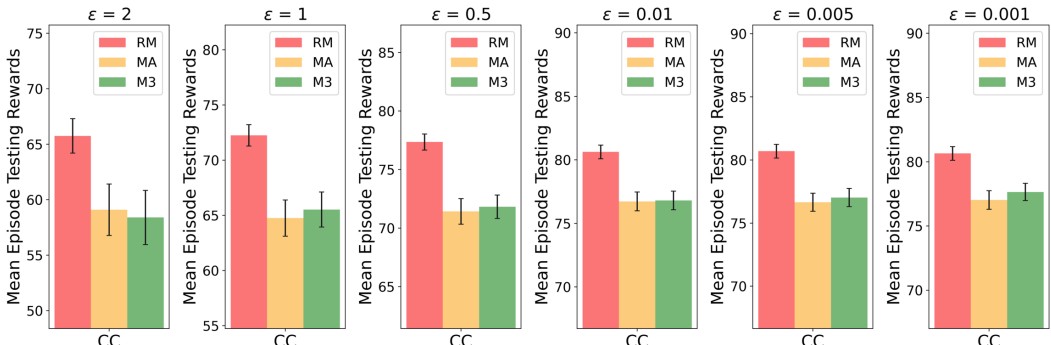

Figure 18: We test the performance of RMAAC(RM), MADDPG(MA), and M3DDPG(M3) policies under the attack of linear noise format when using different values of constraint parameters in the scenario Cooperative Communication. RM denotes our robust MARL algorithm, i.e. RMAAC. MA denotes MADDPG, a MARL baseline algorithm. M3 denotes M3DDPG, a robust MARL baseline algorithm. Our RMAAC algorithm outperforms baseline algorithms in terms of mean episode testing rewards under all situations using different values of constraint parameters.

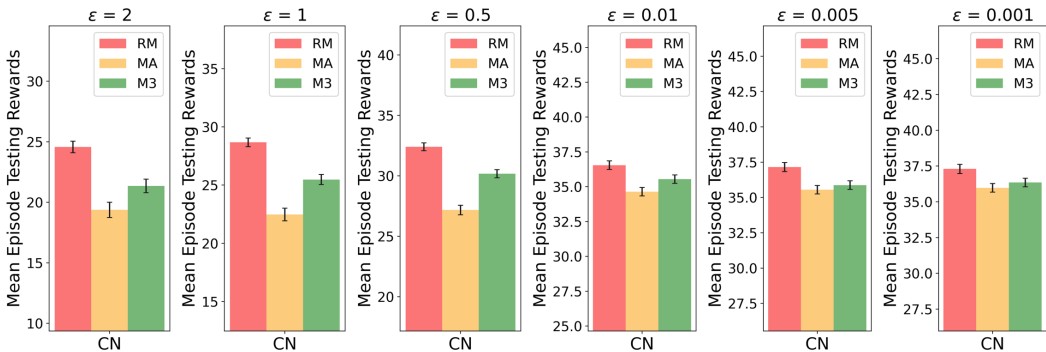

Figure 19: We test the performance of RMAAC(RM), MADDPG(MA), and M3DDPG(M3) policies under the attack of linear noise format when using different values of constraint parameters in the scenario Cooperative Navigation. RM denotes our robust MARL algorithm, i.e. RMAAC. MA denotes MADDPG, a MARL baseline algorithm. M3 denotes M3DDPG, a robust MARL baseline algorithm. Our RMAAC algorithm outperforms baseline algorithms in terms of mean episode testing rewards under all situations using different values of constraint parameters.

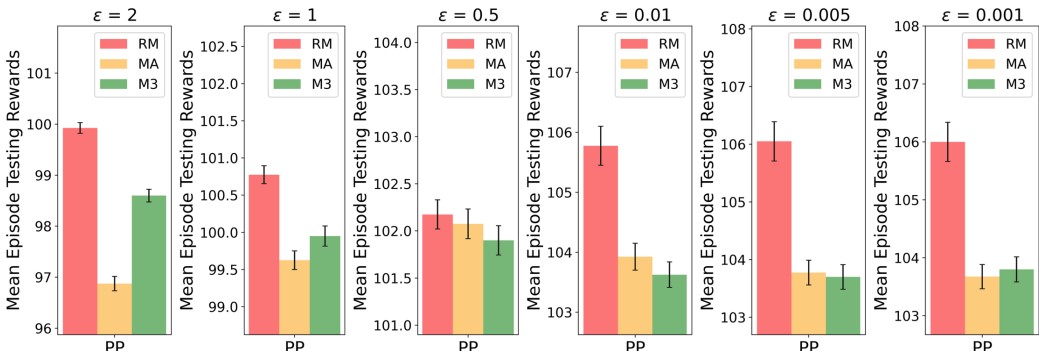

Figure 20: We test the performance of RMAAC(RM), MADDPG(MA), and M3DDPG(M3) policies under the attack of linear noise format when using different values of constraint parameters in the scenario Predator-Prey. RM denotes our robust MARL algorithm, i.e. RMAAC. MA denotes MADDPG, a MARL baseline algorithm. M3 denotes M3DDPG, a robust MARL baseline algorithm. Our RMAAC algorithm outperforms baseline algorithms in terms of mean episode testing rewards under all situations using different values of constraint parameters.

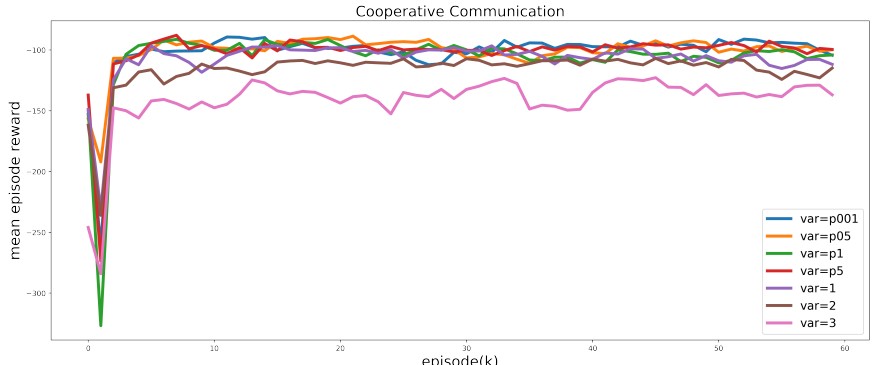

Figure 21: We train RMAAC policies using different values of variance in the scenario Cooperative Communication. In general, the smaller the variance is used, the higher the mean episode rewards RMAAC can achieve.

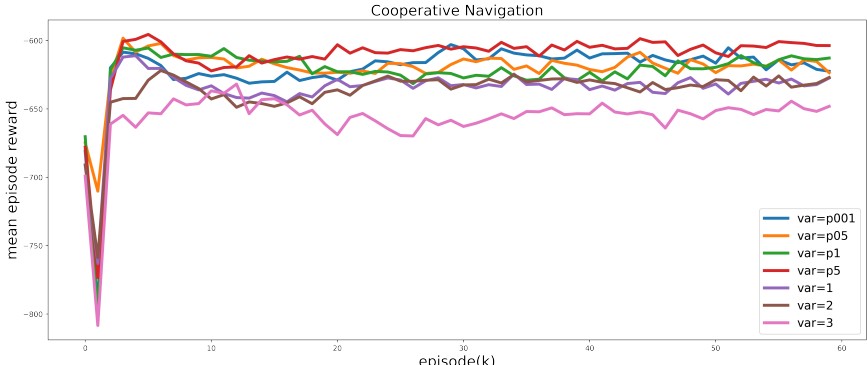

Figure 22: We train RMAAC policies using different values of variance in the scenario Cooperative Navigation. In general, the smaller the variance is used, the higher the mean episode rewards RMAAC can achieve.

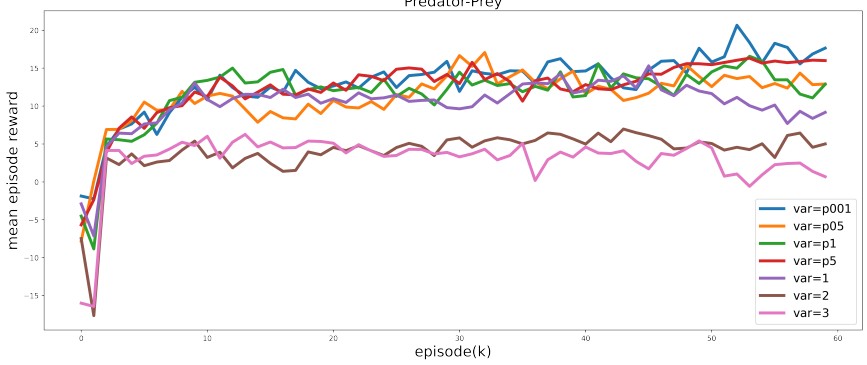

Figure 23: We train RMAAC policies using different values of variance in the scenario Predator-Prey. In general, the smaller the variance is used, the higher the mean episode rewards RMAAC can achieve.

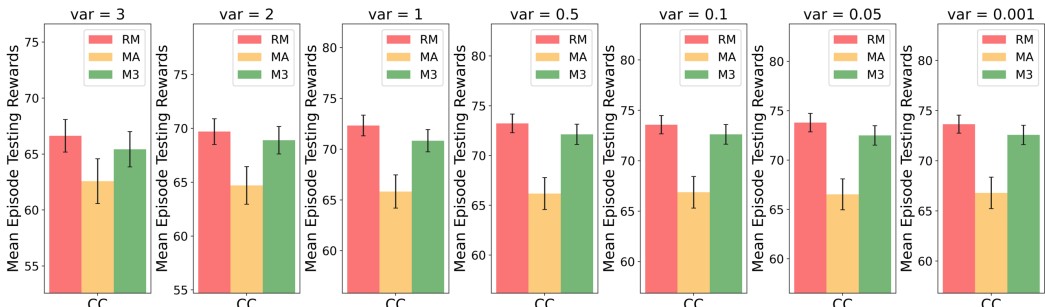

Figure 24: We test the performance of RMAAC(RM), MADDPG(MA), and M3DDPG(M3) policies under the attacks of Gaussian noise format with different variances in the scenario Cooperative Communication. RM denotes our robust MARL algorithm, i.e. RMAAC. MA denotes MADDPG, a MARL baseline algorithm. M3 denotes M3DDPG, a robust MARL baseline algorithm. Our RMAAC algorithm outperforms baseline algorithms in terms of mean episode testing rewards under all Gaussian noise formats with different variances.

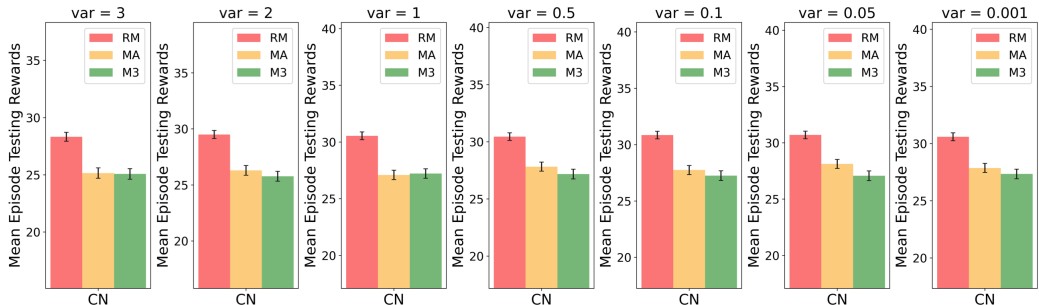

Figure 25: We test the performance of RMAAC(RM), MADDPG(MA), and M3DDPG(M3) policies under the attacks of Gaussian noise format with different variances in the scenario Cooperative Navigation. RM denotes our robust MARL algorithm, i.e. RMAAC. MA denotes MADDPG, a MARL baseline algorithm. M3 denotes M3DDPG, a robust MARL baseline algorithm. Our RMAAC algorithm outperforms baseline algorithms in terms of mean episode testing rewards under all Gaussian noise formats with different variances.

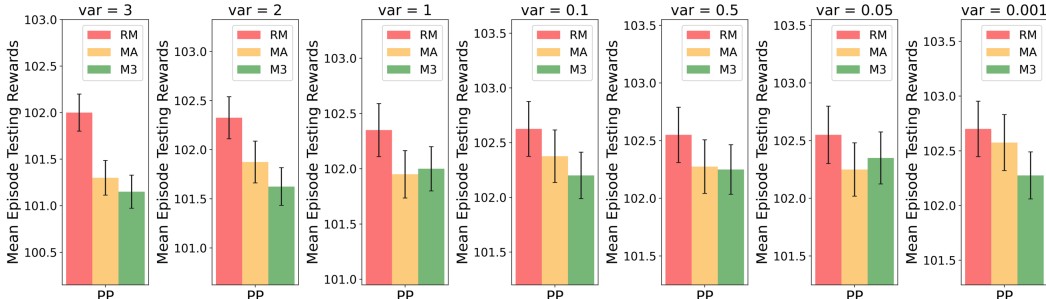

Figure 26: We test the performance of RMAAC(RM), MADDPG(MA), and M3DDPG(M3) policies under the attacks of Gaussian noise format with different variances in the scenario Predator-Prey. RM denotes our robust MARL algorithm, i.e. RMAAC. MA denotes MADDPG, a MARL baseline algorithm. M3 denotes M3DDPG, a robust MARL baseline algorithm. Our RMAAC algorithm outperforms baseline algorithms in terms of mean episode testing rewards under all Gaussian noise formats with different variances.

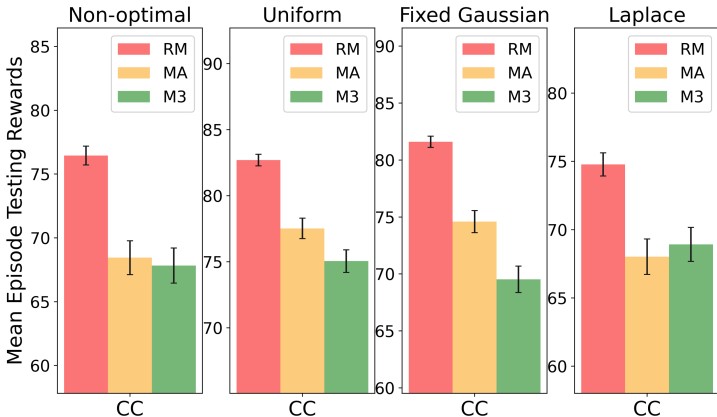

Figure 27: We test the performance of RMAAC(RM), MADDPG(MA), and M3DDPG(M3) policies under the attacks of different noise formats in the scenario Cooperative Communication. RM denotes our robust MARL algorithm, i.e. RMAAC. MA denotes MADDPG, a MARL baseline algorithm. M3 denotes M3DDPG, a robust MARL baseline algorithm. Our RMAAC algorithm outperforms baseline algorithms in terms of mean episode testing rewards under all kinds of attacks.

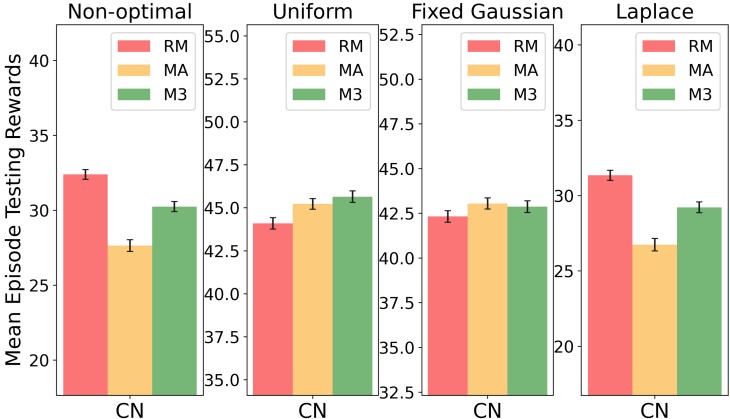

Figure 28: We test the performance of RMAAC policies under the attacks of different noise formats in the scenario Cooperative Navigation. RM denotes our robust MARL algorithm, i.e. RMAAC. MA denotes MADDPG, a MARL baseline algorithm. M3 denotes M3DDPG, a robust MARL baseline algorithm. Our RMAAC algorithm either outperforms or is close to baseline algorithms in terms of mean episode testing rewards under all kinds of attacks.

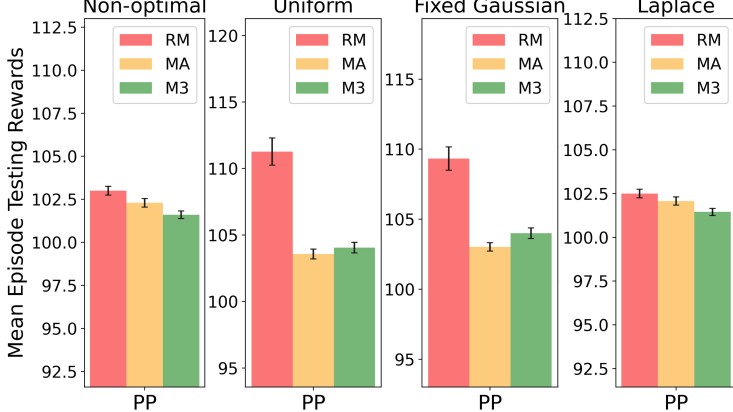

Figure 29: We test the performance of RMAAC policies under the attacks of different noise formats in the scenario Predator-Prey. RM denotes our robust MARL algorithm, i.e. RMAAC. MA denotes MADDPG, a MARL baseline algorithm. M3 denotes M3DDPG, a robust MARL baseline algorithm. Our RMAAC algorithm outperforms baseline algorithms in terms of mean episode testing rewards under all kinds of attacks.

