# OpenReview forum: "Robust Multi-Agent Reinforcement Learning with State Uncertainties"
_ICLR.cc/2023/Conference — Submitted to ICLR 2023_

### Official Review · Reviewer_qQaj · 2022-10-22

**Confidence:** 4
**Correctness:** 3
**Technical Novelty And Significance:** 3
**Empirical Novelty And Significance:** 3
**Recommendation:** 8

**Clarity, Quality, Novelty And Reproducibility:**

**Clarity:** The paper is generally well-written and conveys the main insights well.
**Quality & Novelty:** I agree that the framework of MG-SPA is new, and its related theoretical contributions are novel.
**Reproducibility:** The source code is provided in the supplementary material to reproduce the results.

**Strength And Weaknesses:**

**Strengths:**
1. This paper develops important theoretical contributions based on MG-SPA.
2. The algorithms are also developed in a principled way based on MG-SPA, and RMAAC shows the scalability.
3. The paper is generally well-written and conveys the methods clearly.

**Weakness:**
Because this paper's contribution focuses on robustness, adding robust MARL baselines in the evaluations would strengthen the paper.


**Questions:**
1. In Section 3.1, using the same notation $i$ for the adversary can be confusing compared to the agent $i$. Following the consistency, what about using $\tilde{i}$ for the adversary?
2. Which perturbation behaviors did adversaries learn in the MPE experiments (i.e., the amount of noise in the linear and Gaussian noise formats)?
3. How is the robustness performance affected by varying $\epsilon$?

**Summary Of The Paper:**

This paper develops a robust MARL framework that considers state uncertainty. The paper first formulates the MARL problem with state uncertainty as MG-SPA and develops theoretical contributions, including the new solution concept of Robust equilibrium and the equilibrium's existence. Then, based on the Bellman equations of MG-SPA, the Q-learning method (RMAQ) and actor-critic method (RMAAC) are developed. Finally, evaluations in the two-player game and MPE show robust performance against the adversaries' perturbations.

**Summary Of The Review:**

Overall, I have a positive evaluation of this paper because the paper formalizes state uncertainties as MG-SPA and develops its theoretical and practical contributions. However, I have a concern regarding the absence of robust MARL baselines, and I will make a final decision on the recommendation after the authors' response.

---

> ### Author Response · Authors · 2022-11-18
> **Response to Reviewer qQaj**
>
> **(1) Question to notation:**
>
> We thank the reviewer for this constructive comment. We have changed all “adversary i” to “adversary $\tilde{i}$” to resolve possible confusion. All corresponding subscripts and superscripts are modified as well. Please check the revised paper, especially the blue text.
>
> ------
>
> **(2) Question to perturbation behaviors:**
>
> As we defined in MG-SPA, the goal of the adversaries is to perturb the state information that the agents receive to minimize the agents’ rewards. The perturbation behaviors would be different under different states. For example, in Predator-Prayer, when we use Gaussian noise format with variance 1, constraint  parameter 0.5  in RMAAC to train agents and adversaries’ policies, we have the following instance.  when the true state is
>
> >[[0.0, 0.0, 0.8265, 0.2125, -1.1353, -0.6162, -1.0323, -0.2083, -1.7059, 0.0593, -0.1252, -0.5256, -1.8121, -0.306, 0.0, 0.0], [0.0, 0.0, -0.8794, 0.2718, 0.5707, -0.6755, 0.6736, -0.2676, 1.7059, -0.0593, 1.5807, -0.5849, -0.1062, -0.3654, 0.0, 0.0], [0.0, 0.0, 0.7013, -0.3131, -1.0101, -0.0906, -0.9071, 0.3173, 0.1252, 0.5256, -1.5807, 0.5849, -1.6869, 0.2196, 0.0, 0.0], [0.0, 0.0, -0.9856, -0.0936, 0.6769, -0.3101, 0.7798, 0.0977, 1.8121, 0.306, 0.1062, 0.3654, 1.6869, -0.2196]].
>
> The adversaries produce a noise
>
> >[[-0.0091, -0.0086, -0.0926, 0.1441, 0.0802, -0.0661, -0.0615, -0.0261, -0.0375, 0.069, 0.0652, -0.0036, -0.0027, 0.024, -0.0707, -0.1249], [-0.1756, -0.0184, 0.0406, -0.0009, 0.0034, -0.085, 0.0287, -0.0023, -0.1479, 0.0476, -0.0081, 0.1282, 0.0375, -0.033, -0.0467, -0.0479], [-0.1069, 0.205, 0.1318, 0.0303, -0.112, 0.0078, 0.0276, -0.0552, 0.0501, 0.0355, 0.1127, 0.0257, -0.1044, -0.145, 0.0546, -0.0544], [0.0454, 0.0198, 0.1075, 0.0324, -0.3062, 0.133, 0.1307, -0.1192, -0.0728, -0.0606, -0.0795, 0.028, -0.0526, 0.1522]].
>
> Then the perturbed state is
>
> >[[-0.0091, -0.0086, 0.7339, 0.3566, -1.0551, -0.6823, -1.0938, -0.2344, -1.7434, 0.1283, -0.06, -0.5292, -1.8148, -0.282, -0.0707, -0.1249], [-0.1756, -0.0184, -0.8388, 0.2709, 0.5741, -0.7605, 0.7023, -0.2699, 1.558, -0.0117, 1.5726, -0.4567, -0.0687, -0.3984, -0.0467, -0.0479], [-0.1069, 0.205, 0.8331, -0.2828, -1.1221, -0.0828, -0.8795, 0.2621, 0.1753, 0.5611, -1.468, 0.6106, -1.7913, 0.0746, 0.0546, -0.0544]].
>
> ------
>
> **(3) Question to the effect of constraint parameter $\epsilon$:**
>
> Thank you for this valuable comment. We add experiments to investigate the effect of using different $\epsilon$ in subsections C.2.4 and C.2.5.
>
> In subsection **C.2.4**, we train several RMAAC policies using linear noise format as the state perturbation function in three multi-agent scenarios. And the constraint parameter $\epsilon$ is respectively set as 0.01, 0,05, 0.1, 0.5, 1, and 2. We show the training process in **figures 15, 16, and 17**. In general, the larger the constraint parameter $\epsilon$ is, the stronger the adversaries can be, then the poorer performance the RMAAC policies should have. The performance of the RMAAC policies is close when the constraint parameters are less than 0.5.  For more details about the training setup and results, please check the revised paper, subsection C.2.4.
>
> In subsection **C.2.5**, we test RMAAC policies in perturbed environments where adversaries adopt linear noise format and different constraint parameters (0.01, 0.05, 0.1, 0.5, 1, 2). We test the performance of RMAAC policies,  MADDPG and M3DDPG policies in three multi-agent scenarios. MADDPG is a MARL baseline algorithm. M3DDPG is a robust MARL baseline algorithm. It turns out our RMAAC policies outperform MADDPG and M3DDPG policies in all three scenarios no matter which constraint parameter is used. Our RMAAC algorithm is robust to the state information attacks of linear noise format with different constraint parameters. For more details, please check the revised paper, in particular **subsection C.2.5, figures 18, 19, and 20**.
>
> We hope these experimental results and explanations could resolve the reviewer's concerns.
>
> ------
>
> **(4) Question to robust MARL baseline:**
>
> In our experiments, we use M3DDPG [1] as a robust MARL baseline. M3DDPG algorithm considers robustness with respect to the opponents’ policies altering.
>
> [1] S. Li, Y. Wu, X. Cui, H. Dong, F. Fang, and S. Russell, “Robust multi-agent reinforcement learning via minimax deep deterministic policy gradient,” in Proceedings of the AAAI Conference on Artificial Intelligence, vol. 33, no. 01, 2019, pp. 4213–4220.

---

> > ### Comment · Reviewer_qQaj · 2022-11-19
> > **Response to Rebuttal**
> >
> > I appreciate the authors for updating the paper based on my feedback. The response has addressed my questions.

---

> > > ### Comment · Area_Chair_vJ8m · 2022-11-24
> > > **Thanks for acknowledging, but please expand.**
> > >
> > > If all your concerns were addressed, does that mean you are planning to change your score?
> > > If not, why not?

---

> > > > ### Comment · Reviewer_qQaj · 2022-11-25
> > > > **Reviewer response follow-up**
> > > >
> > > > Yes, I agree with the authors that the theoretical contributions, paper clarity, and empirical evaluations have improved. I have updated my score from week accept to accept.

---

> > > ### Author Response · Authors · 2022-11-25
> > > **We would appreciate it if the reviewer could increase the rating**
> > >
> > > We would like to thanks for the reviewer's positive feedback for our response and paper revision, and agreeing that all the concerns and questions have been addressed. We would appreciate it if you could increase your rating for the paper! We believe that the theoretical contributions, novelties, and experimental eavaluations of this work are sufficient with the revision based on all the reviewers' valuable suggestions.

---

### Official Review · Reviewer_QZNZ · 2022-10-24

**Confidence:** 3
**Correctness:** 3
**Technical Novelty And Significance:** 3
**Empirical Novelty And Significance:** 3
**Recommendation:** 6

**Clarity, Quality, Novelty And Reproducibility:**

The manuscript is overall easy to follow. Proper examples are provided to help the readers understand the context and problem. Algorithm code is provided so it should be reproducible.


**Strength And Weaknesses:**

Strength:
- The authors provide a theoretical analysis of the MARL state uncertainty problem, which provides a theoretical base for the two designed algorithms.

Questions and concerns:
- In the evaluation section, it seems the results are based on a fixed mean and variance of the gaussian noise. How does the proposed algorithm perform under magnitude of perturbation compared to other algorithms?
- In the evaluation section, the attack policy seems only be the one trained with RMAAC, how does the algorithm perform under other types of attacked proposed in the existing literature?
- This might be within the scope of the work, but is it possible that the proposed algorithms find a Robust Equilibrium that is sub-optimal (e.g. if the problem has multiple RE)?
- What happens if the team only receive one single team reward at each time step? Can the current formulation still work?



**Summary Of The Paper:**

This work investigates the robustness of MARL under state uncertainty. The authors try to formally define the state uncertainty problem and provide a theoretical analysis. Based on the state uncertainty formulation, two algorithms (RMAQ and RMAAC) are designed to improve the robustness of MARL where there is state uncertainty.

**Summary Of The Review:**

This work is overall interesting as it is looking at an important problem to improve the robustness of MARL. I think further improving the evaluation section of this work can improve its quality. See above for more details.

---

> ### Author Response · Authors · 2022-11-18
> **Response to Reviewer QZNZ**
>
> **(1) Question to evaluation:**
>
> We thank the reviewer’s time and efforts in reviewing our manuscript. We have added experiments to explore the effect of using different values of variances **in the revision appendix subsection C.2.6 and C.2.7**. We’d like to point out that our results in the first version of the paper are not based on Gaussian noises with a fixed mean. In the experiments using Gaussian noises, the means of the Gaussian noises are not fixed but produced by adversarial policies. Therefore, under different states, the adversaries output different values as the means of the Gaussian noises.
>
> In **subsection C.2.6**, we train several RMAAC policies using Gaussian noise format as the state perturbation function in three multi-agent scenarios. And variance $\sigma$ is respectively set as 0.001, 0,05, 0.1, 0.5, 1, 2, and 3. We show the training process in **figures 21, 22, and 23**. In general, the larger the variance is, the stronger the adversaries can be, then the poorer performance the RMAAC policies should have. The performance of the RMAAC policies is close when the variances are less than 1. For more details about the training setup and results, please check the revised paper, subsection C.2.6.
>
> In **subsection C.2.7**, we test RMAAC policies in perturbed environments where adversaries adopt Gaussian noise format and different values of variances (0.001, 0.05, 0.1, 0.5, 1, 2, 3). We test the performance of RMAAC policies, MADDPG and M3DDPG policies in three multi-agent scenarios. MADDPG is a MARL baseline algorithm. M3DDPG is a robust MARL baseline algorithm. It turns out our RMAAC policies outperform MADDPG and M3DDPG policies in all three scenarios no matter which value of variance is used. Our RMAAC algorithm is robust to the state information attacks of Gaussian noise format with different values of variance. For more details, please check the revised paper, in particular **subsection C.2.7, figures 24, 25, and 26**.
>
> ------
>
> **(2) Question to the performance of RMAAC under other types of attacks:**
>
> We thank the reviewer for this constructive comment. We add experiments to investigate the performance of RMAAC under other types of attacks. Please check **subsection C.2.8** in the revised paper. In C.2.8, we test the well-trained RMAAC policies in perturbed environments where adversaries adopt different types of attacks (non-optimal, Uniform, fixed Gaussian and Laplace). We test RMAAC policies, MADDPG and M3DDPG policies in three multi-agent scenarios. MADDPG is a MARL baseline algorithm, M3DDPG is a robust MARL baseline algorithm. Most of the time, our RMMAC policies outperform the baseline policies in terms of mean episode testing rewards. In general, our RMAAC algorithm is robust to different types of state information attacks. For more details, please check the revised paper, in particular **subsection C.2.8, figures 27, 28 and 29.**
>
> We hope these experimental results and explanations could resolve the reviewer's concern to the evaluation.
>
> ------
>
> **(3) Question to find suboptimal solutions:**
>
> (i) We propose the RMAQ algorithm with a convergence guarantee as proved in Theorem 3.9. It is possible that the algorithm converges to a sub-optimal Robust Equilibrium. Even for an N player general-sum game without any uncertainties, it is possible for the Q-learning algorithm to converge to a sub-optimal Nash equilibrium as discussed in [1].
>
> (ii) We also propose the RMAAC algorithm to deal with high-dimensional state and action spaces with deep neural networks to approximate the policy functions. It is possible that our proposed RMAAC algorithm finds a sub-optimal robust equilibrium. In general, deep learning-based algorithms do not have a convergence guarantee.
>
> [1] J. Hu and M. P. Wellman, “Nash q-learning for general-sum stochastic games,” Journal of machine learning research, vol. 4, no. Nov, pp. 1039–1069, 2003.
>
> ------
>
> **(4) Question to extend formulation to team game:**
>
> Yes, the current formulation still works when the team only receives one single team reward at each time step. This is a special case of the current formulation.

---

> > ### Comment · Reviewer_QZNZ · 2022-12-12
> > **Reply to Authors**
> >
> > I would like to thank the authors for their response and revision. The above response has addressed my concern.

---

> > > ### Author Response · Authors · 2022-12-12
> > > **Thank you for the positive feedback**
> > >
> > > We thank you for the positive feedback on our response and paper revision. We are happy to see our response has addressed your concern! We would appreciate it if you could increase your rating for our paper. Thank you.

---

### Official Review · Reviewer_dBqZ · 2022-10-25

**Confidence:** 4
**Correctness:** 2
**Technical Novelty And Significance:** 3
**Empirical Novelty And Significance:** 3
**Recommendation:** 5

**Clarity, Quality, Novelty And Reproducibility:**

- Quality: The set of results are interesting at the fist glance. However, the formal setting may not be adequate for the problem at hand as explained above. In addition, it is not clear whether the characterization results are entirely precise/correct. The experimental results could also be further expanded, as I indicated in my previous comments.
---
- Clarity: In my option, the paper can be considerably improved in terms of clarity. I was able to follow the main ideas, but the problem is that the technical content was a bit hard to follow due to confusing notation. For example, Definition 3.3 uses $v_*$ notation, without stating what it refers to - does it refer to the RE quantities in definition 3.2 or to the equilibrium of the corresponding EFG? Note that in the proofs $v^*$ is used for EFG equilibria. Similarly, it seems that Theorem 3.13 considers deterministic policies without explicitly stating this, whereas the setting of the paper uses stochastic policies. Also, the theorem considers continuous actions, without stating this, whereas Eq. (4) considers discrete actions. Apart from that, the submission contains quite a few typos.
---
- Novelty: The problem setting appears to be novel and builds on recent works in single-agent adversarial RL and robust multi-agent RL, complementing the latter. This is arguably the main strength of the paper.
---
- Reproducibility: In terms of experimental results, the paper provides a sufficient description of the experimental test-bed. One part that could be improved is the description of the network architectures used in the novel algorithms. This seems to be somewhat hidden in the appendix. In terms of the theoretical results, as I explained in my comments above, some of the proofs would benefit from more detailed descriptions.

**Strength And Weaknesses:**

Arguably, the main strength of the paper is that it studies what appears to be a novel setting - robustness to perturbations of agents' observations in MARL. The results include a theoretical characterization of the formal setting, algorithmic approaches for deriving equilibrium policies, and experimental validation of the proposed algorithms. In other words, the paper contains a rather complete set of results for this framework. Having said that, I have several concerns about the following submission, listed below:
- First, it is not clear to me whether this optimization framework leads to optimally robust policies. More specifically, considering only stochastic stationary policies, as assumed in the preliminaries, may not suffice in this case. Namely, an agent/victim decides on its actions based on corrupted observations, which are correlated across time since adversarial perturbations have a bounded budget and have a functional form. Hence, agents can benefit from having history-dependent policies, e.g., form a belief about the underlying state. Due to this, it is also not clear that the equilibrium notion in Def 3.2 is the right solution concept - shouldn't agent be deciding based on their beliefs about the underlying state, and if so, why is it important to consider a robust version of MPE (Markov perfect equilibrium)?
- Some parts of the formal setting/characterization results may not be precise enough. I tried to check some of the results, but the notation is quite confusing. I give examples in the next section. Some of the results are also not clear to me. E.g., in the proof of Prop. 3.5., I didn't follow why the 3rd equation is due to the NE of the corresponding EFG - why  are $\pi_*^v$ and $\rho_*^v$ important for this inequality and why does the inequality hold if we have $v^i = [1, 1, 1, ..]$ and $u^i = [0, 0, 0, ...]$?
- I'm puzzled by potential implications of Theorem 3.8 and Theorem 3.9. Wouldn't they imply that there always exists an RE? If $v^*$ always exists and given Theorem 3.9, can't we  find an equilibrium by solving the EFG corresponding to $v^*$? If yes, why is Theorem 3.10 important since it covers very specific case of the setting? If not, how does the algorithm from section 3.3 work? It would be helpful to have an additional discussion that compares the theoretical aspects of this work to those from Hu & Wellman (2003), including the assumption needed for convergence (e.g., assumption 3.11c this work vs assumption 3 from Hu & Wellman (2003)).
- Robustness in this paper is measured against adversarial perturbations that have a specific form, given via function $f$. I'm wondering whether the corresponding robust policies generalize across different functions $f$. Judging by Fig. 12, they don't appear to be consistently better than the baselines. In practice, we presumable wouldn't know the exact form of adversarial perturbations that will occur at test-time, so it would be good to include results that show performance under noise models which were not used in training. Furthermore, why weren't policies that optimally disturb the environment trained for each algorithm separately (keeping these algorithms fixed)? It seems that the current adversarial perturbation policies may favor RMAAC, especially if an equilibrium point was not reached in the training phase.

**Summary Of The Paper:**

The paper considers multi-agent RL setting where agents' observations are corrupted at test-time. The paper proposes a novel optimization framework that models adversarial perturbations on agents' observations as additional players in the underlying Markov game. The authors provide characterization results for the considered framework, deriving a version of Bellman conditions for this framework, as well as a sufficient conditions for the existence of a robust equilibrium. They further propose two algorithms for finding equilibrium policies, based on Q-learning and actor-critic approaches, and experimentally test their efficacy in standard MARL environments.

**Summary Of The Review:**

The paper explores a novel MARL setting, and reports a set of results that is of potential interest to researcher working on robust RL. On the flip side, the current submission could benefit from additional discussion points that would clearly explain why more general settings are not needed in this setting. The current set of results, although looking promising, could be extended. The clarity of the paper could be improved as well.

---

> ### Author Response · Authors · 2022-11-18
> **Response to Reviewer dBqZ (Part 3)**
>
> **(5) Question to the robustness and experiments:**
>
> We thank the reviewer for these valuable comments. Due to the space limit, we put the experimental results under other types of attacks in the appendix. We tested the performance of our proposed RMAAC algorithm under a random noise attack. For the reason why our robust methods do not outperform baselines consistently, in the appendix, we have explained from the lens of robust and distributionally robust optimization:
>
> >“This situation also happens in robust optimization [Beyer & Sendhoff, 2007; Boyd & Vandenberghe, 2004] and distributionally robust optimization [Delage & Ye, 2010; Rahimian & Mehrotra, 2019] that the robust solution outperforms other non-robust solutions in the worst-case situation. Similarly, for single-agent RL with state perturbations, robust policies perform better compared with baselines under state perturbations [Zhang et al. (2020b)]. But the robust solutions may get relatively poor performance compared with other non-robust solutions when there is no uncertainty or perturbation in the environment even in a single agent RL problem [Zhang et al. (2020b)]. Improving the robustness of the trained policy may sacrifice the performance of the decisions when the perturbations or uncertainties do not happen. That's why our RMAAC policies only beat all baselines in one scenario when the state uncertainty is eliminated. However, for many real-world systems we can not assume the agents always have accurate information of the states. Hence, improving the robustness of the policies is very important for MARL as we explained in the introduction of this work. It is worth noting that our RMAAC policies also work well in environments with random perturbations instead of worst-case perturbations. As shown in Fig. 12, the performance of our RMAAC policies outperforms the baselines in most scenarios when random noise is introduced into the state.”
>
> We have highlighted the above explanations in blue in the revised paper.
>
> In the revised paper, we also **add more experiments to validate that our RMAAC algorithm is robust to different types of attacks**. Please check **subsection C.2.8** in the revised paper. In C.2.8, we test the well-trained RMAAC policies in perturbed environments where adversaries adopt different types of attacks. We test RMAAC policies, MADDPG and M3DDPG policies in three multi-agent scenarios. MADDPG is a MARL baseline algorithm, M3DDPG is a robust MARL baseline algorithm that utilizes adversarial learning to train robust policies [1]. Most of the time, our RMMAC policies outperform the baseline policies in terms of mean episode testing rewards. In general, our RMAAC algorithm is robust to different types of state information attacks. Please check **figures 27, 28 and 29**.
>
> [1] Li, Shihui, et al. "Robust multi-agent reinforcement learning via minimax deep deterministic policy gradient." Proceedings of the AAAI Conference on Artificial Intelligence. Vol. 33. No. 01. 2019.
>
> ------
>
> **(6) Concern to the notation:**
>
> We thank the reviewer for these constructive comments, and we have added more explanations to the notations. Now we use $(\rho^v_*, \pi^v_*)$ to denote an NE policy for the EFG $(v^1, \cdots, v^N, -v^1, \cdots, -v^N,)$. Please check Lemma A.7 in the revised paper to get more details.
>
> ------
>
> **(7) Concern to Theorem 3.13 and Eq. (4):**
>
> We thank the reviewer for pointing out this concern, and we have added descriptions in the revision to clarify that Theorem 3.13 focuses on deterministic policy gradients and considers continuous action space. We also corrected the typos. Specifically, we have added:
>
> >“Note that we here parameterize all policies $\pi^i, \rho^{\tilde{i}}$ as deterministic policies.”
>
> Please check the blue text in the revised paper, in particular, in subsection 3.4.

---

> > ### Comment · Reviewer_dBqZ · 2022-11-26
> > **Thank you for your response**
> >
> > Thank you for your response. However, I don't think it fully addressed my main concerns, including the ones I list below.
> >
> > -----
> > **Formal setting**
> >
> > I remain to believe that one should use history dependent policies in this formal setting. This appears to be critical, otherwise, the results are not derived in a principled manner.  The arguments in your response seem to suggest that it's ok to consider Markovian policies because POMDPs are hard to solve, which I don't find convincing. After all, one would still use history dependent policies when solving POMDPs, even though finding an optimal one is computationally intractable. My point is that the abstraction you use in the paper, i.e., the formal model, determines what the *right* policy space is.  Whether it's hard to find an optimal policy is a different problem, and can be addressed through relaxations.
> >
> > -----
> > **Generalizability of the results**
> >
> > In the response, you claim that:
> >
> > > Our theory and algorithm are generalizable in which Markovian policies can be extended to history-dependent policies.
> >
> > If your theory generalizes to history dependent policy, why not just modify the setting? Unfortunately, I cannot verify this argument without seeing the modified proofs that formally show this.  The same holds for algorithms and experiments: while the algorithms may be possible to generalize (although, it's not entirely clear to me why the results in Sec 3.3 would immediately follow), the experimental results could end up being different. My main point is that one could end up significantly improving the performance of an agent by including history (e.g., by having access to the last few observations, the agent's policy could try to filter out the noise). Given the problem setting, I believe this analysis would be highly valuable, and in experiments could be accommodated by utilizing an appropriate policy space (i.e., DNN architecture).
> >
> > -----
> > **Theoretical results**
> >
> > The current set of theoretical results appears to be too restrictive, so it's not clear to me how significant the theoretical contribution are. It would also be nice if the authors actually acknowledge that some of their proofs/results had **mistakes** and/or were **incomplete**.  For example, the claim in the response:
> >
> > > We have fixed the typo in the third inequality and added more explanations in the proof
> >
> > doesn't reflect the fact that the new version of Proposition 3.5 is not the same as the old one: the new one requires assumption 3.4, and the new version of this assumption has additional conditions. Similarly for Theorem 3.8 and 3.9 in the old version of the paper, were the statements correct or incorrect?
> >
> > I also didn't follow your discussion related to Theorem 3.8 (in the revised paper) and Assumption 3.4. The claim:
> >
> > > In particular, under Assumption 3.4, an NE of the 2N-player EFG exists, which has been proved in Lemma A.6 in Appendix A.1
> >
> > in Section 3.3 and the claim:
> >
> > > In particular, we have proved that Assumption 3.8-(3) holds when Assumption 3.4 holds.
> >
> > in your response seem to suggest that, at the moment, you can only guarantee convergence under assumption 3.4. You then argue that:
> >
> > > In practice, this assumption is found to be not necessary for the learning algorithm to converge [1,2].
> >
> > However, this argument seems to be based on Q-learning type of algorithms applied to different settings, not the one studied in this paper. On a separate note, the statement of Lemma A.6 doesn't seem to assume 3.4, whereas your claim above seems to suggest that the result depends on this assumption: is the current statement correct?
> >
> > -----
> > Unfortunately, the response didn't help me appreciate more the contributions of this work. In fact, the new version of the theoretical results appear to be weaker, arguably because the previous versions were not correct.

---

> > > ### Author Response · Authors · 2022-12-06
> > > **2nd Round Response to Reviewer dBqZ (Question 1- Continue) [7/7]**
> > >
> > > >(Definitions of Value Functions) $v^{\pi,\rho} = (v^{\pi,\rho,1}, \cdots, v^{\pi,\rho,N}), q^{\pi, \rho} = (q^{\pi,\rho,1}, \cdots, q^{\pi,\rho,N})$ are defined as the state-value function or value function for short, and the action-value function, respectively. The $i$th element $v^{\pi,\rho,i}$ and $q^{\pi,\rho,i}$ are defined as $$q^{\pi,\rho,i}(s, a, b) = \mathbb{E} \left[ \sum\_{t=1}^{\infty} \gamma^{t-1} r\_t^i |
> > >             s\_1 = s, a\_1 = a, b\_1 = b, a\_t \sim \pi(\cdot | \tilde{s}\_{h,t}), b\_t \sim \rho(\cdot | s\_{h,t}), \tilde{s}\_{h,t}^i = (f(s\_t^i, b\_t^{\tilde{i}}), \tilde{s}\_{t-1}, \cdots, \tilde{s}\_{t-h+1}) \right],$$
> > >             $$v^{\pi,\rho,i}(s) = \mathbb{E} \left[ \sum\_{t=1}^{\infty} \gamma^{t-1} r\_t^i |
> > >             s\_1 = s, a\_t \sim \pi(\cdot | \tilde{s}\_{h,t}), b\_t \sim \rho(\cdot | s\_{h,t}), \tilde{s}\_{h,t}^i = (f(s\_t^i, b\_t^{\tilde{i}}), \tilde{s}\_{t-1},\cdots, \tilde{s}\_{t-h+1})\right],$$ respectively.
> > >
> > > We can find the main difference between the new formulation (history dependent policy) and old formulation (Markov policy) is the definition and notations of policies. In the rest of paper, we will accordingly use $\pi(\cdot| \tilde{s}\_h)$ and $\rho(\cdot|s\_h)$ instead of $\pi(\cdot| \tilde{s})$ and $\rho(\cdot|s)$. In the old formulation, we construct EFG based on the current state $s\_t$. In the new formulation, we construct EFG based on the new defined states $s\_h$ and $\tilde{s}\_{h}$ that includes the current state $s\_t$ and history state information $s\_{t-1}, \cdots, s\_{t-h+1}, \tilde{s}\_{t-1}, \cdots, \tilde{s}\_{t-h+1}$. Then the subsequent theoretical analysis still hold with $s\_{h,t}$ and $\tilde{s}\_{h,t}$ to represent the states in the analysis. Hence, in the revision paper, we did not repeat the detail process of problem formulation and theoretical analysis, where the writing can be very repetitive compared to our history independent policy version. We will add one remark in the final version, to explain that we can consider history dependent policies by defining $s\_{h,t}$ and $\tilde{s}\_{h,t}$, and the problem formulation, theoretical analysis should also be adapted with the history of state definition.
> > >
> > > We also conducted extra experiments in which RMAAC algorithm is used to train history dependent policies. Other than the current information, We also use history information in the latest three time frames, i.e. $h = 4$. In the following table, we show the mean and variance of mean episode rewards in 10 runs. We can see that in all five scenarios, history dependent policies outperform Markov policies.
> > >
> > > | Scenarios                      | History Dependent Policy | Markov Policy      |
> > > |--------------------------------|--------------------------|--------------------|
> > > | Cooperative communication (CC) | -52.83 $\pm$ 1.51        | -54.75 $\pm$ 3.03  |
> > > | Cooperative navigation (CN)    | -208.19 $\pm$ 1.68       | -210.41 $\pm$ 1.13 |
> > > | Physical deception (PD)        | 7.72 $\pm$ 0.33          | 5.71 $\pm$ 0.19    |
> > > | Keep away (KA)                 | -20.69 $\pm$ 0.09        | -21.18 $\pm$ 0.14  |
> > > | Predator prey (PP)             | 7.10 $\pm$ 0.17          | 6.116 $\pm$ 0.24   |
> > >
> > > [1] Huan Zhang, Hongge Chen, Chaowei Xiao, Bo Li, Mingyan Liu, Duane Boning, and Cho-Jui Hsieh.
> > > Robust deep reinforcement learning against adversarial perturbations on state observations. Advances
> > > in Neural Information Processing Systems, 33:21024–21037, 2020.
> > >
> > > [2] Huan Zhang, Hongge Chen, Duane Boning, and Cho-Jui Hsieh. Robust reinforcement learning on state
> > > observations with learned optimal adversary. arXiv preprint arXiv:2101.08452, 2021.
> > >
> > > [3] Michael Everett, Bj ̈orn L ̈utjens, and Jonathan P How. Certifiable robustness to adversarial state un-
> > > certainty in deep reinforcement learning. IEEE Transactions on Neural Networks and Learning Systems,
> > > 2021.

---

> > > ### Author Response · Authors · 2022-12-06
> > > **2nd Round Response to Reviewer dBqZ (Question 1) [6/7]**
> > >
> > > **(1) Question to Formal setting and Generalizability of the results**
> > >
> > > We thank the reviewer for further clarifying the question. First, we would like to clarify the generalization steps to extend Markov policy to history dependent policy. We first introduce $s\_{h,t}^i$, a concatenated state consisting of true historical states, and $\tilde{s}\_{h,t}^i$,  a concatenated state consisting of perturbed historical states.
> > >
> > > We then define history dependent policies $\pi^i(\cdot|\tilde{s}\_{h,t}^i)$      and           $\rho^{\tilde{i}}(\cdot|s\_{h,t}^i)$  in an MG-SPA, where the subscripts $h$ and $t$ respectively denote "historical" and time step, the superscript $i$/$\tilde{i}$ denote the index of agent/adversary. We give the new formulation using history-dependent policies as follows:
> > >
> > > > We use a tuple $\tilde{G} := (\mathcal{N}, \mathcal{M}, \{S^i\}\_{i \in \mathcal{N}}, \{A^i\}\_{i \in \mathcal{N}}, \{B^{\tilde{i}}\}\_{\tilde{i} \in \mathcal{M}}, \{r^i\}\_{i \in \mathcal{N}}, p, f, \gamma)$ to denote a Markov game with state perturbation adversaries (MG-SPA). In an MG-SPA, we introduce an additional set of adversaries {$\mathcal{M} = \{\tilde{1}, \cdots, \tilde{N}\}$} to a Markov game (MG) with an agent set $\mathcal{N}$. Each agent $i$ is associated with an adversary $\tilde{i}$ and a true state $s^i \in {S}^i$ in the absence of adversarial perturbation. Each adversary $\tilde{i}$ is associated with an action $b^{\tilde{i}} \in {B}^{\tilde{i}}$ and the same state $s^i \in {S}^i$ as agent $i$. We define the adversaries' joint action as $ b = (b^{\tilde{1}}, ..., b^{\tilde{N}}) \in {B}$, ${B} = {B}^{\tilde{1}} \times \cdots \times {B}^{\tilde{N}}$. At time $t$, adversary $\tilde{i}$ can manipulate the corresponding agent $i$'s state. We consider history dependent policies for adversaries. Once adversary $\tilde{i}$ gets the true state $s^i\_t$ at time $t$, it chooses an action $b^{\tilde{i}}\_t$ according to a history-dependent policy $\rho^{\tilde{i}}: S\_h^i \rightarrow \Delta(B^{\tilde{i}})$, where $s\_{h,t}^i = (s_t^i, \cdots, s\_{t-h+1}^i) \in S\_h^i $ is a concatenated state consists of the latest $h$ states. According to a perturbation function $f$, adversary $\tilde{i}$ perturbs state $s^i\_t$ to $\tilde{s}^i\_t = f(s^i\_t, b^{\tilde{i}}\_t) \in {S}^i$. Here we define the adversaries' joint policy $\rho = \prod_{\tilde{i} \in \mathcal{M}}\rho^{\tilde{i}}: {S}\_h \rightarrow \Delta(B)$, where $S\_h = S_h^1 \times S\_h^2 \times \cdots \times S\_h^N$. The definitions of agent action and agents' joint action are the same as their definitions in an MG. Agent $i$ chooses its action $a^i_t$ for $\tilde{s}^i\_{h,t} = (\tilde{s}\_t^i, \cdots, \tilde{s}\_{t-h+1}^i) \in S_h^i$ with probability $\pi^i (a^i\_t|\tilde{s}^i\_{h,t})$ according to a history dependent policy $\pi^i: {S}^i\_h \rightarrow \Delta(A^i)$. Agents execute the agents' joint action $a\_t$, then at time $t+1$, the joint state $s\_t$ turns to the next state $s_\{t+1}$ according to a transition probability function $p: {S} \times {A} \times {B} \rightarrow \Delta({S})$. Each agent $i$ gets a reward according to a state-wise reward function $r^i\_t: {S} \times {A} \times {B} \rightarrow {\mathbb{R}}$. Each adversary $\tilde{i}$ gets an opposite reward $-r^i\_t$. In an MG, the transition probability function and reward function are considered as the model of the game. In an MG-SPA, the perturbation function $f$ is also considered as a part of the model, i.e., the model of an MG-SPA is consisted of $f,p$ and $\{r^i\}\_{i \in \mathcal{N}}$.
> > > To incorporate realistic settings into our analysis, we restrict the power of each adversary, which is a common assumption for state perturbation adversaries in the RL literature [1,2,3]. We define perturbation constraints $\tilde{s}^i \in \mathcal{B}\_{dist}(\epsilon, s^i) \subset S^i$ to restrict the adversary $\tilde{i}$ to perturb a state only to a predefined set of states. $\mathcal{B}\_{dist}(\epsilon, s^i)$ is a $\epsilon$-radius ball measured in metric $dist(\cdot, \cdot)$, which is often chosen to be the $l$-norm distance: $dist(s^i, \tilde{s}^i) = \|s^i - \tilde{s}^i\|\_l$. We omit the subscript $dist$ in the following context. For each agent $i$, it attempts to maximize its expected sum of discounted rewards, i.e. its objective function $J^i(\pi, \rho) = \mathbb{E} \left[ \sum_{t=1}^{\infty} \gamma^{t-1} r\_t^i(s_t,a_t, b_t) | s\_1 = s, a\_t \sim \pi(\cdot | \tilde{s}\_{h,t}), b\_t \sim \rho(\cdot | s\_{h,t}) \right]$.
> > > Each adversary $\tilde{i}$ aims to minimize the objective function of agent $i$ and is considered as receiving an opposite reward of agent $i$, which also leads to a value function $-J^i(\pi, \rho)$ for adversary $\tilde{i}$. We further define the value functions in an MG-SPA as follows:

---

> > > ### Author Response · Authors · 2022-12-06
> > > **2nd Round Response to Reviewer dBqZ (Question 2-(4,5,6)) [5/7]**
> > >
> > > **(Question 2-(4)): in your response seem to suggest that, at the moment, you can only guarantee convergence under assumption 3.4.**
> > >
> > > Theorem 3.9 is proved based on Assumption 3.8. The convergence is guaranteed under Assumption 3.8.
> > >
> > >
> > > >(Assumption (new) 3.8)
> > > >
> > > >(1) State and action pairs have been visited infinitely often.
> > > >
> > > >(2) The learning rate $\alpha_t$ satisfies the following conditions: $0 \leq \alpha_t < 1$, $\sum\_{t \geq 0}\alpha_t^2 \leq \infty$; if $(s,a,b) \neq (s\_t, a\_t, b\_t)$, $\alpha\_t(s, a, b) = 0$.
> > > >
> > > >(3) An NE of the $2N$-player EFG based on $(q\_t^1, \cdots, q\_t^N, -q\_t^1, \cdots, -q\_t^N)$ exists at each iteration $t$.
> > >
> > >
> > > Assumption 3.4 is a sufficient but not necessary condition for Assumption 3.8-(3). These claims “In particular, under Assumption 3.4, an NE of the 2N-player EFG exists, which has been proved in Lemma A.6 in Appendix A.1” and “In particular, we have proved that Assumption 3.8-(3) holds when Assumption 3.4 holds.” are discussions about Assumption 3.8-(3) and are used to give an example when Assumption 3.8-(3) held, e.g. Assumption 3.8-(3) holds under Assumption 3.4. We do not mean the convergence can only be guaranteed under Assumption 3.4.
> > >
> > >
> > > **(Question 2-(5)): You then argue that: "In practice, this assumption is found to be not necessary for the learning algorithm to converge [1,2]." However, this argument seems to be based on Q-learning type of algorithms applied to different settings, not the one studied in this paper.**
> > >
> > > We thank the reviewer for further clarifying her/his question. Your understanding of this sentence that "this argument seems to be based on Q-learning type of algorithms applied to different settings,
> > > not the one studied in this paper." is correct. This sentence is to answer the question "compare assumptions in our paper and in [7]" in the first round response.
> > >
> > > >(Our first round response) "These kinds of assumptions (the existence of NE for some games during the algorithms learning process) have been widely accepted and used by the research community. In practice, this assumption is found to be not necessary for the learning algorithm to converge [1,2].”
> > >
> > > “This assumption” is not the Assumption 3.8-(3) in our paper but the Assumption 3 used in [7]. "the learning algorithm" refers to the Nash Q-learning algorithm in Hu's paper. And this sentence is a summary of Hu's origin text as following:
> > >
> > > > ” Nevertheless, in our experiments reported below, we found that convergence is not necessarily so sensitive to properties of the stage games during learning. In a game that satisfies the property at equilibrium, we found consistent convergence despite the fact that stage games during learning do not satisfy Assumption 3. This suggests that there may be some potential to relax the conditions in our convergence proof, at least for some classes of games. ” [ [7], section 4.2]
> > >
> > >
> > > **(Question 2-(6)): On a separate note, the statement of Lemma A.6 doesn't seem to assume 3.4, whereas your claim above seems to suggest that the result depends on this assumption: is the current statement correct?**
> > >
> > > Yes, the current statement of Lemma A.6 is correct.  Lemma A.6 does rely on Assumption 3.4, in particular, 3.4-(2) and 3.4-(5), which are rephrased instead of being referenced in the statement of the lemma. This can be seen from a comparison between Lemma A.6 and Assumptions 3.4-(2) and 3.4-(5):
> > >
> > > >(Lemma (new) A.6) Suppose $v^1 = \cdots = v^N$, and $S, A$ are finite. An NE $(\lambda_*, \chi_*)$ of the EFG based on $(v^1, \cdots, v^N, -v^1, \cdots, -v^N)$ exists.
> > >
> > > >(Assumption (new) 3.4-(2)) (2) Finite state and action spaces; all ${S}^i, A^i, B^{\tilde{i}}$ are finite.
> > >
> > > >(Assumption (new) 3.4-(5)) (5) All agents share one common reward function.
> > >
> > >
> > > We thank the reviewer's precious suggestion. In the final version, we will use direct referencing in the statement of Lemma A.6 to better highlight the dependence on Assumption 3.4: "Under Assumption 3.4, in particular 3.4-(2), 3.4-(5), An NE $(\lambda_*, \chi_*)$ of the EFG based on $(v^1, \cdots, v^N, -v^1, \cdots, -v^N)$ exists." to make it more clear.
> > >
> > > **Reference**
> > >
> > > [4] Tamer Bas ̧ar and Geert Jan Olsder. Dynamic noncooperative game theory. SIAM, 1998.
> > >
> > > [5] Burkhard C Schipper. Kuhn’s theorem for extensive games with unawareness. Available at SSRN
> > > 3063853, 2017.
> > >
> > > [6] B Slantchev. Game theory: Perfect equilibria in extensive form games. UCSD script, 2008.
> > >
> > > [7] Junling Hu and Michael P Wellman. Nash q-learning for general-sum stochastic games. Journal of
> > > machine learning research, 4(Nov):1039–1069, 2003.

---

> > > ### Author Response · Authors · 2022-12-06
> > > **2nd Round Response to Reviewer dBqZ (Question 2-(2,3)) [4/7]**
> > >
> > > **(Question 2-(2)): Similarly for Theorem 3.8 and 3.9 in the old version of the paper, were the statements correct or incorrect?**
> > >
> > > Yes, the statements of Theorems 3.8 and 3.9 in the old version were correct. We'd like to compare the new version and the old version as follows.
> > >
> > > >(Existence of optimal value function, Theorem (old) 3.8.)
> > > Suppose $0 \leq \gamma  < 1$, $S$ is finite or countable, and $r(s,a,b)$ is bounded, there exists a unique $v\_* \in \mathbb{V}$ satisfying $Lv_* = v_*$, i.e. for all $i \in \mathcal{N}, L^i v^i\_* = v^i\_*$. Thus a optimal value function exists.
> > >
> > > >(Robust Equilibrium and Bellman Equation, Theorem (old) 3.9.)
> > > A joint policy $d_* = (\pi_*, \rho_*)$ where $\pi_* = (\pi^1_*, \cdots, \pi^N_*)$ and $\rho_* = (\rho^1_*, \cdots, \rho^N_*)$ is a robust equilibrium if and only if $v^{d_*}$ satisfies all Bellman Equations.
> > >
> > > As we can see above that Theorem 3.8 states that there exists a unique solution to the Bellman Equations. Considering in Theorem 3.7 we have proved that a solution to the Bellman Equations is an optimal value function, Theorem 3.8 also indicates that a optimal value function exists. Theorem 3.9 shows the relationship between an RE and a solution to the Bellman Equations. We show the modified versions of Theorem 3.8 (Theorem 3.7-(2)) and 3.9 (Theorem 3.7-(3)) as follows. We can see that Theorem 3.7-(2) is almost the same as Theorem 3.8 (old version) by only deleting the corollary that "Thus a optimal value function exists" by applying Theorem 3.7-(1) (new version)/Theorem 3.7 (old version). Theorem 3.7-(3) is also almost same to Theorem 3.9 (old version). The only difference is that we require $v^{d_*}$ is the optimal value function instead of requiring $v^{d\_*}$ is the solution to the Bellman Equations, while the solution to the Bellman Equations is the optimal value function which has been proved in Theorem 3.7-(1) (new version)/Theorem 3.7 (old version).
> > >
> > > >(Theorem (old) 3.7 $\rightarrow$ Theorem (new) 3.7-(1)) (1) (Solution of Bellman Equation)
> > > A value function $v\_* \in \mathbb{V}$ is an optimal value function if for all $i \in \mathcal{N}$, the point-wise value function $v^i\_* \in V$ satisfies the corresponding Bellman Equation (2), i.e. $v^i\_* = L^i v^i\_*$  for all $i \in \mathcal{N}$.
> > >
> > > >(Theorem (old) 3.8 $\rightarrow$ Theorem (new) 3.7-(2)) (2) (Existence and uniqueness of optimal value function)
> > > There exists a unique $v_* \in \mathbb{V}$ satisfying $Lv_* = v_*$, i.e. for all $i \in \mathcal{N}$, $L^i v^i\_* = v^i\_*$.
> > >
> > > >(Theorem (old) 3.9 $\rightarrow$ Theorem (new) 3.7-(3)) (3) (Robust Equilibrium (RE) and optimal value function)
> > > A joint policy $d_* = (\pi_*, \rho_*)$, where $\pi_* = (\pi^1_*, \cdots, \pi^N_*)$ and $\rho\_* = (\rho^{\tilde{1}}\_\*, \cdots, \rho^{\tilde{N}}\_\*)$, is a robust equilibrium if and only if $v^{d_*}$ is the optimal value function.
> > >
> > > **(Question 2-(3)): I also didn't follow your discussion related to Theorem 3.8 (in the revised paper) and Assumption 3.4. The claim:
> > > "In particular, under Assumption 3.4, an NE of the 2N-player EFG exists, which has been proved in Lemma A.6 in Appendix A.1."
> > > in Section 3.3 and the claim: "In particular, we have proved that Assumption 3.8-(3) holds when Assumption 3.4 holds."**
> > >
> > > These claims “In particular, under Assumption 3.4, an NE of the 2N-player EFG exists, which has been proved in Lemma A.6 in Appendix A.1” and “In particular, we have proved that Assumption 3.8-(3) holds when Assumption 3.4 holds.” are in section 3.3 to help readers understanding the convergence conditions of RMAQ. Section 3.3 is to introduce the robust multi-agent Q-learning algorithm (RMAQ).
> > >
> > > These discussions are right after Theorem 3.9 and are related to Theorem 3.9 and Assumption 3.8. They are not related to Theorem 3.8 and Assumption 3.4.

---

> > > ### Author Response · Authors · 2022-12-06
> > > **2nd Round Response to Reviewer dBqZ (Question 2-(1)-Continue) [3/7]**
> > >
> > >  While in the new version, we first prove NE existence for EFGs in Lemma A.6 and construct the relationship between an NE for the constructed EFGs $(v^1_*, \cdots, v^N_*, -v^1_*, \cdots, -v^N_*)$ and an RE for an MG-SPA whose optimal value functions are $v^1_*, \cdots, v^N_*$ through Lemma A.7 and its corollary. Lemma A.6 assumes Assumption 3.4 and in particular depends on 3.4-(2), Finite state and action spaces and 3.4-(5), common reward functions. Lemma A.7 assumes Assumption 3.4 and in particular depends on 3.4-(4), bijection perturbation function.
> > >
> > > >(Lemma (new) A.6) Suppose $v^1 = \cdots = v^N$, and $S, A$ are finite. An NE $(\lambda_*, \chi_*)$ of the EFG based on $(v^1, \cdots, v^N, -v^1, \cdots, -v^N)$ exists.
> > >
> > >  >Lemma (new) A.7) Suppose $f$ is a bijection when $s^i$ is fixed for all $i = 1, \cdots, N$. For an EFG $(v^1, \cdots, v^N, -v^1, \cdots, -v^N)$ with an NE $(\lambda_*, \chi_*)$, we call a joint policy $(\pi\_\*^v, \rho\_\*^v)$ as the joint policy implied from the NE $(\lambda\_\*, \chi\_\*)$ , where $\rho\_\*^v(b|s)=\lambda\_\*(\tilde{s}=f_s(b)|s), \pi\_\*^v(a|\tilde{s} = f_s(b))=\chi\_\*(a|\tilde{s})$. The joint policy  $(\pi_*^v, \rho_*^v)$ satisfies $L^i v^i(s) = r^i_{(\pi_*^v, \rho_*^v)}(s) + \gamma \sum_{s^\prime \in S} p_{(\pi_*^v, \rho_*^v)}(s^\prime | s) v^i(s^\prime)$ for all $s \in S$.
> > >
> > > Then in Proposition 3.5 and its proof, we replace the NE existence assumption with assumption 3.4 and applying Lemma A.6 to get the NE existence.
> > >
> > > > (Proposition (new) 3.5) Suppose $0 \leq \gamma < 1$ and Assumption 3.4 hold. Then ${L}$ is a contraction mapping on $\mathbb{V}$.
> > >
> > > > (Partial proof of proposition (new) 3.5) Given Assumption 3.4, these two EFGs $(u^1, \cdots, u^{N}, -u^1, \cdots, -u^{N})$, $(v^i, \cdots, v^{N}, -v^i, \cdots, -v^{N})$ both have at least one mixed Nash Equilibrium according to Lemma A.6.
> > >
> > > That’s why the new version of assumption 3.4 has additional items. The proof order and presentations are modified but the techniques used in the proof and main texts of proof remain the same. The conditions of existence of RE for an MG-SPA in the new version are the same as in the old version.

---

> > > ### Author Response · Authors · 2022-12-06
> > > **2nd Round Response to Reviewer dBqZ (Question 2-(1)) [2/7]**
> > >
> > > **(Question 2-(1)): The current set of theoretical results appears to be too restrictive, so it's not clear to me how significant the theoretical contribution are. It would also be nice if the authors actually acknowledge that some of their proofs/results had mistakes and/or were incomplete. For example, the claim in the response: "We have fixed the typo in the third inequality and added more explanations in the proof" doesn't reflect the fact that the new version of Proposition 3.5 is not the same as the old one: the new one requires assumption 3.4, and the new version of this assumption has additional conditions.**
> > >
> > > Assumption 3.4 was used in the proof in the old version but was not stated in the proposition due to our negligence. We modified the statement of proposition 3.5 to make it more clear and avoid misunderstandings. In the old version, the proof of Proposition 3.5 relies on the fact that at least one NE exists for the corresponding EFGs; then this fact was proved later in the proof of Theorem 3.10. In the new version, we first prove NE existence of EFGs in Lemma A.6, then prove Proposition 3.5 by applying Lemma A.6. In summary, the proof order and presentations are modified but the techniques used in the proof and main texts of proof remain the same. The conditions of existence of RE for an MG-SPA in the new version are the same as in the old version. In the following, we'd like to explain in detail how we modify proposition 3.5 and why the new version uses the same assumptions as the old one.
> > >
> > > We first summarize our poof of Proposition (old) 3.5. We can see the partial proof of Proposition (old) 3.5 in the following, in which we suppose the NE exists for the corresponding EFGs.
> > >
> > > >(Partial proof of proposition (old) 3.5) Let $u$ and $v$ be in $\mathbb{V}$ and suppose these two EFGs $(u^1, \cdots, u^{N}, -u^1, \cdots, -u^{N})$, $(v^i, \cdots, v^{N}, -v^i, \cdots, -v^{N})$ both contain at least one mixed Nash Equilibrium.
> > >
> > > Then, given conditions of finite state and action spaces (Assumption 3.4-(2) in the new version as well as in the old version), bijection perturbation function (Assumption (new) 3.4-(4))) and common reward functions (Assumption (new) 3.4-(5)), we prove the NE existence for the corresponding EFGs in the proof of Theorem 3.10 and construct the relationship between an NE for the constructed EFGs $(v^1_*, \cdots, v^N_*, -v^1_*, \cdots, -v^N_*)$ and an RE for an MG-SPA whose optimal value functions are $v^1_*, \cdots, v^N_*$ in Lemma A.12.
> > >
> > > >(Existence of Robust Equilibrium, Theorem (old) 3.10)
> > > Assume $S^i$ is finite, $A^i$ and $B^i$ are finite for all $i \in \mathcal{N}$, $f$ is a bijection when $s^i$ is fixed, then there exists a mixed RE for a team MG-SPA.
> > >
> > > >(Proof of Theorem (old) 3.10) Through Lemma 12, we establish the connection between an NE of EFG and an RE of the corresponding MG-SPA when $f(s^i, \cdot)$ is a bijection for all $i = 1, \cdots, N$. In a team MG-SPA, we have $r^1 = \cdots = r^N$, the EFG based on $(v^1_*, \cdots, v^N_*, -v^1_*, \cdots, -v^N_*)$ degenerates to a zero-sum two-person extensive-form game with finite strategies and perfect recall. If the optimal value function $v_*$ for the MG-SPA exists, the NE of the EFG exists [4,5,6], therefore the RE of the MG-SPA exists. While we have proved that the optimal
> > > value function of an MG-SPA exists in Theorem 3.8, the existence of Robust Equilibrium thereby get
> > > proved.
> > >
> > > We cite [4,5,6] to indicate a zero-sum two-person extensive-form game with finite strategies and perfect recall has at least on mixed NE. Thus, the assumption of NE existence in proposition 3.5 is established.
> > >
> > > **Please continue reading in the next response box: 2nd Round Response to Reviewer dBqZ (Question 2-(1)-Continue) [2/7]**

---

> > > ### Author Response · Authors · 2022-12-06
> > > **2nd Round Response to Reviewer dBqZ (summarized response for questions to Theoretical results) [1/7]**
> > >
> > > >Reviewer dBqZ: Unfortunately, the response didn't help me appreciate more the contributions of this work. In fact, the new version of the theoretical results appear to be weaker, arguably because the previous versions were not correct.
> > >
> > > We thank the reviewer for the effort and time to further clarify her/his concerns. In the new version, we mainly revise the presentation of the theorems, lemmas, and assumptions, to make it more straightforward to understand the connections and relations of the main theory results.  The main results and the theoretical proof process are almost the same as the old version. We give detailed explanations in this 2nd round response.
> > >
> > > In https://openreview.net/forum?id=Rl4ihTreFnV&noteId=FmyjkLFuVXH and  https://openreview.net/forum?id=Rl4ihTreFnV&noteId=BGNSu3QU_S7 we explain why and how Assumption 3.4 was modified and why the conditions of existence of RE for an MG-SPA in the new version are the same as in the old version.
> > >
> > > In https://openreview.net/forum?id=Rl4ihTreFnV&noteId=44xqhcCfek, we show that Theorem 3.8 and 3.9 in the old version are almost the same as in the new version.
> > >
> > > In https://openreview.net/forum?id=Rl4ihTreFnV&noteId=oV6XlhqOjYx, we explain the convergence of RMAQ is not only guaranteed under Assumption 3.4.

---

> > > ### Author Response · Authors · 2022-12-06
> > > **2nd Round Response to Reviewer dBqZ (Summary)**
> > >
> > > We thank the reviewer for the effort and time to further clarify her/his concerns. This response is to guide the reviewer to easily find our responses to each question. We classify the reviewer's questions into two main questions: Question 1 and Question 2.
> > >
> > > Question 1 includes questions about the formal setting and generalizability of the results. Question 2 includes questions about the theoretical results. We identified 6 subproblems from reviewer dBqZ's response to Theoretical results. We use Question 2-(1), ..., Question 2-(6) to denote these 6 subproblems.
> > >
> > > Our responses to Question 1 are in https://openreview.net/forum?id=Rl4ihTreFnV&noteId=L0Uz6_mRLPG **(Question 1) [6/7]** and https://openreview.net/forum?id=Rl4ihTreFnV&noteId=NR27kMHin4 **(Question 1 - Continue) [7/7]**.
> > >
> > > Our responses to Question 2 can be found in:
> > >
> > > **Summarized response for questions to Theoretical results [1/7]**: https://openreview.net/forum?id=Rl4ihTreFnV&noteId=-4tOapiBaH3
> > >
> > > **Question 2-(1) [2/7]**: https://openreview.net/forum?id=Rl4ihTreFnV&noteId=FmyjkLFuVXH
> > >
> > > **Question 2-(1)-Continue [3/7]**: https://openreview.net/forum?id=Rl4ihTreFnV&noteId=BGNSu3QU_S7
> > >
> > > **Question 2-(2,3) [4/7]**: https://openreview.net/forum?id=Rl4ihTreFnV&noteId=44xqhcCfek
> > >
> > > **Question 2-(4,5,6) [5/7]**: https://openreview.net/forum?id=Rl4ihTreFnV&noteId=oV6XlhqOjYx
> > >
> > > In summary, there are 7 response boxes in this 2nd round response. Thank you so much for reading this summary!

---

> > > > ### Comment · Reviewer_dBqZ · 2022-12-13
> > > > **Thank you for the additional clarifications**
> > > >
> > > > Thank you for providing the detailed comments, I appreciate them. The latest response partly addresses my concerns, but some of the claims are still not clear to me. I remain to believe that for this setting history dependent policies are important — the empirical results that the authors provided in the response seem to suggest this as well. What confuses me is that the authors’ claim in the latest response “Then the subsequent theoretical analysis hold… ” for policies that take into account history, and the argument is based on introducing true historical states. But I don’t see why this claim immediately follows, e.g., the current set of results rely on the assumption that state and action spaces are finite, whereas state space based on historical states ($s_{h, t}$ and $\tilde s_{h, t}$) may not satisfy this assumption.
> > > >
> > > > Perhaps I’m missing something, but at the moment, I don’t feel comfortable with the authors just adding a remark stating that their theory generalizes, unless adequate proofs are provided. In my opinion, the whole formal setting should be based on history dependent policies, and the experiments should reflect this (e.g., by using a policy space akin to the one used in the latest response). Unfortunately, this may require a major revision of the paper. Hence, I will keep my score as it is.

---

> > > > > ### Author Response · Authors · 2022-12-13
> > > > > **Thank you for your active response and furture clarification about historical states**
> > > > >
> > > > > We would like to thank the reviewer for the quick response to our clarifications and the active engagement through this period! We would like to clarify two main points here: (1). "Then the subsequent theoretical analysis hold… " for policies that take into account history, and the argument is based on introducing perturbed historical states for the agents, and the true historical states are only known by the adversaries. We do not assume true historical states for the agents. This setting of the agents get perturbed states and the adversaries get true states is the same as our original MG-SPA problem definition. (2). In our response with historical states, we defined a concatenated state consists of the latest h states, and h is a finite number. Hence, the state space is still finite. This definition of a finite length of historical state is also consistent with the algorithm part and what is a practical policy, for instance, we use h=4 in experiments and show the evaluation results.

---

> ### Author Response · Authors · 2022-11-18
> **Response to Reviewer dBqZ (Part-2)**
>
> **(2) Question to the proof of proposition 3.5:**
>
> We thank the reviewer for these valuable comments. We have fixed the typo in the third inequality and added more explanations in the proof. The third inequality follows from the definition of Nash Equilibrium [1]: under a Nash equilibrium, a player does not gain anything from unilaterally deviating from its current strategy, when the other players keep their strategies unchanged. To make the proof clear, we have added all 4 inequalities used in the proof. Please check the blue text in the revised paper, in particular subsection A.2.
>
> \begin{align}
>     r^i_{(\pi_*^u, \rho_*^v)}(s) + \gamma \sum_{s^\prime \in S} p_{(\pi_*^u, \rho_*^v)}(s^\prime | s) v^i(s^\prime) \leq L^i v^i(s) \leq r^i_{(\pi_*^v, \rho_*^u)}(s) + \gamma \sum_{s^\prime \in S} p_{(\pi_*^v, \rho_*^u)}(s^\prime | s) v^i(s^\prime),
> \end{align}
> \begin{align}
>     r^i_{(\pi_*^v, \rho_*^u)}(s) + \gamma \sum_{s^\prime \in S} p_{(\pi_*^v, \rho_*^u)}(s^\prime | s) u^i(s^\prime) \leq L^i u^i(s) \leq r^i_{(\pi_*^u, \rho_*^v)}(s) + \gamma \sum_{s^\prime \in S} p_{(\pi_*^u, \rho_*^v)}(s^\prime | s) u^i(s^\prime), \nonumber
> \end{align}
>
>
> [1] Facchinei, Francisco, and Christian Kanzow. "Generalized Nash equilibrium problems." 4or 5.3 (2007): 173-210.
>
> ------
>
> **(3) Question to the relationship between Theorem 3.8, 3.9 and Theorem 3.10:**
>
> We thank the reviewer for these comments. First, to prove Theorem 3.8 (Proposition 3.5), we assume the corresponding extensive-form games have at least one mixed NE. In Theorem 3.10, we actually give the condition under which the corresponding extensive-form game has at least one mixed NE. We have modified the text to make our theoretical results more clear. We also modify the organization of the proof to make the proof more readable. In the revised paper, we first give all assumptions (assumption 3.4) considered in the paper, then we prove proposition 3.5 and theorem 3.7 given assumption 3.4 held.
>
> The modified text is in blue. Please check Theorem 3.7 and Appendix A.3 (Proof of Theorem 3.7) in the revised paper.
>
> ------
>
> **(4) Question to compare Assumption 3.11-(c) to Assumption 3 from Hu & Wellman (2003)):**
>
> We thank the reviewers for this constructive comment. Assumption 3 from Hu & Wellman [1] is:
>
> Assumption 3:  One of the following conditions holds during learning:
>
> Condition A. Every stage game (Q1(s),...,Qn(s)), for all t and s, has a global optimal point, and agents’ payoffs in this equilibrium are used to update their Q-functions.
>
> Condition B. Every stage game (Q1(s),...,Qn(s)), for all t and s, has a saddle point, and agents’ payoffs in this equilibrium are used to update their Q-functions.
>
> Our Assumption 3.11-(3) (now Assumption 3.8-(3) in the revised paper) is:
>
> An NE of the $2N$-player EFG based on $(q_t^1, \cdots, q_t^N, -q_t^1, \cdots, -q_t^N)$ exists at each iteration $t$.
>
> Both of these assumptions require the existence of an NE during the algorithm’s learning process. However, they consider different games. Condition A holds for common-payoff stochastic games. Condition B holds, for example, for zero-sum stochastic games. Our Assumption 3.11-(3) is an assumption for the NE existence in extensive-form games (EFGs). These kinds of assumptions (the existence of NE for some games during the algorithms learning process) have been widely accepted and used by the research community. In practice, this assumption is found to be not necessary for the learning algorithm to converge [1,2]. In particular, we have proved that Assumption 3.8-(3) holds when Assumption 3.4 holds. You can check the proof in the revised paper, specifically Lemma A.6, Appendix A.1. We also revised the paper accordingly, please see the blue text in the revised paper.
>
> [1]  J. Hu and M. P. Wellman, “Nash q-learning for general-sum stochastic games,” Journal of machine learning research, vol. 4, no. Nov, pp. 1039–1069, 2003.
>
> [2] Yang, Yaodong, et al. "Mean field multi-agent reinforcement learning." International conference on machine learning. PMLR, 2018.

---

> ### Author Response · Authors · 2022-11-18
> **Response to Reviewer dBqZ (Part-1)**
>
> **(1) Question to the optimization framework:**
>
> We thank the reviewer’s time and efforts in reviewing our manuscript. To keep us on the same page, we first summarize the reviewer’s question about the solution concept: the optimal robust solution/optimization framework should be history-dependent while the current solution concept defined in Definition 3.2 is Markovian. Our response covers two aspects. First, the current solution concept can be extended to be history-dependent. Second, finding a global optimal history-dependent policy is still a challenging and open problem even for single-agent POMDPs. It remains challenging to provide solid theoretical analysis about the conditions of getting an optimal or sub-optimal history-dependent policy for MARL under state perturbations. We believe it is worth a new paper and is of future interest to work on the theories and algorithms of history-dependent policies, instead of solving them in this work. We explain the details as follows.
>
> (a) Our theory and algorithm are generalizable in which Markovian policies can be extended to history-dependent policies:
>
> Our algorithms can be adapted to history-dependent policies by using recent observations as the policy input. For example, DQN [1] maps history–action pairs to scalar estimates of Q-value. It uses the history (4 most recent frames) of the states and the action as the inputs of the neural network.
>
> (b) Partially observable Markov decision processes (POMDPs) are extremely challenging [4]:
>
> Finding a globally optimal policy is a challenging and open task even in single-agent POMDP. Even for planning in POMDPs, when the model is fully known[8], this problem is computationally intractable in general, and most of the theoretical results need strong assumptions [3,4,9]. Since POMDPs lack the Markovian property, an optimal policy should depend on the entire history of actions and observations. This is called the Curse of History: an optimal policy could take exponential space to describe since the number of possible action and observation histories depends exponentially on the horizon [11]. Papadimitriou and Tsitsiklis have shown that computing an optimal policy is PSPACE-hard [3]. Recently, Golowich et al.proposed a quasipolynomial-time algorithm for planning in (one-step) observable POMDPs under certain strict assumptions. It implies that in general POMDPs, it is not true that near-optimal policies admit quasi-succinct descriptions [8]. When the model is unknown, finding an optimal policy not only suffers from computational issues but also issues of exploration and sample efficiency. Recent theoretical work on learning POMDPs either assumes access to an oracle [6,7] that solves the planning problem or requires very strong assumptions [5]. It remains challenging to provide solid theoretical analysis about the conditions of getting an optimal or sub-optimal history-dependent policy for MARL under state perturbations. We believe it is worth a new paper and is of future interest to work on the theories and algorithms of history-dependent policies, instead of solving them in this work.
>
> [1] Mnih, Volodymyr, et al. "Human-level control through deep reinforcement learning." nature 518.7540 (2015): 529-533.
>
> [2] Meuleau, Nicolas, et al. "Solving POMDPs by searching the space of finite policies." arXiv preprint arXiv:1301.6720 (2013).
>
> [3] Papadimitriou, Christos H., and John N. Tsitsiklis. "The complexity of Markov decision processes." Mathematics of operations research 12.3 (1987): 441-450.
>
> [4] Mundhenk, Martin, et al. "Complexity of finite-horizon Markov decision process problems." Journal of the ACM (JACM) 47.4 (2000): 681-720.
>
> [5] Krishnamurthy, Akshay, Alekh Agarwal, and John Langford. "PAC reinforcement learning with rich observations." Advances in Neural Information Processing Systems 29 (2016).
>
> [6] Jin, Chi, et al. "Sample-efficient reinforcement learning of undercomplete POMDPs." Advances in Neural Information Processing Systems 33 (2020): 18530-18539.
>
> [7] Azizzadenesheli, Kamyar, Alessandro Lazaric, and Animashree Anandkumar. "Reinforcement learning of POMDPs using spectral methods." Conference on Learning Theory. PMLR, 2016.
>
> [8] Golowich, Noah, Ankur Moitra, and Dhruv Rohatgi. "Planning in observable POMDPs in quasipolynomial time." arXiv preprint arXiv:2201.04735 (2022).
>
> [9] Littman, Michael L., Judy Goldsmith, and Martin Mundhenk. "The computational complexity of probabilistic planning." Journal of Artificial Intelligence Research 9 (1998): 1-36.
>
> [11] Pineau, Joelle, Geoffrey Gordon, and Sebastian Thrun. "Anytime point-based approximations for large POMDPs." Journal of Artificial Intelligence Research 27 (2006): 335-380.

---

### Official Review · Reviewer_wL7B · 2022-10-29

**Confidence:** 4
**Correctness:** 4
**Technical Novelty And Significance:** 3
**Empirical Novelty And Significance:** 3
**Recommendation:** 6

**Clarity, Quality, Novelty And Reproducibility:**

Clarity: The paper has a clear motivation of the studied problem, and all the assumptions and main theoretical results are stated clearly.

Quality: The formulation of the MARL problem with state uncertainty into a Markov game, together with the structural results on existence of equilibrium is one of the key and novel results in this paper, which I like the most. The experimental results on the actor-critic type method also look appealing.

Originality: Both the theoretical and experimental part of this paper looks original.

**Strength And Weaknesses:**

Strength:

I appreciate the paper's effort in modeling the state uncertainties as part of the Markov game and proving the existence of the equilibrium before proposing algorithms to find it. The motivation for studying state uncertainties is also clear and well articulated. Though the methods itself do not come with finite-time guarantees, having asymptotic convergence looks good enough for this challenging problem, and the empirical evidence provided in Section 4 looks convincing. I also appreciate the author's effort in highlighting important core arguments before presenting the main theorems in Section 3.2.

Weakness:

There are some assumptions being posed with concrete context provided to the reader. For example, in Assumption 3.11, the last condition is posed without any explanation on why this is needed for the convergence of the RMAQ method. It is not immediate to me why this condition relates to the convergence of the Q-learning type method.

Minor Comment:
Can the author comment on the per-iteration complexity of implementing the RMAQ method? Specifically, how does it scale with the number of agents in the system (does it suffer from the exponential blow-up in the state-action space)?

**Summary Of The Paper:**

This paper studies the problem of multi-agent RL under state uncertainty, by formulating the state uncertainties as an adversary and formulating the problem into a Markov game. Sufficient conditions are established under which an equilibrium exists for such a Markov game. Furthermore, both a value-based method and a policy-based method are proposed in search of such an equilibrium. Numerical experiments are also conducted to demonstrate the effectiveness of the proposed methods.

**Summary Of The Review:**

This paper provides a rigorous formulation to a problem (MARL with state uncertainties) that is gaining interest in recent literature. The proposed Q-learning type method, although not scalable in its current form, does come with convergence guarantees. The proposed actor-critic type method enjoys favorable empirical performances. I believe this paper contains enough original contributions to this field.

---

> ### Author Response · Authors · 2022-11-18
> **Response to Reviewer wL7B**
>
> **(1) Question to Assumption 3.11-(3):**
>
> We thank the reviewer for reviewing our manuscript. Assumption 3.11-(3) (now Assumption 3.8-(3) in the revised paper, and we use the new assumption orders in the following text.) provides a guarantee that an NE policy of Extensive-form games exists during the RMAQ algorithm’s training process such that can be used to update the action value function (Q-value). Details about Q-value updating can be found in Equation (3) in the revised paper. In particular, we have proved that Assumption 3.8-(3) holds when Assumption 3.4 holds. You can check the proof in the revised paper, specifically Lemma A.6, Appendix A.1. We also revised the paper accordingly, please see the blue text in the revised paper.
>
> ------
>
> **(2) Question to the complexity and scalability of RMAQ:**
>
> According to the descriptions of the tabular RMAQ algorithm, each learning agent has to maintain $N$ action-value functions. The total space requirement is $N|S||A|^N|B|^N$ if $|A^1| =  |A^2| = \cdots = |A^N|, |B^1| = |B^2|= \cdots = |B^N|$. This space complexity is linear in the number of joint states, polynomial in the number of agents' joint actions and adversaries' joint actions, and exponential in the number of agents. The computational complexity is mainly related to algorithms to solve an Extensive-form game [5]. However, even for general-sum normal-form games, computing an NE is known to be PPAD-complete, which is still considered difficult in game theory literature [1, 2, 3, 4]. This computation complexity concern is inevitable for value-based multi-agent reinforcement learning algorithms, which motivates us to further derive the policy gradient and propose our robust multi-agent actor-critic (RMAAC) algorithm for high-dimensional state-action scenarios. And as a first attempt to the theoretical analysis for the problem of multi-agent reinforcement learning with state uncertainty, our RMAQ algorithm provides a prototype for finding out the solution (the Robust Equilibrium), with convergence guarantee under certain conditions.
>
> We also revised the paper accordingly, please check the revision paper.
>
> [1] Daskalakis, Constantinos, Paul W. Goldberg, and Christos H. Papadimitriou. "The complexity of computing a Nash equilibrium." SIAM Journal on Computing 39.1 (2009): 195-259.
>
> [2] Chen, Xi, Xiaotie Deng, and Shang-Hua Teng. "Settling the complexity of computing two-player Nash equilibria." Journal of the ACM (JACM) 56.3 (2009): 1-57.
>
> [3]  V. Conitzer and T. Sandholm, “Complexity results about nash equilibria,” arXiv preprint cs/0205074, 2002.
>
> [4]  K. Etessami and M. Yannakakis, “On the complexity of nash equilibria and other fixed points,” SIAM Journal on Computing, vol. 39, no. 6, pp. 2531–2597, 2010.
>
> [5] Kroer, Christian, et al. "Faster algorithms for extensive-form game solving via improved smoothing functions." Mathematical Programming 179.1 (2020): 385-417.

---

> > ### Comment · Reviewer_wL7B · 2022-11-24
> > **Thanks for clarifications and changes to the paper**
> >
> > I would like to thank the authors for providing detailed response to my original comments.
> >
> > I found author's clarification on the complexity of RMAQ convincing, by pointing our the fundamental hardness of computation for large number of agents.
> >
> > The revision of the paper to include the core definitions, and the additional discussions on the assumptions in Section 3 are also appreciated.
> >
> > I intend to keep my score, favoring the acceptance of this paper.

---

### Public Comment · ~Ziyuan_Zhou1 · 2022-11-15
**Comments on this paper**

In this paper, the derivations of all the theorems are logical and elegant. I'm very interested in this work since the definition of MG-SPA seems to be very similar to that of the state-adversarial stochastic game (SaSG) in our previous work [1]. Our work is about the extension of SA-MDP to multi-agent reinforcement learning. We prove that under the joint optimal adversarial perturbation, the Nash equilibrium of SaSG may not always exist and analyze the feasibility of adversarial perturbations. This paper defines robust equilibrium and analyses the existence of robust equilibrium. I hope to have more discussions with you.
[1] Zhou Z, Liu G. RoMFAC: A Robust Mean-Field Actor-Critic Reinforcement Learning against Adversarial Perturbations on States[J]. arXiv preprint arXiv:2205.07229, 2022.

---

### Author Response · Authors · 2022-11-18
**General Response to all reviewers and AC.**

We have uploaded a revision of the manuscript, which incorporates all the comments, suggestions, and clarifications during the review process. We have also added experimental results to investigate the performance of the proposed robust MARL method in the following situations (1) under different types of attacks, (2) using different values of variances, and (3) using different constraint parameters, as asked by reviewers (see Appendix C.2.4 - C.2.8, figure 15 - 29).

We thank the reviewers and AC for the precious feedback and remain available for any further questions or discussion.

---

### Comment · Area_Chair_vJ8m · 2022-11-24
**Reviewers - please engage**

Dear reviewers,

We are now approaching the end of the discussion period and so far nobody has engaged in discussion with the authors.
The authors both updated the paper and provided detailed responses to the reviews. Please reply, clarifying whether your concerns were addressed and if not, why not.
Also, if your concerns were indeed addressed either update your score or clearly state why you still believe the score is appropriate.

Many thanks for making this conference a success and for taking your role seriously.

AC

---

### Author Response · Authors · 2023-06-12
**To readers**

Thank you for your interest in this paper and the topic of robust MARL, this work has been **published** on **TMLR (Transactions on Machine Learning Research)**. Please go to this link for the final version:  https://openreview.net/forum?id=CqTkapZ6H9 and get source code through: https://github.com/SihongHo/Robust_MARL_with_State_Uncertainty

And welcome to **cite** our paper using the following **bibtex**:

@article{

he2023robust,

title={Robust Multi-Agent Reinforcement Learning with State Uncertainty},

author={Sihong He and Songyang Han and Sanbao Su and Shuo Han and Shaofeng Zou and Fei Miao},

journal={Transactions on Machine Learning Research},

issn={2835-8856},

year={2023},

url={https://openreview.net/forum?id=CqTkapZ6H9},

note={}
}

---

### Decision · Program_Chairs · 2023-01-20

**Decision:**

Reject

**Justification For Why Not Higher Score:**

There were some unresolved reviewer concerns from one of the reviewers regarding the specific assumptions required for the theory of the paper to hold.
A major open concern is around history dependent policies, which is currently not at all covered in the paper and could potentially change both the theory and the experimental results.
Specifically, in the updated version of the paper, “h” is now treated as a parameter and all the assumptions have to be satisfied w.r.t. this parameter. E.g., the assumption 3.4.(4), which appears to be critical for the theory, making the theory even more restrictive, if for nothing else then because the "effective" state space becomes larger.
Another unresolved / unclear aspect is that it now appears that the agents, including the attacker, may need to coordinate on h; at least this is what I would now conclude from the new definitions of value functions.

So to summarise: To address some of the reviewers’ concerns the authors had to make substantial changes to the paper which in turn created more follow-up questions. Some of these are substantial enough to warrant a new review of the entire work. While this might seem unfortunate I am optimistic that this process will strengthen the paper overall.


**Justification For Why Not Lower Score:**

NA

**Metareview: Summary, Strengths And Weaknesses:**

This paper introduces a theoretical framework for robustifying multi-agent reinforcement learning against adversarial perturbations of the state. The topic is highly relevant and most of the reviewers saw clear merit in the theoretical contributions of the paper. There is also some amount of experimental work. The paper contains both an extensive amount of theory and an empirical method that uses a novel robust policy gradient for policy improvement.

There was a considerable amount of discussion between the authors and the reviewers that managed to address some of the concerns. However, the changes introduced by the authors also led to follow-up questions and ultimately there remained sufficient unresolved questions (see in particular “Justification For Why Not Higher Score” below) for me to recommend a resubmission of the work at a future venue. I do think the paper could be included if there is sufficient space, assuming that the open issues can be addressed in the camera ready copy or through the scientific community in follow-up work.



**Summary Of Ac-Reviewer Meeting:**

There were some unresolved reviewer concerns from one of the reviewers regarding the specific assumptions required for the theory of the paper to hold.
A major open concern is around history dependent policies, which is currently not at all covered in the paper and could potentially change both the theory and the experimental results.
Specifically, in the updated version of the paper, “h” is now treated as a parameter and all the assumptions have to be satisfied w.r.t. this parameter. E.g., the assumption 3.4.(4), which appears to be critical for the theory, making the theory even more restrictive, if for nothing else then because the "effective" state space becomes larger.
Another unresolved aspect is that it now appears that the agents may need to coordinate on h, including the attacker; at least this is what I would now conclude from the new definitions of value functions.


So to summarise: To address some of the reviewers’ concerns the authors had to make substantial changes to the paper which in turn created more follow-up questions. Some of these are substantial enough to warrant a new review of the entire work. While this might seem unfortunate I am optimistic that this process will strengthen the paper overall.